# Is a Modular Architecture Enough?

**Sarthak Mittal**[†], **Yoshua Bengio**, **Guillaume Lajoie**
Mila, Université de Montréal

## Abstract

Inspired from human cognition, machine learning systems are gradually revealing advantages of sparser and more modular architectures. Recent work demonstrates that not only do some modular architectures generalize well, but they also lead to better out-of-distribution generalization, scaling properties, learning speed, and interpretability. A key intuition behind the success of such systems is that the data generating system for most real-world settings is considered to consist of sparsely interacting parts, and endowing models with similar inductive biases will be helpful. However, the field has been lacking in a rigorous quantitative assessment of such systems because these real-world data distributions are complex and unknown. In this work, we provide a thorough assessment of common modular architectures, through the lens of simple and known modular data distributions. We highlight the benefits of modularity and sparsity and reveal insights on the challenges faced while optimizing modular systems. In doing so, we propose evaluation metrics that highlight the benefits of modularity, the regimes in which these benefits are substantial, as well as the sub-optimality of current end-to-end learned modular systems as opposed to their claimed potential.[1]

## 1 Introduction

Deep learning research has an established history of drawing inspiration from neuroscience and cognitive science. From the way hidden units combine afferent inputs, to how connectivity and network architectures are designed, many breakthroughs have relied on mimicking brain strategies. It is no surprise then that modularity and attention have been leveraged, often together, in artificial networks in recent years (Bahdanau et al., 2015; Andreas et al., 2016; Hu et al., 2017; Vaswani et al., 2017; Kipf et al., 2018; Battaglia et al., 2018; Goyal et al., 2019, 2021), with impressive results. Indeed, work from cognitive neuroscience (Baars, 1997; Dehaene et al., 2017) suggests that cortex represents knowledge in a modular way, with different such modules communicating through the bottleneck of working memory (where very few items can simultaneously be represented), in which content is selected by attention mechanisms. In recent work from the AI community (Bengio, 2017; Goyal & Bengio, 2020), it was proposed that these characteristics could correspond to meaningful inductive biases for deep networks, i.e., statistical assumptions about the dependencies between concepts manipulated at the higher levels of cognition. Both sparsity of the dependencies between these high-level variables and the decomposition of knowledge into recomposable pieces that are as independent as possible (Peters et al., 2017; Bengio et al., 2019; Goyal & Bengio, 2020; Ke et al., 2021) would make learning more efficient. Out-of-distribution (OoD) generalization would be facilitated by making it possible to sequentially compose the computations performed by these modules where new situations can be explained by novel combinations of existing concepts.

Although a number of recent results hinge on such modular architectures (Graves et al., 2014; Andreas et al., 2016; Hu et al., 2017; Vaswani et al., 2017; Kipf et al., 2018; Santoro et al., 2018; Battaglia et al., 2018; Goyal et al., 2019, 2021; Locatello et al., 2020; Mittal et al., 2020; Madan et al., 2021),

---

[†]Correspondence authors sarthmit@gmail.com
[1]Open-sourced implementation is available at https://github.com/sarthmit/Mod_Arch

the abundance of tricks and proposed architectural modifications makes it challenging to parse real, usable architectural principles. It is also unclear whether the performance gains obtained by such Mixture-of-Experts (MoE) based modular systems are actually due to good specialization, as is often claimed, or due to other potential confounding factors like ease of optimization.

In this work, we extend the analysis from Rosenbaum et al. (2019); Maziarz et al. (2019); Cui & Jaech (2020); Csordás et al. (2020) and propose a principled approach to evaluate, quantify, and analyse common ingredients of modular architectures, supported by either standard MLP-like connectivity, recurrent connections or attention (Bahdanau et al., 2015; Vaswani et al., 2017) operations. To do so, we develop a series of benchmarks and metrics aimed at probing the efficacy of a wide range of modular networks, where computation is factorized. This reveals valuable insights and helps identify not only where current approaches succeed but also *when and how they fail*. Whereas previous work on disentangling (Bengio, 2013; Higgins et al., 2016; Kim & Mnih, 2018) has focused on factoring out the different high-level variables that explain the data, here we focus on disentangling system modules from each other, the structural ingredients of network that can facilitate this factorization, and how such ingredients relate to the data generating distributions being parsed and processed.

Given the recent increased interest in sparse modular systems (Rahaman et al., 2021; Fedus et al., 2021; Du et al., 2021; Mittal et al., 2021), we believe that this work will provide a test-bed for investigating the workings of such models and allow for research into inductive biases that can push such models to achieve good specialization. Through detailed experiments and evaluation metrics, we make the following observations and contributions:

- We develop benchmark tasks and metrics based on probabilistically selected rules to quantify two important phenomena in modular systems, the extent of *collapse* and *specialization*.
- We distill commonly used modularity inductive biases and systematically evaluate them through a series of models aimed at extracting commonly used architectural attributes (*Monolithic*, *Modular*, *Modular-op* and *GT-Modular* models).
- We find that specialization in modular systems leads to significant boosts in performance when there are many underlying rules within a task, but not so much with only few rules.
- We find standard modular systems to be often sub-optimal in both their capacity on focusing on the right information as well as in their ability to specialize, suggesting the need for additional inductive biases.

## 2 Notation / Terminology

In this paper, we study how a family of modular systems performs on a common set of tasks, prescribed by a synthetic data generating process which we call *rule-based data*. Below, we introduce the notation for key ingredients: (1) *rules* and how they form *tasks*, (2) *modules* and how they can take different *model architectures*, (3) *specialization* and how we evaluate models. We refer the reader to Figure 1 for an illustration of our setup.

**Rules.** To properly understand modular systems and analyze their benefits and shortcomings, we consider synthetic settings that allow fine-grained control over different aspects of task requirements. In particular, operations must be learned on the data-generating distri-

Figure 1: **Illustration of modularity evalutation framework.** Task configurations define the rules in the data-generating process while model parameters define the kind of model to be trained on it.

bution illustrated in Equations 1-3, which we also refer to as *rules*. Details about the exact operations used in experiments are described in Section 3.

Given this distribution, we define a rule to be an expert of this distribution, that is, rule $r$ is defined as $p_y(\cdot \mid \mathbf{x}, \mathbf{c} = r)$ where $\mathbf{c}$ is a categorical variable representing context, and $\mathbf{x}$ is an input sequence. For example, consider $\mathbf{x} = (1, 2)$ and $\mathbf{c}$ to select between addition and multiplication. Then, depending on $\mathbf{c}$, the correct

$$\mathbf{c} \sim \text{Categorical}(\cdot) \quad (1)$$
$$\mathbf{x} \sim p_x(\cdot) \quad (2)$$
$$\mathbf{y} \mid \mathbf{x}, \mathbf{c} \sim p_y(\cdot \mid \mathbf{x}, \mathbf{c}). \quad (3)$$

output should be either $\mathbf{y} = 3$ or $\mathbf{y} = 2$. More details about the specifics of these data distributions are presented in Section 3. Systems will be trained to infer $\mathbf{y}$ given $\mathbf{c}$ and $\mathbf{x}$. This simple setup is meant to capture context-dependent tasks on variable data distributions, e.g. reasoning according to different features (e.g. shape, color, etc.). However, unlike such complex systems, ground truth knowledge of required operations is known for our synthetic task, allowing for deeper quantitative analysis.

**Tasks.** A task is described by the set of rules (data-generating distribution) illustrated in Equations 1-3. Different sets of $\{p_y(\cdot \,|\, \mathbf{x}, \mathbf{c})\}_{\mathbf{c}}$ imply different tasks. For a given number of rules, we train models on multiple tasks to remove bias towards any particular task.

**Modules.** A modular system comprises a set of neural network modules, each of which can contribute to the overall output. One can see this through the functional form $\mathbf{y} = \sum_{m=1}^{M} p_m \, \mathbf{y}_m$, where $\mathbf{y}_m$ denotes the output and $p_m$ the activation of the $m^{th}$ module. Details about the different modular systems are outlined in Section 4.

From this point onwards, we exclusively use *rules* to refer to the specialized components *in the data-generating process*, and *modules* to refer to the experts that are *learned by a modular system*. Further, for ease of quantitative assessment, we always set the number of modules equal to the number of rules, except when evaluating monolithic models (with a single module). Modules can be implemented in three different architectures, as described next.

**Model Architectures.** Model architectures describe the choice of architecture considered for each module of a modular system, or the single module in a monolithic system. Here we consider Multi-Layer Perceptron (MLP), Multi-Head Attentions (MHA), and Recurrrent Neural Network (RNN). Importantly, the rules (or data generating distributions) are adapted to the model architecture, and we often refer to them as such (e.g. MLP based rules). Details about the data distributions and models considered in this work are provided in Sections 3 and 4 respectively.

**Perfect Specialization.** When training modular systems on rule-based data, we would like the modules to specialize according to the rules in the data-generating distribution. Thus, there is an important need to quantify what constitutes perfect specialization of the system to the data. To allow for easier quantification, we always consider an equal number of modules and rules. However, future work should evaluate the ability of modular systems to automatically infer the required number of modules.

## 3 Data Generating Process

Since we aim to study modular systems through synthetic data, here we flesh out the data-generating processes operating based on the rules scheme described above (see Equations 1-3). We use a simple Mixture-of-Experts (MoE) Yuksel et al. (2012); Masoudnia & Ebrahimpour (2014) styled data-generating process (Mixture Distribution), where we expect different modules to specialize to the different mixture components (rules). It is important to note that this system is slightly different from the traditional flat MoE since the experts are more plug-and-play and can be composed to solve a particular problem. As an example, if we consider a mixture of recurrent systems, different tokens (time-points) in the input sequence can undergo computations according to different rules (e.g. a switching linear dynamical system), as opposed to the choice of expert being governed by the whole sequence.

We now look at more specific setups of the data-generating systems in consideration, the general template of which was outlined above. To do so, we explain the data-generating processes amenable to our three model architectures: MLP, MHA, and RNN. Additionally, each of the following tasks have two versions: regression, and classification. These are included to explore potential differences these distinct loss types may induce.

**MLP.** Here, we define the data scheme that is amenable for learning of modular MLP-based systems. In this synthetic data-generating scheme, a data sample consists of two independent numbers and a choice of rule being sampled from some distribution. Different rules lead to different linear combinations of the two numbers to give the

$$c \sim \mathcal{U}\{1, R\} \quad (4)$$
$$\mathbf{x}_1, \mathbf{x}_2 \overset{\text{iid}}{\sim} \mathcal{N}(0, \mathbf{I}) \quad (5)$$
$$\mathbf{y} = \alpha_c \mathbf{x}_1 + \beta_c \mathbf{x}_2 \quad (6)$$

output. That is, the choice of linear combination is dynamically instantiated based on the rule drawn. This is mathematically formulated in Equations 4-6, where $\alpha_c$ and $\beta_c$ are the data parameters, $\mathbf{I}$ the identity matrix and $\mathbf{y}$ denotes the label for the regression tasks and $\text{sign}(\mathbf{y})$ for the classification tasks.

Hence, the data comes from a MoE distribution where $c$ denotes which linear combination governs the conditional distribution $p_y(\cdot \,|\, \mathbf{x}_1, \mathbf{x}_2, c)$. When training modular architectures on such data, one expects each module in the trained system to specialize according to a unique rule.

**MHA.** Now, we define the data scheme that is tuned for learning in modular MHA based systems. Essentially, a MHA module can be understood through a set of searches (query-key interactions), a set of corresponding retrievals (values) and then some computation of the retrieved values, as explained by Mittal et al. (2021). Accordingly, we design the data-generating distribution with the following properties: Each rule is composed of a different notion of search, retrieval and the final

$$\mathbf{c}_n \overset{\text{iid}}{\sim} \mathcal{U}\{1, R\} \quad (7)$$

$$\mathbf{q}_{nr}, \mathbf{q}'_{nr}, \mathbf{v}_{nr}, \mathbf{v}'_{nr} \overset{\text{iid}}{\sim} \mathcal{N}(0, \mathbf{I}) \quad (8)$$

$$\mathbf{s}_n = \min_{i \neq n} d\left(\mathbf{q}_{nc_n}, \mathbf{q}_{ic_n}\right) \quad (9)$$

$$\mathbf{s}'_n = \min_{i \neq n} d\left(\mathbf{q}'_{nc_n}, \mathbf{q}'_{ic_n}\right) \quad (10)$$

$$\mathbf{y}_n = \alpha_{c_n} \mathbf{v}_{\mathbf{s}_n c_n} + \beta_{c_n} \mathbf{v}'_{\mathbf{s}'_n c_n} \quad (11)$$

linear combination of the retrieved information respectively. We mathematically describe the process in Equations 7-11, where $n = 1, ..., N$ and $r = 1, ..., R$ with $N$ as the sequence length and $R$ the number of rules. We denote the tuple $(\mathbf{q}_{nr}, \mathbf{q}'_{nr}, \mathbf{v}_{nr}, \mathbf{v}'_{nr})$ as $\mathbf{x}_n$. Further, $\mathbf{y}_n$ denotes the label for the regression tasks while for classification, we consider the categorical label to be $\text{sign}(\mathbf{y}_n)$.

Thus, we can see that $\mathbf{c}_n$ denotes the rule for the $n^{th}$ token. This rule governs which two tokens are closest to the $n^{th}$ token, demonstrated as $\mathbf{s}_n$ and $\mathbf{s}'_n$. It also governs what features are retrieved from the searched tokens, which are $\mathbf{v}_{\mathbf{s}_n c_n}$ and $\mathbf{v}_{\mathbf{s}'_n c_n}$. These retrieved features then undergo a rule-dependent linear combination (on $\mathbf{c}_n$). Here, too, when training a modular MHA architecture, we want each MHA module in the system to be able to specialize to a unique MHA rule in the data system.

**RNN.** For recurrent systems, we define a rule as a kind of linear dynamical system, where one of multiple rules can be triggered at any time-point. Mathematically, this process can be defined through Equations 12-15, where $n = 1, ...N$, with $N$ describing the sequence length. Each rule thus describes a different procedure for the update of the state $\mathbf{s}_t$ as well as the effect of the input $\mathbf{x}_t$ to the state. Thus, we can see that $\mathbf{c}_n$

$$\mathbf{c}_n \overset{\text{iid}}{\sim} \mathcal{U}\{1, R\} \quad (12)$$

$$\mathbf{x}_n \overset{\text{iid}}{\sim} \mathcal{N}(0, \mathbf{I}) \quad (13)$$

$$\mathbf{s}_n = A_{c_n} \mathbf{s}_{n-1} + B_{c_n} \mathbf{x}_n \quad (14)$$

$$\mathbf{y}_n = \mathbf{w}^T \mathbf{s}_n \quad (15)$$

denotes the rule to be used at the $n^{th}$ time-point. Further, $\mathbf{y}_n$ denotes the label for the regression tasks while for classification, we consider the labels as $\text{sign}(\mathbf{y}_n)$.

Hence, in all settings, the data comes from a MoE distribution where $c$ denotes the rule and governs the conditional $p_y(\cdot \,|\, \mathbf{x}, c)$. When training modular architectures on such data, one expects each module in the trained system to specialize according to a unique rule. Our aim is to use these synthetic rule-based data setting to study and analyse modular systems and understand whether end-to-end trained modular systems concentrate on the right information to specialize based on, i.e. based on $\mathbf{c}$, whether they do learn perfect specialization and whether perfect specialization actually helps in these settings. To properly understand this, we detail the different kinds of models considered in Section 4 as well as the different metrics proposed in Section 5 to analyse trained systems.

For this work, we limit our analysis to infinite-data regime where each training iteration operates on a new data sample Future work would perform similar analysis in the regime of limited data.

## 4 Models

Several works claim that end-to-end trained modular systems outperform their monolithic counterparts, especially in out-of-distribution settings. However, there is a lack of step-by-step analysis on the benefits of such systems and whether they actually specialize according to the data generating distribution or not. To perform an in-depth analysis, we consider four different types of models that allow for varying levels of specialization, which are: *Monolithic*, *Modular*, *Modular-op*, and *GT-Modular*. We give the formulations for each of these models below and then discuss the different analysis we can perform through them. We also illustrate these models in Table 1 and depending on the data-generating procedure described in Section 3, $f$ and $f_m$ can be implemented as either MLP, MHA or RNN cells in this work.

**Monolithic.** A monolithic system is a big neural network that takes the entire data $(\mathbf{x}, \mathbf{c})$ as input and makes

| Model | Functional Form |
|---|---|
| *Monolithic* | $\hat{\mathbf{y}} = f(\mathbf{x}, \mathbf{c})$ |
| *Modular* | $\hat{\mathbf{y}}_m, p_m = f_m(\mathbf{x}, \mathbf{c})$ 
 $\hat{\mathbf{y}} = \sum_{m=1}^{R} p_m \, \hat{\mathbf{y}}_m$ |
| *Modular-op* | $\hat{\mathbf{y}}_m = f_m(\mathbf{x}, \mathbf{c})$ 
 $\mathbf{p} = g(\mathbf{c})$ 
 $\hat{\mathbf{y}} = \sum_{m=1}^{R} p_m \, \hat{\mathbf{y}}_m$ |
| *GT-Modular* | $\hat{\mathbf{y}}_m = f_m(\mathbf{x}, \mathbf{c})$ 
 $\hat{\mathbf{y}} = \sum_{m=1}^{R} c_m \, \hat{\mathbf{y}}_m$ |

Table 1: **Functional Forms of Different Models.** Exact functional forms of the different models considered in this work, given the data $(\mathbf{x}, \mathbf{c})$. Depending on context, $f$ and $f_m$ are either MLP, MHA or RNN architectures.

predictions $\hat{\mathbf{y}}$ based on it. There is no inductive bias about modularity or sparsity explicitly baked in the system and it is completely up to back-propagation to learn whatever functional form is needed to solve the task. An example of such a system is a traditional Multi-Head Attention (MHA) based system, eg. a Transformer.

**Modular.** A modular system is composed of a number of modules, each of which is a neural network of a given architectural type (MLP, MHA, or RNN). Each module $m$ takes the data $(\mathbf{x}, \mathbf{c})$ as input and computes an output $\hat{\mathbf{y}}_m$ and a confidence score, normalized across modules into an activation probability $p_m$. The activation probability reflects the contribution of each module's output to the final output $\hat{\mathbf{y}}$ of the system. Thus, there is an explicit baked-in inductive bias of modularity but it is still up to system-wide back-propagation to figure out the right specialization. An example of such a system is a mixture of MLPs or reusable RNNs, reusable across different time/positions.

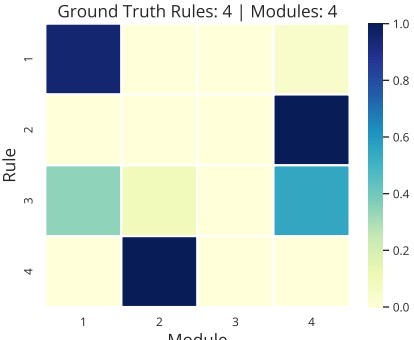

Figure 2: **Example of Collapse.** Entry $(i, j)$ denotes the activation probability of module $j$ on rule $i$. We see that Module 3 never activates, signifying collapse, while Module 4 covers two rules.

**Modular-op.** A modular-op (for *operation only*) system is very similar to the modular system with just one small difference. Instead of the activation probability $p_m$ of module $m$ being a function of $(\mathbf{x}, \mathbf{c})$, we instead make sure that the activation is decided only by the rule context $\mathbf{c}$. Hence, unlike modular systems, modular-op cannot be distracted by $\mathbf{x}$ in figuring out specialization of different modules. Even though the operation required is explicitly provided, this model still needs to learn specialization through back-propagation.

**GT-Modular.** A GT-modular system (for *ground truth*) serves as an oracle benchmark, i.e., a modular system that specializes perfectly. In particular, the activation probability $p'_m$s of modules are just set according to $c$, which is the indicator present in the data $(\mathbf{x}, \mathbf{c})$. Thus, this is a perfectly specializing system that chooses different modules sparsely and perfectly according to the different data rules.

Given enough capacity, we can see that there is a hierarchy of models based on the functions they can implement, with *GT-Modular* $\subseteq$ *Modular-op* $\subseteq$ *Modular* $\subseteq$ *Monolithic*. Put differently, models from *Monolithic* to *GT-Modular* increasingly incorporate the inductive biases for modularity and sparsity. This is proved in Appendix C by inspecting the function classes implemented by these models.

In what follows, we want to analyse the benefits of having simple end-to-end trained modular systems as opposed to monolithic ones. This can be understood through a comparison of various performance based metrics between *Monolithic* and *Modular* models, explained in the next section. This will allow us to answer if a modular architecture is always better for various distinct rule-based data generating systems. For instance, a comparison between the *Modular* and *Modular-op* models will show whether the stan-

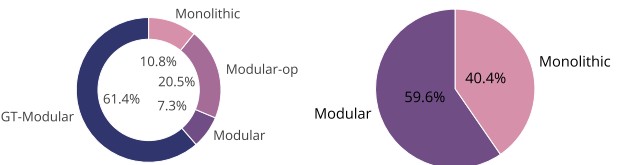

Figure 3: **Ranking Metric.** Spread indicates the number of experiments where the corresponding model did better. Plots include **Left:** All models, **Right:** Models trained without explicit rule-based module selection. Note that (a) explicit specialization (*GT-Modular*) helps, and (b) *Modular* systems outperform *Monolithic* but with small margin.

dard modular systems are able to focus on the right information and ignore the distractors in driving specialization. To study this, we will look at performance as well as collapse and specialization metrics between these class of models. A comparison between *GT-Modular* and *Modular-op* will show the benefits of having a sparse activation pattern with proper resource allocation of modules as opposed to an end-to-end learned specialization on the right information (without distractors).

Finally, we note that *GT-Modular* is a modular system which obtains perfect specialization. Through this model, we aim to analyse whether perfect specialization is in-fact important and if so, how far are typical modular systems from obtaining similar performance and specialization through end-to-end training. We now describe the metrics used for these evaluations.

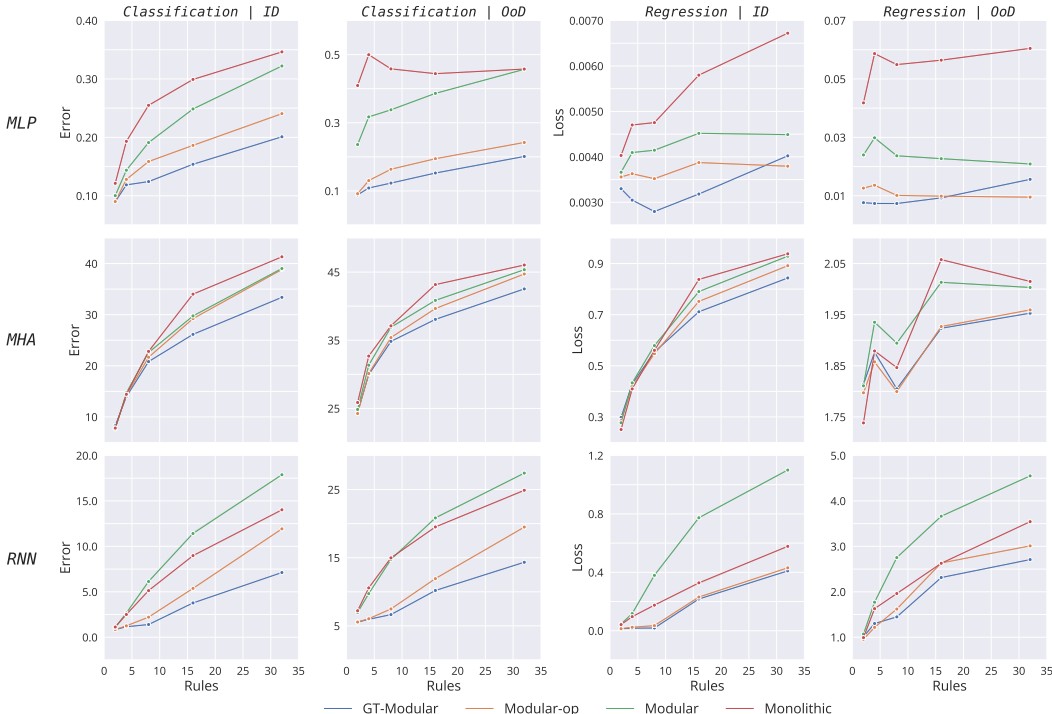

Figure 4: **Performance Results.** Performance of different models with MLP *(top row)*, MHA *(middle row)* and RNN *(bottom row)* architectures against varying number of rules, evaluated both in-distribution and out-of-distribution. Modular systems generally outperform monolithic ones *(lower is better)* but a typical end-to-end trained modular system (green) is neither able to concentrate on the right information (compare with orange) nor able to get optimal specialization (compare with blue).

## 5 Metrics

To reliably evaluate modular systems, we propose a suite of metrics that not only gauge the performance benefits of such systems but also evaluate them across two important modalities: *collapse* and *specialization*, which we use to analyse the extent of resource allocation (in terms of parameters/modules) and specialization respectively of a modular system.

**Performance.** The first set of evaluation metrics are based on performance of the models in both in-distribution as well as out-of-distribution (OoD) settings. These metrics capture how well the different models perform on a wide variety of different tasks. For classification settings, we report the classification error while for regression settings, we report the loss.

*In-Distribution.* This refers to the in-distribution performance, evaluated by looking at both the final performance as well as convergence speeds of the different models.

*Out-of-Distribution.* This refers to the OoD performance of different models. We consider very simple forms of OoD generalization: either (a) change in distribution of $\mathbf{x}$ by increasing variance, or (b) different sequence lengths, wherever the possibility presents (eg. in MHA and RNN).

**Collapse Metrics.** We propose a set of metrics *Collapse-Avg* and *Collapse-Worst* that quantify the amount of collapse suffered by a modular system. Collapse refers to the degree of under-utilization of the modules. An example of this is illustrated in Figure 2, where we can see that Module 3 is never used. We consider the setting where all the data rules are equi-probable and the number of modules in the model are set to be the same as the number of data rules, to $R$. High collapse thus refers to under-utilization of resource (parameters) provided to the model, illustrating that certain modules are never being used and concurrently meaning that certain modules are being utilized for multiple rules.

*Collapse-Avg.* Given the data-setting with $R$ equi-probable rules, and hence $R$ modules in the model, we let $p(m)$ be the marginal probability distribution of activation of module $m$. Then, we define the *Collapse-Avg* metric $C_A$ as in Equation

$$C_A = \frac{R}{R-1} \sum_{m=1}^{R} \max\left(0, \frac{1}{R} - p(m)\right) \quad (16)$$

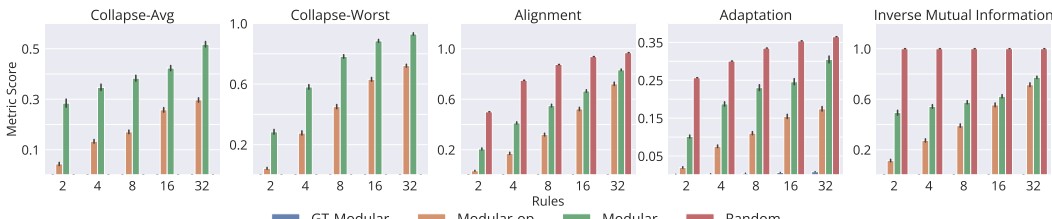

Figure 5: **Metrics against Increasing Number of Rules.** Evaluation of different modular systems through collapse (first two columns) and specialization (next three columns) metrics (*lower is better*) while varying the number of rules. We see that the problems increase with more rules.

16, where $\frac{R}{R-1}$ is for normalization. This metric captures the amount of under-utilization of all the modules of the system. A lower number is preferable for this metric, as a lower number demonstrates that all the modules are equally utilized.

*Collapse-Worst.* Given the same data and model setting as above, the *Collapse-Worst* metric $C_W$ is defined as in Equation 17. This
$$C_W = 1 - R \min_m p(m) \quad (17)$$
metric captures the amount of under-utilization of the least used module of the system. Again, a low number is preferable as it signifies that even the least used module is decently utilized by the model.

**Specialization Metrics.** To complement collapse metrics, we also propose a set of metrics, (1) *Alignment*, (2) *Adaptation* and (3) *Inverse Mutual Information* to quantify the amount of specialization obtained by the modular systems. We again consider the setting of equi-probable rules and the same number of modules and rules $R$. These metrics are aimed at capturing how well the modules specialize to the rules, that is, whether different modules stick to different rules (good specialization) or whether all modules contribute almost equally to all rules (poor specialization).

*Alignment.* Given a modular system trained on rule-based data with $R$ rules and modules, one can obtain the activation matrix $\mathbf{A}$, where
$$s_d = \min_{\mathbf{P} \in \mathbf{S}_R} d(\mathbf{A}, \mathbf{P}) \quad (18)$$
$\mathbf{A}_{rm}$ denotes $p(\text{module} = m \mid \text{rule} = r)$, that is, the probability of activation of module $m$ conditioned on rule $r$. Further, given a distance metric $d(\cdot, \cdot)$ over the space of matrices, perfect specialization can be quantified through Equation 18, where $\mathbf{S}_R$ denotes the space of permutation matrices over $R$ objects. We consider $d(\cdot, \cdot)$ as a normalized $L_1$ distance. The score $s_d$ demonstrates the distance between the activation matrix $A$ and its closest permutation matrix, with distances computed according to the metric $d(\cdot, \cdot)$. Note that $s_d \to 0$ implies that each module specializes to a *unique* rule, thereby signifying perfect specialization. Since the space of permutation matrices $\mathbf{S}_R$ grows exponentially at the rate of $\mathbf{\Theta}(R!)$, computing $s_d$ naively soon becomes intractable. However, we use the Hungarian algorithm (Kuhn, 1955) to compute it in polynomial time. This metric shows how close the learned modular system is to a perfectly specializing one, where a low score implies better specialization.

*Inverse Mutual Information.* Given $R$ as the number of rules and modules and let the joint distribution $p(m, r)$ denote the
$$S_{IMI} = 1 - \frac{1}{\log R} \mathbb{E}_{p(m,r)} \left[ \log \frac{p(m,r)}{p(m) \times p(r)} \right] \quad (19)$$
activation probability of module $m$ on rule $r$, the *Inverse Mutual Information* metric $S_{IMI}$ is defined as in Equation 19. A low inverse mutual information metric is preferable as it denotes that the modules are more specialized to the rules as opposed to multiple modules contributing to a single rule.

*Adaptation.* Let $R$ be the number of rules and modules and $\mathcal{P}$ a distribution over the $R$-dimensional simplex. Further, let $p(\cdot)$ be the distribution over rules (not equi-probable in this metric) and $q(\cdot)$ the corresponding
$$S_A = \mathbb{E}_{p \sim \mathcal{P}} \left[ \sum_{i=1}^{R} \left| p(\hat{r}_i) - q(\hat{m}_i) \right| \right] \quad (20)$$
distribution obtained over the modules. Note that the distribution $q(\cdot)$ is dependent on $p(\cdot)$. Given these distributions, we define the *Adaptation* metric $S_A$ in Equation 20, where $\hat{r}_i$ and $\hat{m}_i$ are such that $p(\hat{r}_1) \le p(\hat{r}_2) \le ... \le p(\hat{r}_R)$ and $q(\hat{m}_1) \le q(\hat{m}_2) \le ... \le q(\hat{m}_R)$ and $\mathcal{P}$ is a dirichlet distribution.

This metric can be understood as the amount by which the modules adapt (signified through the distribution $q(\cdot)$) to changes in the rule distributions (which are $p(\cdot)$ sampled from $\mathcal{P}$). The matching between the rule and module is obtained through a simple sort as defined above. A low adaptation score implies that the marginal distribution of the modules adapt well according to the distribution of the rules. That is, when a rule is weakly present in the data, there exists a module which weakly contributes in the corresponding output, averaged over multiple different rule distributions.

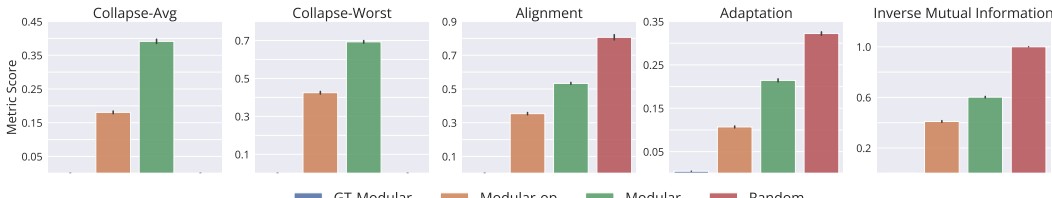

Figure 6: **Metrics for Different Models.** While end-to-end training of activation decisions leads to reduced collapse (first two columns) and better specialization (next three columns) (*lower is better*) than random activations, it is still far from a perfectly specializing system. This signifies that the models are not able to learn good specialization and actually suffer from increased collapse when learned solely through back-propagation.

To understand these metrics, note that uniform random activation patterns for the modules lead to low collapse metrics but high alignment, adaptation and inverse mutual information metrics, implying little collapse but poor specialization, as expected. On the other hand, *GT-Modular* systems necessarily lead to low collapse metrics as well as low alignment, adaptation and inverse mutual information, denoting little collapse and good specialization, which is expected since specialization is given as oracle.

## 6   Experiments

We are now ready to report experiments on the models outlined in Section 4 with associated data generation processes described in Section 3. For each level of modularity (i.e. Monolithic, Modular, Modular-op, GT-Modular), we analyse models learning over five different number of rules, ranging from few (2) to many (32), five different model capacities (number of parameters) and two different training settings, i.e. regression and binary classification. To remove any biases towards particular task parameters (e.g. $\alpha_c, \beta_c$ in Equation 6), we randomly select new rules to create five different tasks per setting and, train five seeds per task. In essence, we train $\sim$20,000 models[2] to properly analyse the benefits of modularity, the level of specialization obtained by end-to-end trained systems, the impact of number of rules and the impact of model capacity.

**Performance.** We refer the readers to Figure 3 for a compressed overview on the performance of various models. We see that GT-Modular system wins most of the times (*left*), indicating the benefits of perfect specialization. We also see that between standard end-to-end trained Modular and Monolithic systems, the former outperforms but not by a huge gap. Together, these two pie charts indicate that current end-to-end trained modular systems do not achieve good specialization and are thus sub-optimal by a substantial margin.

We then look at the specific architectural choices (MLP, MHA and RNN cells for functions $f$ and $f_m$ in Table 1) and analyse their performance and trends across increasing number of rules. Figure 4 shows that while there are concrete benefits of a perfectly specializing system (GT-Modular) or even models that know what information to drive specialization from (Modular-op), typical end-to-end trained Modular systems are quite sub-optimal and not able to realize these benefits, especially with increasing number of rules which is where we see substantial benefits of good specialization (contrast Modular vs GT-Modular and Modular-op). Moreover, while such end-to-end Modular systems do generally outperform the Monolithic ones, it is often only by a small margin.

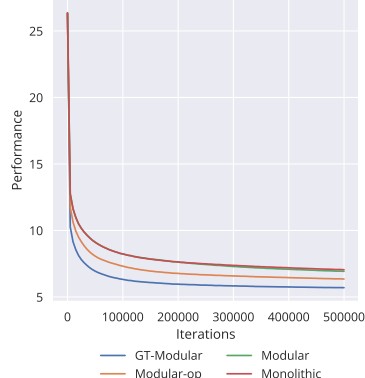

Figure 7: **Training Curve.** Averaged over different model architectures, training settings, model capacities and number of rules. We see that end-to-end trained modular systems are still far from the benefits of perfect specialization.

We also see the training pattern of different models averaged over all other settings, with the average containing error for classification and loss for regression, in Figure 7. We can see that good specialization not only leads to better performances but also faster training.

**Collapse.** We evaluate all the models on the two collapse metrics outlined in Section 5. Figure 5 shows the two collapse metrics, *Collapse-Avg* and *Collapse-Worst*, for different models against

---

[2]All models are trained on single V100 GPUs, each taking a few hours.

varying number of rules, averaged over the different model architectures (MLP, MHA and RNN), training settings (Classification and Regression), model capacities, tasks and seeds. First, we notice that a Random activation baseline and the GT-Modular system do not have any collapse, which is expected. Next, we notice that both Modular and Modular-op suffer from the problems of collapse and this problem becomes worse with increasing number of rules. Figure 6 further shows similar information averaged over the number of rules too, highlighting that Modular-op has less collapse than Modular in general. However, we still see that the problem of collapse is significant whenever back-propagation is tasked with finding the right activation patterns, especially in the regime of large number of rules. This clearly indicates the need for investigation into different forms of regularizations to alleviate some of the collapse problems.

**Specialization.** Next, we evaluate through the proposed specialization metrics in Section 5 whether the end-to-end trained modular systems actually specialize according to the data-generating distribution. Figure 5 shows the three specialization metrics, *Alignment*, *Adaptation* and *Inverse Mutual Information*, for different models against varying number of rules, again averaged over different model architectures, training settings, model capacities, tasks and seeds. As expected, we see that the Random activation baseline has poor specialization (high metrics) while the GT-Modular system has very good specialization. We further see that end-to-end trained Modular systems as well as Modular-op suffer from sub-optimal specialization, as indicated by the high metrics. As with collapse, we again see that it becomes harder to reach optimal specialization with increasing number of rules. Figure 6 shows that while Modular-op has marginally better specialization than standard Modular systems, they are indeed quite sub-optimal when compared to a perfectly specializing system, i.e. GT-Modular.

We refer the readers to Appendix D, E and F for training details as well as additional experiments regarding the effect of model sizes for MLP, MHA and RNN architectures respectively.

## 7 Conclusion and Discussion

We provide a benchmark suitable for the analysis of modular systems and provide metrics that not only evaluate them on in-distribution and out-of-distribution performance, but also on collapse and specialization. Through our large-scale analysis, we uncover many intriguing properties of modular systems and highlight potential issues that could lead to poor scaling properties of such systems.

*Perfect Specialization.* We discover that perfect specialization indeed helps in boosting performance both in-distribution and out-of-distribution, especially in the regime of many rules. On the contrary, monolithic systems often do comparatively or sometimes better when there are only a few rules, but do not rely on specialization to do so.

*End-to-End Trained Modular systems.* While Modular systems outperform Monolithic ones, the margin of improvement is often small. This is because when solely relying on back-propagation of the task-losses, these models do not discover perfect specialization. In fact, the problem of poor specialization and high collapse becomes worse with increasing number of rules. This is slightly mitigated by allowing contextual information from the task to be used explicitly, as is the case for Modular-op, but the problems still persist and get worse over large number of rules.

In summary, through systematic and extensive experiments, this work shows that modularity, when supporting good and distributed specialization (i.e. little collapse), can outperform monolithic models both in and out of distribution testing. However, we also find that although perfectly specialized solutions are attainable by modular networks, end-to-end training does not recover them, often even with explicit information about task context (as in Modular-op). Since real-world data distributions are often complex and unknown, we cannot get access to oracle networks like GT-Modular for analysis. An important conclusion is that additional inductive biases are required to learn adequately specialized solutions. These could include other architectural features to facilitate module routing, or regularization schemes (e.g. load-balancing Fedus et al. (2021)) or optimization strategies (e.g. learning rate scheduling) to promote module specialization. We refer the reader to Appendices A and B for further discussion on these exciting prospects and extensions to real-world domains. We believe the framework proposed in this work is ideal to drive research into such inductive biases and a necessary stepping stone for applications of these designs at scale.

Finally, we highlight that the use of network architectures that promote contextual specialization, such as the use of modules as studied here, could potentially promote unwanted biases when deployed in models use by the public due to collapse or ill-distributed specialization. The framework proposed in this work could help mitigate this potentially problematic impact on society.

## Acknowledgments and Disclosure of Funding

SM would like to acknowledge the support of scholarships from UNIQUE and IVADO as well as compute resources from Alliance and its regional partner organizations (ACENET, Calcul Québec, Compute Ontario, the BC DRI Group and the Prairie DRI Group) towards his research. YB and GL acknowledge the support from Canada CIFAR AI Chair Program, as well as Samsung Electronics Co., Ldt. GL acknowledges NSERC Discovery Grant [RGPIN-2018-04821].

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
