# A    Limitations and Future Work

We perform large-scale analysis on a variety of modular systems through simple *rule-based* data generating distributions. Through our analysis, we uncover several interesting insights into the regimes where modularity, sparsity and perfect specialization helps and how sub-optimal standard modular systems are in terms of collapse and specialization.

**Limitations**. While we provide an in-depth analysis of modular systems and the sub-optimalities they suffer from, we do realize that our experimentation is done from a purely synthetic perspective. It would be quite interesting to see how it extends and extrapolates to more real-world domains. To this end, we would point the reader to the list of future works outlined below to extend it to more complex, but still synthetic, domains and also to Appendix B which discusses in detail ways to leverage insights from this work to perform experimentation in real-world settings as well as considerations that should be kept in mind when designing the same.

**Future Work**. We believe that this is a first step towards better benchmarking and understanding of modular systems. However, there are still a number of important and interesting directions that have not been explored in this work. We discuss some of these important future directions here.

*Stochasticity.* In Section 3, we see that we consider deterministic formulations of $p_y(\cdot \mid \mathbf{x}, \mathbf{c})$ and there is no labeling noise. One direction of exploration is to extend the settings considered here to noisy domains and investigate whether similar analyses still hold.

*Hard Attention.* In this work, we only considered simple soft-attention based activation decisions in end-to-end trained modular systems described in Section 4. An interesting future work involves benchmarking hard-attention based modular systems to explore whether they perform or specialize better, and also if the problem of collapse is exacerbated in such models.

*Finite-Data Regime.* Since this is the first work that provides such an analysis in this field, we decided to stick to the simplest setting of infinite-data regime to limit the effects of overfitting. We believe that a useful future work would be to consider a low-data regime to see whether the inductive biases of modularity and sparsity lead to even better generalization when there is only little data to learn from.

*Complex Rules.* While we consider the simplest setting for each *rule*, one could consider more complex distributions where $\mathbf{x}$ is also conditional on the rule $\mathbf{c}$ and where the labeling function, denoted by $\mathbf{y}$ in Section 3, is a complex non-linear function instead of a simple linear function. Analysis on trends between modular and monolithic models with increasing complexity of rules would not only lead to better understanding of such systems but also bridge it closer to real-world settings.

*Harder OoD Settings.* Modular systems are often shown to lead to better OoD generalization. In this work, we considered the simplest possible OoD settings that were often heavily correlated with in-distribution performances. Future work should investigate more complex OoD settings where either the support of $\mathbf{x}$ is dis-joint between in-distribution and OoD, or there are combinatorial computations required to obtain labels $\mathbf{y}$, such that certain rule permutations are with-held in training and used for evaluations.

*Better Inductive Biases.* A very important next step is to discover and investigate various inductive biases and regularization procedures that bridge the gap between the current modular systems and the perfectly specializing systems. Our benchmark provides the perfect opportunity that allows for analysis into the levels of specialization of different models.

The above points highlight only some of the immediate future works that would paint a richer and more intricate picture of what current modular systems are capable of, and what are the benefits we *can* obtain through perfect specialization.

Apart from extensions to the rule and module settings, we believe that it would also be important to investigate further into the quantitative evaluation metrics that we consider, in particular -

*Generalizing to non equi-probable rules.* Our current metrics on *collapse* and *specialization* rely on the need for equi-probable rules in the data generating distribution. It would be important to extend this to a more general setting where certain rules could be present more than the others.

*Different number of rules and modules.* For ease of quantitative evaluation, we only consider a well-specified system where the number of rules and the number of modules are kept the same. An

important next step is to formulate the *collapse* and *specialization* metrics to work in the settings where the number of modules could be more (or less) than the number of rules present in the data.

While investigating all the above possibilities is surely exciting, we believe that our work and setting provides the test-bed to allow for extensions and analysis into all the laid-out possibilities. This would not only allow for a more thorough understanding of modular systems but also lead to investigations into inductive biases that benefit such systems on various real-world settings.

## B  Impact, Extensions and Connections to Real-World Domains

While we do not perform explicit experimentation on real-world domains, we believe that the analysis done here would extrapolate to real-world settings and allow for better and more sample efficient large scale models. In this section, we highlight ways in which the analysis can either be extended to or impact real world settings as well as additional considerations for researchers to take into account when considering the different real-world extensions proposed here.

**Understanding Large MoE Models**. MoE based models have also been shown to be quite successful in large-scale domains Fedus et al. (2021); Shazeer et al. (2017); Lepikhin et al. (2020); Zuo et al. (2021); Wang et al. (2022). However, it is not clear whether they only offer ease of optimization or also benefits in performance through some notion of specialization. We believe it is an important research question to understand if their performance gains are linked to specialization, and if they are, how far are we from perfect specialization and how to reach there. To this end, we believe that our metrics can provide concrete quantitative assessment of the level of specialization obtained, and we believe that improving the capacity for specialization in our settings would extrapolate to more complex domains too. There are already some partial works that try to address the problems that we quantify; for example, the switch transformer uses a load-balancing term to prevent collapse. Certain works Zuo et al. (2021); Wang et al. (2022) also show that context dependent routing often doesn't provide additional benefits over random routing and one possible reason for this could be that context dependent routing is often severely sub-optimal in obtaining specialization as seen in our experiments.

In sum, consider our synthetic task setup along with evaluation metrics as an integrated tool for model architecture evaluation, rather than toy tasks. We think such architecture evaluation will be important to develop in the future, as networks start to exploit more modular structure. We consider our work as a systematic contribution toward this, helping to go beyond trial-and -error network design.

**Extension to Real World Settings**. The reason we consider synthetic settings is to have a very clear definition of specialization, that is, there is only one criteria which should drive specialization in our experiments and that is the rule context c. However, in more complex domains and multi-task settings, it is not so clear anymore. For example, between CIFAR10 and ImageNet classification, specialization could be at the level of dataset (CIFAR10 vs ImageNet) or at the level of object types (living vs non-living objects or ground vs water vs sky objects) or even at the lower level details (like presence of features like eyes, wings, wheels, etc.). Another example is Multilingual Language Modeling, where the notion of specialization could be tied to individual languages, or to different language families.

Even though the level of specialization is unclear in such complex domains, a possible way forward is taking a handful of notions of specialization and testing whether any of them leads to better performance in MoE models (eg. whether specialization at the level of language leads to better multilingual language modeling metrics in large MoE network styled as GT-Modular). That is, through GT-Modular and Modular-op styled MoE models, we can at least now test whether a designed notion of specialization is good for the task or not. Further, we can also extrapolate ways of improving specialization and reducing collapse from our synthetic domain to large-scale MoE systems which might lead not only to better performance but also more optimal sparse gating systems.

**Pre-training and Fine-tuning Extensions** . It is also possible to extend this analysis to test for transfer ability of models by constructing a set of tasks to pre-train on and then another set of tasks to perform fine-tuning on, with the hypothesis that a well-specialized system should learn better or faster during fine-tuning. However, we would like to point out that this is not a simple extension since it also requires a clear notion of consistency/similarity between different rules. One could assume that training on certain rules and testing on completely unrelated rules is not of as much importance,

and hence it requires a notion of similarity between tasks (eg. KL divergence between different mixture components as a notion of similarity; but it would require a move away from deterministic computations to noisy rules). Even after obtaining such a metric, it would provide another axis of study; i.e. how much similarity should be there between tasks for modularity to provide benefits. While an important question, we believe that it is a different research question from what we try to answer, which is the sub-optimality of modular systems in obtaining specialization.

**Synthetic Language Task Extensions**. Our setup can also be extended to testing of language models (LMs) by modeling the data distribution as some form of a mixture distribution in an underlying probabilistic context free grammar (pCFG) and analyzing whether current MoE systems specialize on the notion of experts in this setting.

**Usage in Statistical Modeling and Neuroscience**. Mixture distributions and Mixture of Experts based models have been widely used in Machine Learning and are applied in a number of real-world scenarios. They are often used to model statistical populations with subpopulations where each subpopulation could be modeled by a specific density and the mixture weights would reflect the proportion of each subpopulation. In this regard, we can look at our analysis at trying to determine in a general case of mixture distributions, how well can an MoE model discover the subpopulations, how can we evaluate it and whether it leads to any benefits in terms of performance.

In the recurrent domain, connections of the proposed data-distribution and modeling assumption can be made with switching linear dynamical systems (sLDS) which have been shown to be widely successful in modeling non-stationary interactions between high-dimensional neural populations Fox et al. (2008, 2010); Wulsin et al. (2013); Linderman et al. (2016); Glaser et al. (2020). Our recurrent-based data is reflective of the modeling assumptions in sLDS and our RNN models can be seen as an implementation of a flexible mixture-of-experts based system in this domain, however without incorporating the bayesian or stochastic perspective which is an important next step as outlined in Appendix A. Since such works rely on learning to discover low-dimensional structure in neurons through mixtures, we believe that our analysis would benefit this direction of research too by quantifying the extent to which an expert orients with a subpopulation.

# C   Proof of Model Relations

**To prove:** Given enough representational capacity, we need to show that *GT-Modular* $\subseteq$ *Modular-op* $\subseteq$ *Modular* $\subseteq$ *Monolithic*.

We prove this step-by-step by using the functional forms of different models as described in Table 1.

**Claim:** *GT-Modular* $\subseteq$ *Modular-op*

**Proof.** Given the formulation of a *GT-Modular* system as

$$\hat{\mathbf{y}}_m = f_m(\mathbf{x}, \mathbf{c}) \tag{21}$$

$$\hat{\mathbf{y}} = \sum_{m=1}^{R} c_m\,\hat{\mathbf{y}}_m \tag{22}$$

and of the *Modular-op* system as

$$\hat{\mathbf{y}}_m = f'_m(\mathbf{x}, \mathbf{c}) \tag{23}$$

$$\mathbf{p} = g(\mathbf{c}) \tag{24}$$

$$\hat{\mathbf{y}} = \sum_{m=1}^{R} p_m\,\hat{\mathbf{y}}_m \tag{25}$$

We see that we obtain *GT-Modular* from *Modular-op* simply by setting $g(\cdot)$ as the identity function and $f_m = f'_m$

**Claim:** *Modular-op* $\subseteq$ *Modular*

**Proof.** The formulation of a *Modular-op* system is given as

$$\hat{\mathbf{y}}_m = f_m(\mathbf{x}, \mathbf{c}) \tag{26}$$

$$\mathbf{p} = g(\mathbf{c}) \tag{27}$$

$$\hat{\mathbf{y}} = \sum_{m=1}^{R} p_m \, \hat{\mathbf{y}}_m \tag{28}$$

and that of the *Modular* system as

$$\hat{\mathbf{y}}_m, p_m = f'_m(\mathbf{x}, \mathbf{c}) \tag{29}$$

$$\hat{\mathbf{y}} = \sum_{m=1}^{R} p_m \, \hat{\mathbf{y}}_m \tag{30}$$

We can describe $f'_m(\mathbf{x}, \mathbf{c}) = \left( f'_{m_1}(\mathbf{x}, \mathbf{c}), f'_{m_2}(\mathbf{x}, \mathbf{c}) \right)$ for the *Modular* system. Setting $f'_{m_1}(\mathbf{x}, \mathbf{c}) = f_m(\mathbf{x}, \mathbf{c})$ and $f'_{m_2}(\mathbf{x}, \mathbf{c}) = g(\mathbf{c})$ for all $\mathbf{x}, \mathbf{c}$, we recover *Modular-op* from the *Modular* system.

**Claim:** *Modular* $\subseteq$ *Monolithic*

**Proof.** Let the formulation of a *Modular* system be

$$\hat{\mathbf{y}}_m, p_m = f_m(\mathbf{x}, \mathbf{c}) \tag{31}$$

$$\hat{\mathbf{y}} = \sum_{m=1}^{R} p_m \, \hat{\mathbf{y}}_m \tag{32}$$

where $f_m(\mathbf{x}, \mathbf{c}) = \left( f_{m_1}(\mathbf{x}, \mathbf{c}), f_{m_2}(\mathbf{x}, \mathbf{c}) \right)$. The formulation of the the *Monolithic* system is

$$\hat{\mathbf{y}} = f(\mathbf{x}, \mathbf{c}) \tag{33}$$

We can recover the *Modular* system from *Monolithic* simply by setting $f(\mathbf{x}, \mathbf{c}) = \sum_{m=1}^{R} f_{m_2}(\mathbf{x}, \mathbf{c}) \cdot f_{m_1}(\mathbf{x}, \mathbf{c})$.

Given enough capacity of all the systems, all the functional assignment are possible. This completes the proof, and shows that each choice provides an additional inductive bias by potentially restricting the functional class.

# D MLP

We provide detailed results of our MLP based experiments highlighting the effects of the training setting (Regression or Classification), the number of rules (ranging from 2 to 32) and the different model capacities. In these set of experiments, we use the MLP version of the data generating process (as highlighted in Section 3) and consider the models (highlighted in Section 4) with $f$ and $f_m$ modeled using MLP architectures.

**Task Setup.** We follow the task setup as described in Section 3. We consider 1-dimensional data samples for $\mathbf{x}_1$ and $\mathbf{x}_2$ and for the task parameters, we sample $\alpha_c, \beta_c \overset{\text{iid}}{\sim} \mathcal{N}(0, I)$. Further, for the OoD generalization setup, we instead sample input from a different distribution, i.e., $\mathbf{x}_1, \mathbf{x}_2 \overset{\text{iid}}{\sim} \mathcal{N}(0, 2\,I)$.

**Model Setup.** Input consists of numbers $x_1 \in \mathbb{R}$ and $x_2 \in \mathbb{R}$ as well as the rule context $c \in \{0, \dots, R\}$, with $R$ being the total number of rules. The model consists of two encoders $E_x$ and $E_c$ where $E_x$ maps $x_1$ and $x_2$ independently to $\mathbb{R}^d$ and $E_c$ maps $c$ to $\mathbb{R}^d$. Each of the encoders are implemented as non-linear neural networks with a single hidden layer. The encoded inputs are then concatenated together and fed to a model chosen from Monolithic, Modular, Modular-op and GT-Modular. The output of this model lies in $\mathbb{R}^d$ and is fed to a non-linear decoder with a single hidden layer to provide the final prediction $\hat{y}$.

*Monolithic*: This model consists of a non-linear single layered neural network that gets the concatenation of the three encodings as input and outputs a vector in $\mathbb{R}^d$.

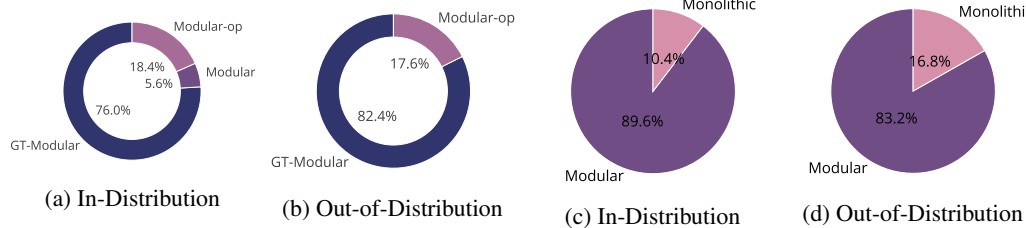

(a) In-Distribution    (b) Out-of-Distribution    (c) In-Distribution    (d) Out-of-Distribution

Figure 8: **Ranking Metric for MLP-Classification.** Each pie chart shows the number of times a model wins the competition (*higher is better*), which means outperforms the other models on a single task. Ranking is based on all models with in-distribution ranking in **(a)** and out-of-distribution ranking in **(b)**. On the contrary, rankings are based only on completely end-to-end trained Modular and Monolithic systems with in-distribution ranking in **(c)** and out-of-distribution ranking in **(d)**.

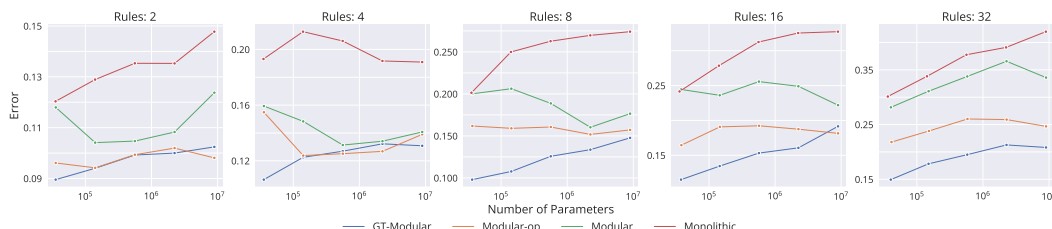

Figure 9: **In-Distribution Performance of MLP-Classification Models.** Performance (*lower is better*) of different models of varying capacities trained across different number of rules. Each point on the graph is obtained from an average over five tasks, each with five seeds, totaling 25 runs.

*Modular*: This model consists of R different non-linear single layered neural networks (modules), each of which gets the concatenation of the three encodings as input and outputs a corresponding activation score $p_m$ and prospective output $h_m$. The actual output of this modular system can be understood as $\sum_m p_m h_m$ which incorporates the output of each module in a soft manner as $p_m$ is obtained through a softmax. This output is then fed to a decoder, as in the other models.

*Modular-op*: This model is quite similar to the Modular system, with the only difference being that $p_m$ is not obtained from each module's computations but instead from a separate non-linear network with one hidden layer which gets the encoding of c as input and outputs a probability vector $p \in \Delta_R$, i.e. the activation probability of each module.

*GT-Modular*: This model is also quite similar to the Modular system, with the only difference being that $p_m = 1$ if m is equal to c, otherwise $p_m = 0$. Thus, there is a unique sparse one-to-one correspondence between rule context and module selection. It can also be thought of as a Modular-op model with the separate network being an identity mapping.

For our experiments, we ablate over the encoding dimension $d$ and the hidden layer size of the model over the set {(32, 128), (64,256), (128, 512), (256, 1024), (512, 2048)} and control for the number of parameters between the four different kinds of models considered.

**Training Details.** We train all the models for 100,000 iterations with a batch-size of 256 and the Adam optimizer with learning rate of 0.0001. For the classification tasks, we consider binary cross entropy loss, while for regression we consider the $l_1$ loss.

**Classification.** We first look at the results on the binary classification based MLP tasks. For reporting performance metrics, we consider all model capacities as well as number of rules while for collapse and specialization metrics, we consider the smallest, mid-size and largest models and report over the different number of rules.

*Performance.* For ease of readability, we first provide a snapshot of the results through rankings in Figure 8. The rankings are based on the votes obtained by the different models. Given a task, averaged over the five training seeds, a vote is given to the model that performs the best. This provides a quick view of the number of times each model outperformed the rest.

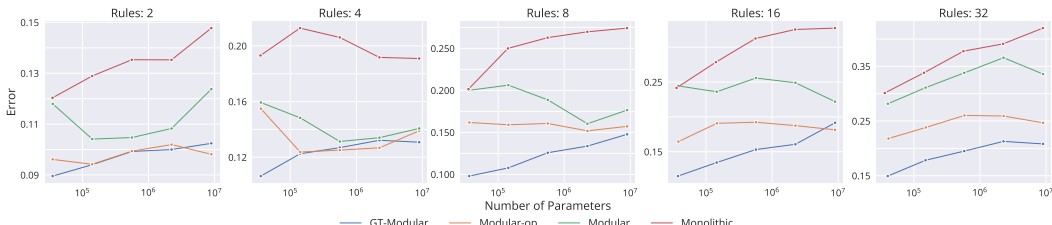

Figure 10: **Out-of-Distribution Performance of MLP-Classification Models.** Performance (*lower is better*) of different models of varying capacities trained across different number of rules. Each point on the graph is obtained from an average over five tasks, each with five seeds, totaling 25 runs.

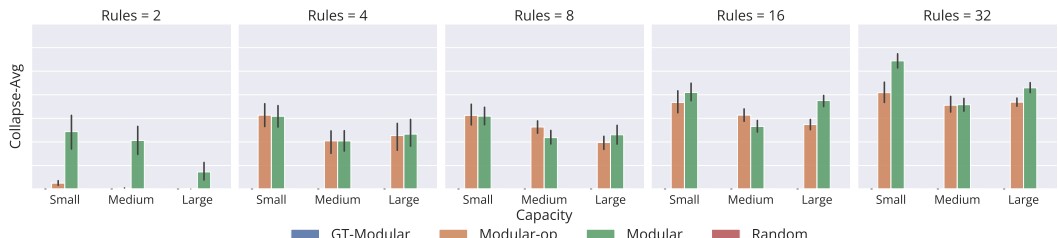

Figure 11: **Collapse-Avg Metric for MLP-Classification Models.** Highlights the amount of collapse (*lower is better*) suffered by different models of varying capacities trained across different number of rules. Each bar on the graph is obtained from an average over five tasks, each with five seeds, totaling 25 runs.

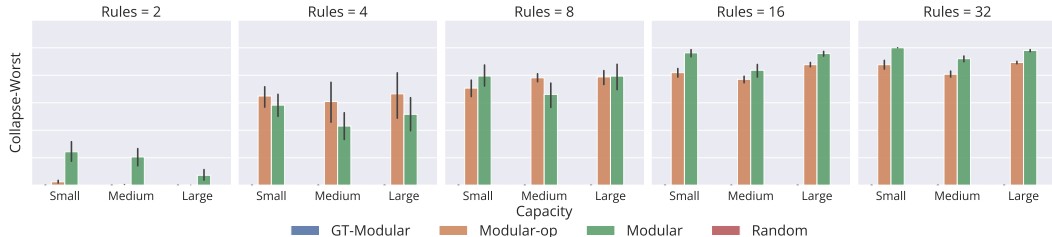

Figure 12: **Collapse-Worst Metric for MLP-Classification Models.** Highlights the amount of collapse (*lower is better*) suffered by different models of varying capacities trained across different number of rules. Each bar on the graph is obtained from an average over five tasks, each with five seeds, totaling 25 runs.

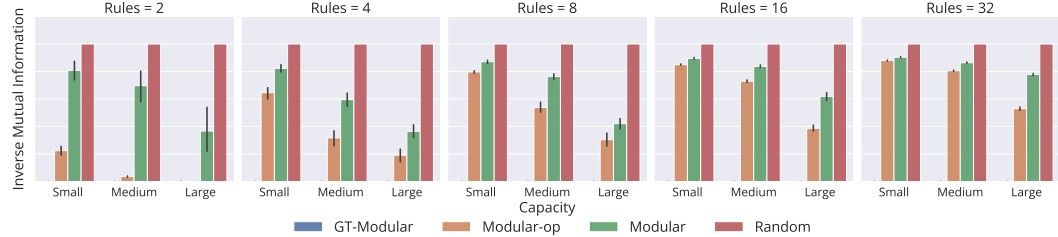

Figure 13: **Inverse Mutual Information Metric for MLP-Classification Models.** Highlights how poor the specialization (*lower is better*) is of different models of varying capacities trained across different number of rules. Each bar on the graph is obtained from an average over five tasks, each with five seeds, totaling 25 runs.

Next, we refer the readers to Figure 9 for the in-distribution and Figure 10 for the out-of-distribution performance of the various models across both different model capacities as well as different number of rules.

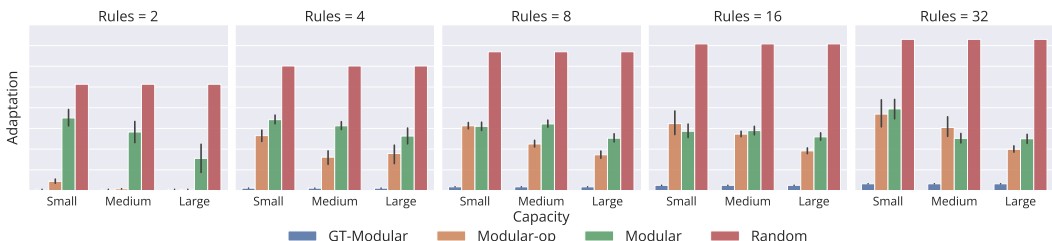

Figure 14: **Adaptation Metric for MLP-Classification Models.** Highlights how poor the specialization (*lower is better*) is of different models of varying capacities trained across different number of rules. Each bar on the graph is obtained from an average over five tasks, each with five seeds, totaling 25 runs.

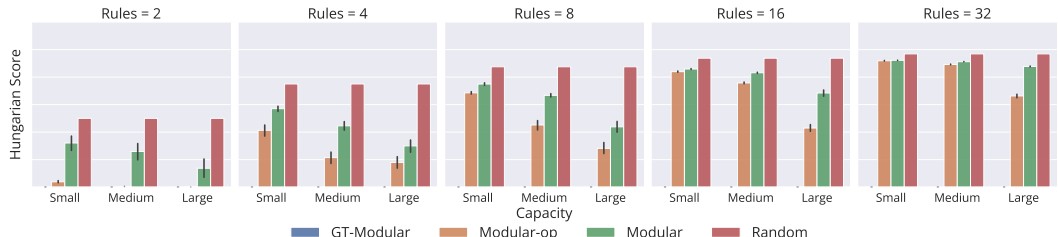

Figure 15: **Alignment Metric for MLP-Classification Models.** Highlights how poor the specialization (*lower is better*) is of different models of varying capacities trained across different number of rules. Each bar on the graph is obtained from an average over five tasks, each with five seeds, totaling 25 runs.

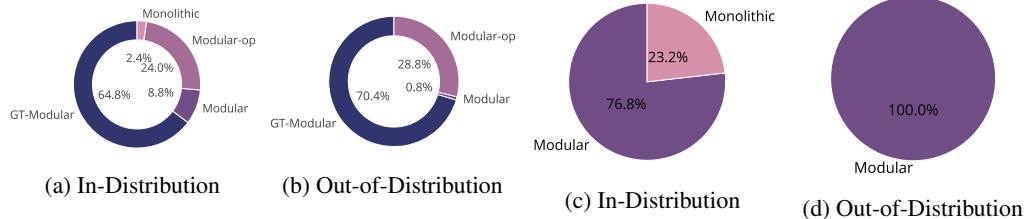

Figure 16: **Ranking Metric for MLP-Regression.** Each pie chart shows the number of times a model wins the competition (*higher is better*), which means outperforms the other models on a single task. Ranking is based on all models with in-distribution ranking in (**a**) and out-of-distribution ranking in (**b**). On the contrary, rankings are based only on completely end-to-end trained Modular and Monolithic systems with in-distribution ranking in (**c**) and out-of-distribution ranking in (**d**).

*Collapse-Avg.* For each rule setting, we report the *Collapse-Avg* metric score of the different models and the three different model capacities in Figure 11.

*Collapse-Worst.* For each rule setting, we report the *Collapse-Worst* metric score of the different models and the three different model capacities in Figure 12.

*Inverse Mutual Information.* For each rule setting, we report the *Inverse Mutual Information* metric score of the different models and the three different model capacities in Figure 13.

*Adaptation.* For each rule setting, we report the *Adaptation* metric score of the different models and the three different model capacities in Figure 14.

*Alignment.* For each rule setting, we report the *Alignment* metric score of the different models and the three different model capacities in Figure 15.

**Regression.** Next, we look at the results on the regression based MLP tasks. For reporting performance metrics, we consider all model capacities as well as number of rules while for collapse and

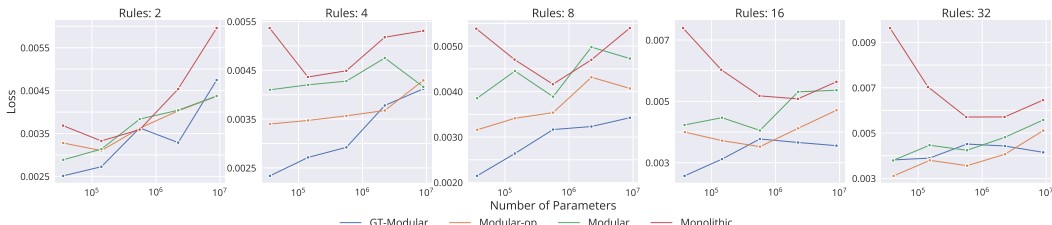

Figure 17: **In-Distribution Performance of MLP-Regression Models.** Performance (*lower is better*) of different models of varying capacities trained across different number of rules. Each point on the graph is obtained from an average over five tasks, each with five seeds, totaling 25 runs.

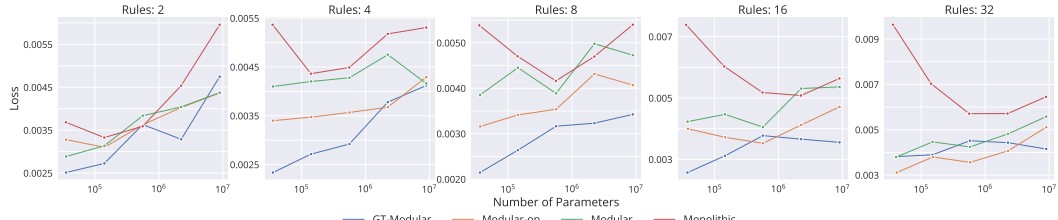

Figure 18: **Out-of-Distribution Performance of MLP-Regression Models.** Performance (*lower is better*) of different models of varying capacities trained across different number of rules. Each point on the graph is obtained from an average over five tasks, each with five seeds, totaling 25 runs.

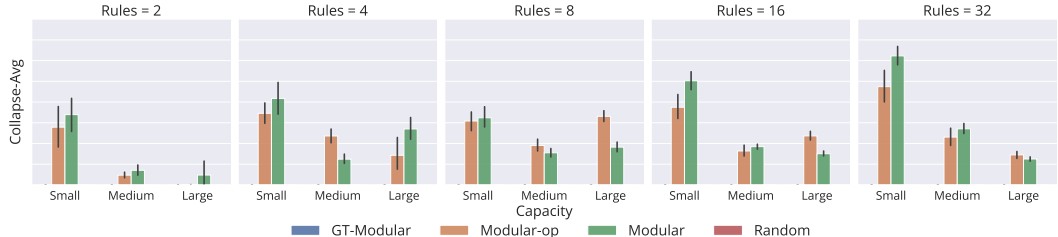

Figure 19: **Collapse-Avg Metric for MLP-Regression Models.** Highlights the amount of collapse (*lower is better*) suffered by different models of varying capacities trained across different number of rules. Each bar on the graph is obtained from an average over five tasks, each with five seeds, totaling 25 runs.

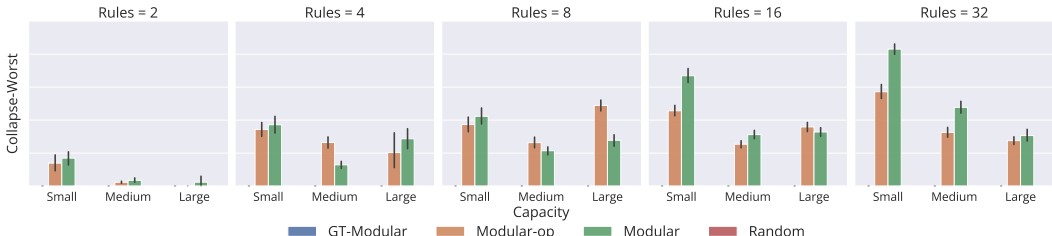

Figure 20: **Collapse-Worst Metric for MLP-Regression Models.** Highlights the amount of collapse (*lower is better*) suffered by different models of varying capacities trained across different number of rules. Each bar on the graph is obtained from an average over five tasks, each with five seeds, totaling 25 runs.

specialization metrics, we consider the smallest, mid-size and largest models and report over the different number of rules.

*Performance.* For ease of readability, we first provide a snapshot of the results through rankings in Figure 16. The rankings are based on the votes obtained by the different models. Given a task,

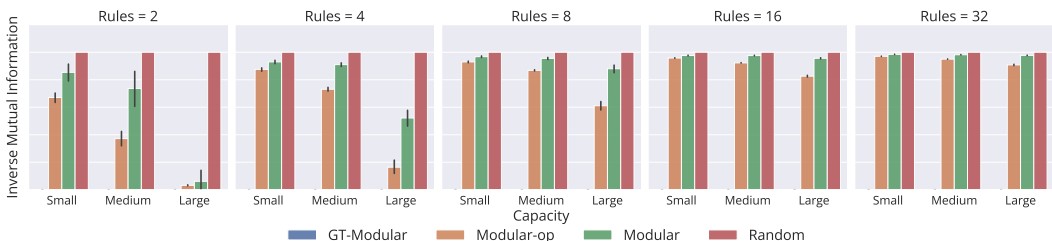

Figure 21: **Inverse Mutual Information Metric for MLP-Regression Models.** Highlights how poor the specialization (*lower is better*) is of different models of varying capacities trained across different number of rules. Each bar on the graph is obtained from an average over five tasks, each with five seeds, totaling 25 runs.

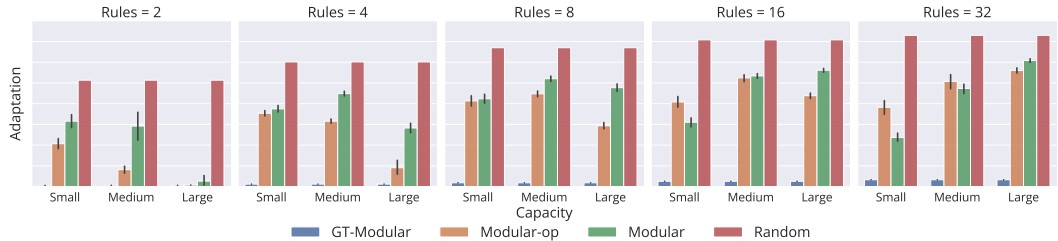

Figure 22: **Adaptation Metric for MLP-Regression Models.** Highlights how poor the specialization (*lower is better*) is of different models of varying capacities trained across different number of rules. Each bar on the graph is obtained from an average over five tasks, each with five seeds, totaling 25 runs.

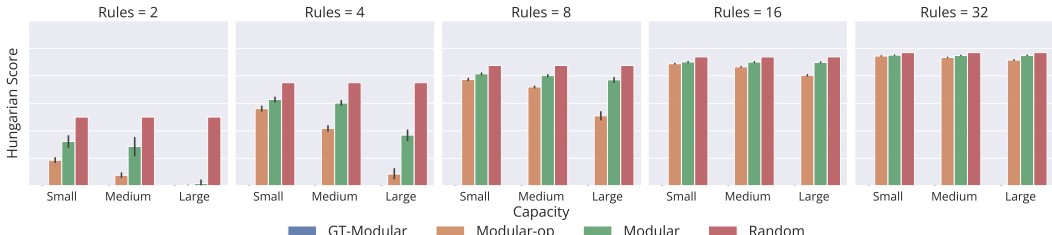

Figure 23: **Alignment Metric for MLP-Regression Models.** Highlights how poor the specialization (*lower is better*) is of different models of varying capacities trained across different number of rules. Each bar on the graph is obtained from an average over five tasks, each with five seeds, totaling 25 runs.

averaged over the five training seeds, a vote is given to the model that performs the best. This provides a quick view of the number of times each model outperformed the rest.

Next, we refer the readers to Figure 17 for the in-distribution and Figure 18 for the out-of-distribution performance of the various models across both different model capacities as well as different number of rules.

*Collapse-Avg.* For each rule setting, we report the *Collapse-Avg* metric score of the different models and the three different model capacities in Figure 19.

*Collapse-Worst.* For each rule setting, we report the *Collapse-Worst* metric score of the different models and the three different model capacities in Figure 20.

*Inverse Mutual Information.* For each rule setting, we report the *Inverse Mutual Information* metric score of the different models and the three different model capacities in Figure 21.

*Adaptation.* For each rule setting, we report the *Adaptation* metric score of the different models and the three different model capacities in Figure 22.

*Alignment.* For each rule setting, we report the *Alignment* metric score of the different models and the three different model capacities in Figure 23.

# E    MHA

We provide detailed results of our MHA based experiments highlighting the effects of the training setting (Regression or Classification), the number of rules (ranging from 2 to 32) and the different model capacities. In these set of experiments, we use the MHA version of the data generating process (as highlighted in Section 3) and consider the models (highlighted in Section 4) with $f$ and $f_m$ modeled using MHA architectures.

**Task Setup.** We follow the task setup as described in Section 3. We consider the sequence length as 10 for training and the input $\mathbf{v}_{nr}$ and $\mathbf{v}'_{nr}$ to be 1-dimensional per token per rule. We consider two different notions of search $d(\cdot, \cdot)$, outlined as

- Search-Version 1: Here $\mathbf{q}_{nr}$ and $\mathbf{q}'_{nr}$ are 1-dimensional per token per rule. The search rule is defined as $d(a, b) = |a - b|$.

- Search-Version 2: Here $\mathbf{q}_{nr}$ and $\mathbf{q}'_{nr}$ are 2-dimensional per token per rule and are instead sampled independently from a 2-dimensional hyper-sphere. The search rule is defined as $d(a, b) = a^T b$.

For the task parameters, we sample $\alpha_c, \beta_c \overset{\text{iid}}{\sim} \mathcal{N}(0, \mathbf{I})$. Further, for the OoD generalization setup, we instead sample input from a different distribution, i.e., we use $\mathcal{N}(0, 2\,\mathbf{I})$ instead of standard normal and in case of sampling from hyper-sphere, we instead consider one with double the radius. We also test with different sequence lengths, specifically 3, 5, 10, 20 and 30.

**Model Setup.** Input consists of a set of vectors $\{v_i\}_{i=1}^N$ where each vector $v_i$ is of dimensionality $4R$ for Search-Version 1 and $6R$ for Search-Version 2 as well as a set of rule contexts $\{c_i\}_{i=1}^N$ where $c_i \in \{0, \ldots, R\}$ with $R$ being the total number of rules. As in the MLP setup, we first use a single layered non-linear feed-forward network to independently encode each tuple $(v_i, c_i)$ to some latent space of dimension $d$. The encoded input then goes through a choice of model ranging from Monolithic, Modular, Modular-op and GT-Modular which gives an output in $\mathbb{R}^d$ which is then fed to a single-layered non-linear feed-forward decoder network to give the final prediction $\{\hat{y}_i\}_{i=1}^N$.

*Monolithic*: This model consists of a single Multi-Head Attention block with $2R$ heads that gets the encoded input set and outputs a corresponding set of vectors in $\mathbb{R}^d$. We keep the number of heads as $2R$ to allow for learning for all rules, as each rule requires 2 heads.

*Modular*: This model consists of $R$ different Multi-Head Attention blocks (modules) with 2 heads each, each of which gets the encoded input set and outputs a corresponding activation score $p_{i,m}$ and prospective output $h_{i,m}$ for the $i^{th}$ token. The actual output of this modular system at each token can be understood as $\sum_m p_{i,m} h_{i,m}$ which incorporates the output of each module in a soft manner as $p_{i,m}$ is obtained through a softmax. This output is then fed to a decoder, as in the other models.

*Modular-op*: This model is quite similar to the Modular system, with the only difference being that $p_{i,m}$ is not obtained from each module's computations but instead from a separate non-linear single-layered feed-forward network which gets $c_i$ as input and outputs a probability vector $p_i \in \Delta_R$ for each token $i$, i.e. the activation probability of each module for each token.

*GT-Modular*: This model is also quite similar to the Modular system, with the only difference being that $p_{i,m} = 1$ if $m$ is equal to $c_i$, otherwise $p_{i,m} = 0$. Thus, there is a unique sparse one-to-one correspondence between rule context and module selection. It can also be thought of as a Modular-op model with the separate network being an identity mapping.

For our experiments, we ablate over the encoding dimension $d$ and the hidden size which defines the heads dimensionality of the model over the set $\{(32, 128), (64,256), (128, 512), (256, 1024), (512, 2048)\}$ and control for the number of parameters between the four different kinds of models considered. This requires a non-trivial computation because in an MHA system, a monolithic one with attention layer size 256 has more parameters than a mixture of 4 MHA models with 64 size each, but has less parameters than a mixture of 4 MHA models with 256 size due to the final output MLP

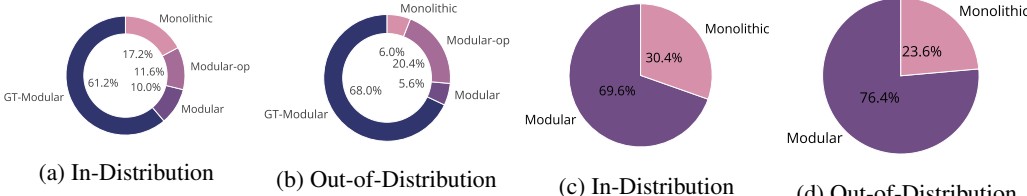

(a) In-Distribution          (b) Out-of-Distribution          (c) In-Distribution          (d) Out-of-Distribution

Figure 24: **Ranking Metric for MHA-Classification.** Each pie chart shows the number of times a model wins the competition (*higher is better*), which means outperforms the other models on a single task. Ranking is based on all models with in-distribution ranking in **(a)** and out-of-distribution ranking in **(b)**. On the contrary, rankings are based only on completely end-to-end trained Modular and Monolithic systems with in-distribution ranking in **(c)** and out-of-distribution ranking in **(d)**. For OoD, we only consider the extreme setting with largest sequence length and change in distribution of individual tokens.

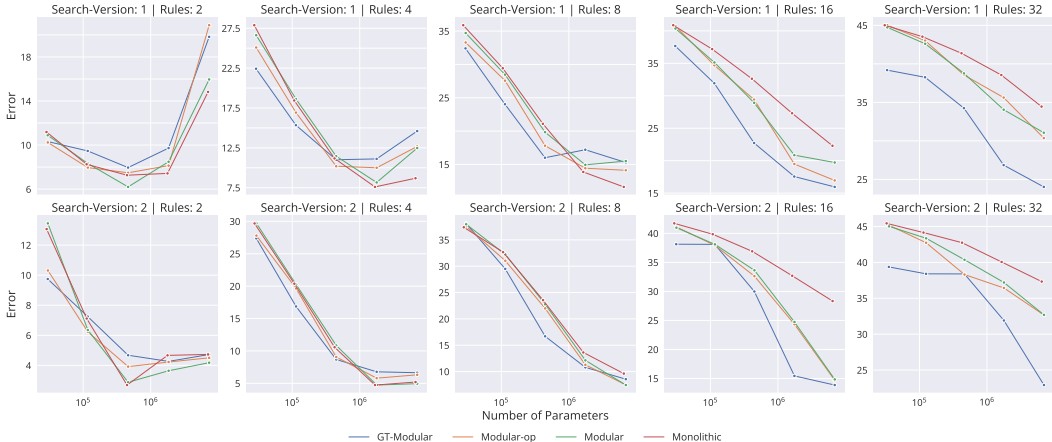

Figure 25: **In-Distribution Performance of MHA-Classification Models.** Performance (*lower is better*) of different models of varying capacities, different search versions in data and trained across different number of rules. Each point on the graph is obtained from an average over five tasks, each with five seeds, totaling 25 runs.

that combines multiple heads. The computations done in the code are to obtain an estimate of the layer size that maintains similar total number of parameters in the two systems.

**Training Details.** We train all the models for 500,000 iterations with a batch-size of 256 and the Adam optimizer with learning rate of 0.0001. For the classification tasks, we consider binary cross entropy loss, while for regression we consider the $l_1$ loss.

**Classification.** We first look at the results on the binary classification based MHA tasks. For reporting performance metrics, we consider all model capacities, all number of rules as well as the two search versions while for collapse and specialization metrics, we consider the smallest, mid-size and largest models and report over the different number of rules as well the search versions.

*Performance.* For ease of readability, we first provide a snapshot of the results through rankings in Figure 24. The rankings are based on the votes obtained by the different models. Given a task, averaged over the five training seeds, a vote is given to the model that performs the best. This provides a quick view of the number of times each model outperformed the rest.

We refer the readers to Figure 25 for the in-distribution performance and Figure 26 for the most extreme out-of-distribution performance (where we use the largest sequence length (30) and also change the distribution from which individual tokens are sampled) of the various models across both different model capacities, search functions as well as different number of rules. We also refer the reader to Figures 56 - 59 for the out-of-distribution case where only the sequence length is varied

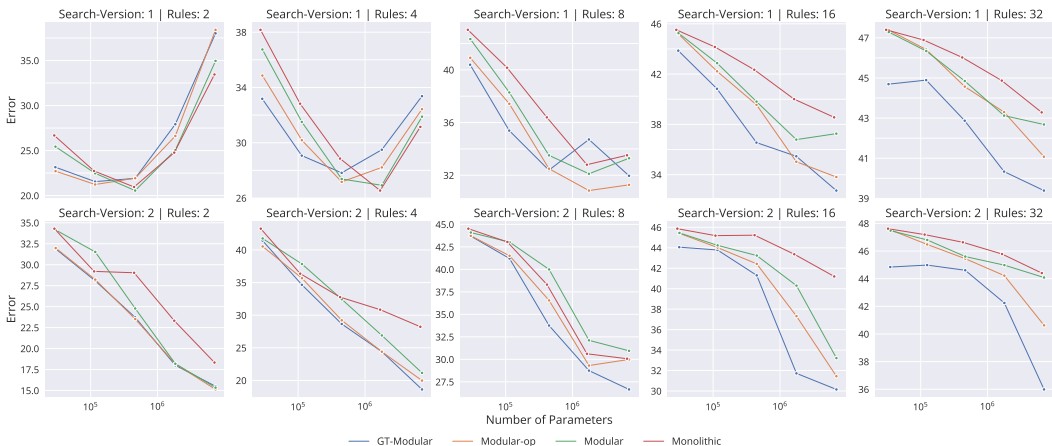

Figure 26: **Out-of-Distribution (Sequence Length: 30 - Individual Token Sampling: Altered) Performance on MHA-Classification Models.** Performance (*lower is better*) of different models of varying capacities, different search versions in data and trained across different number of rules. Each point on the graph is obtained from an average over five tasks, each with five seeds, totaling 25 runs.

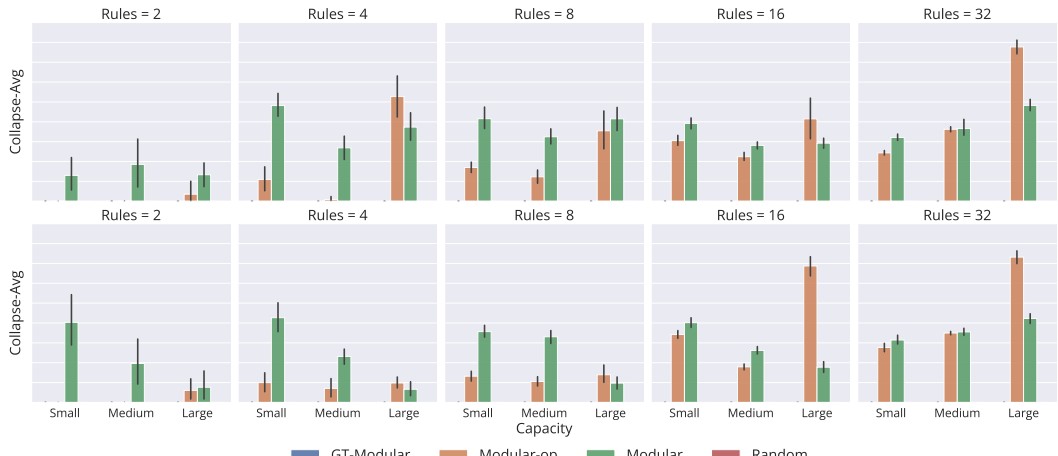

Figure 27: **Collapse-Avg Metric for MHA-Classification Models.** Highlights the amount of collapse (*lower is better*) suffered by different models of varying capacities trained across different number of rules. Each bar on the graph is obtained from an average over five tasks, each with five seeds, totaling 25 runs.

in the data and to Figures 60 - 64 where both the sequence length is potentially altered and also the distribution from which individual data points are sampled.

*Collapse-Avg.* For each rule setting and search version, we report the *Collapse-Avg* metric score of the different models and the three different model capacities in Figure 27.

*Collapse-Worst.* For each rule setting and search version, we report the *Collapse-Worst* metric score of the different models and the three different model capacities in Figure 28.

*Inverse Mutual Information.* For each rule setting and search version, we report the *Inverse Mutual Information* metric score of the different models and the three different model capacities in Figure 29.

*Adaptation.* For each rule setting and search version, we report the *Adaptation* metric score of the different models and the three different model capacities in Figure 30.

*Alignment.* For each rule setting and search version, we report the *Alignment* metric score of the different models and the three different model capacities in Figure 31.

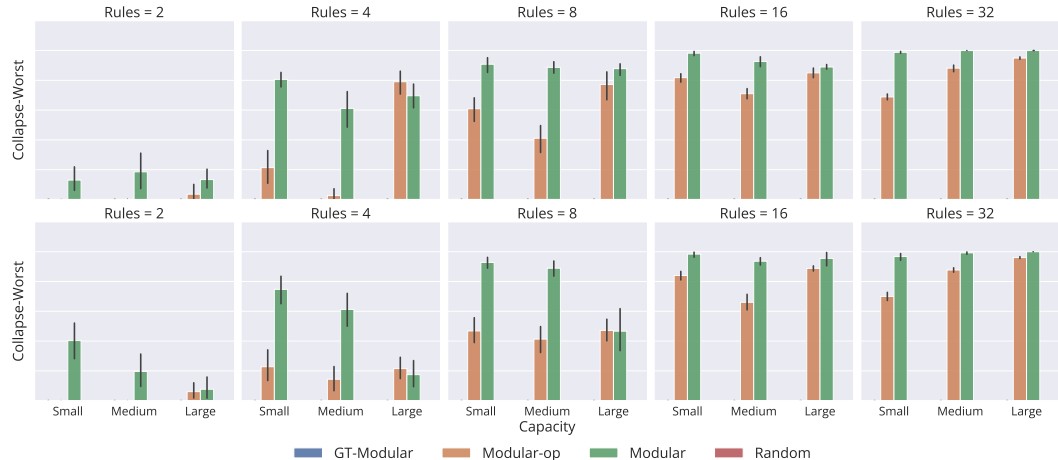

Figure 28: **Collapse-Worst Metric for MHA-Classification Models.** Highlights the amount of collapse (*lower is better*) suffered by different models of varying capacities trained across different number of rules. Each bar on the graph is obtained from an average over five tasks, each with five seeds, totaling 25 runs.

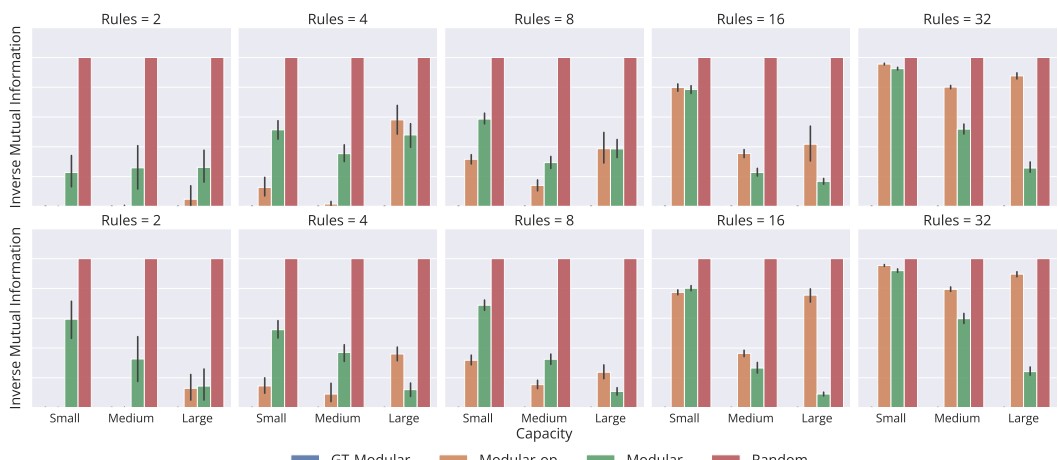

Figure 29: **Inverse Mutual Information Metric for MHA-Classification Models.** Highlights how poor the specialization (*lower is better*) is of different models of varying capacities trained across different number of rules. Each bar on the graph is obtained from an average over five tasks, each with five seeds, totaling 25 runs.

**Regression.** Next, we look at the results on the regression based MHA tasks. For reporting performance metrics, we consider all model capacities, all number of rules as well as the two search versions while for collapse and specialization metrics, we consider the smallest, mid-size and largest models and report over the different number of rules as well the search versions.

*Performance.* For ease of readability, we first provide a snapshot of the results through rankings in Figure 32. The rankings are based on the votes obtained by the different models. Given a task, averaged over the five training seeds, a vote is given to the model that performs the best. This provides a quick view of the number of times each model outperformed the rest.

We refer the readers to Figure 33 for the in-distribution performance and Figure 34 for the most extreme out-of-distribution performance (where we use the largest sequence length (30) and also change the distribution from which individual tokens are sampled) of the various models across both different model capacities, search functions as well as different number of rules. We also refer the reader to Figures 65 - 68 for the out-of-distribution case where only the sequence length is varied

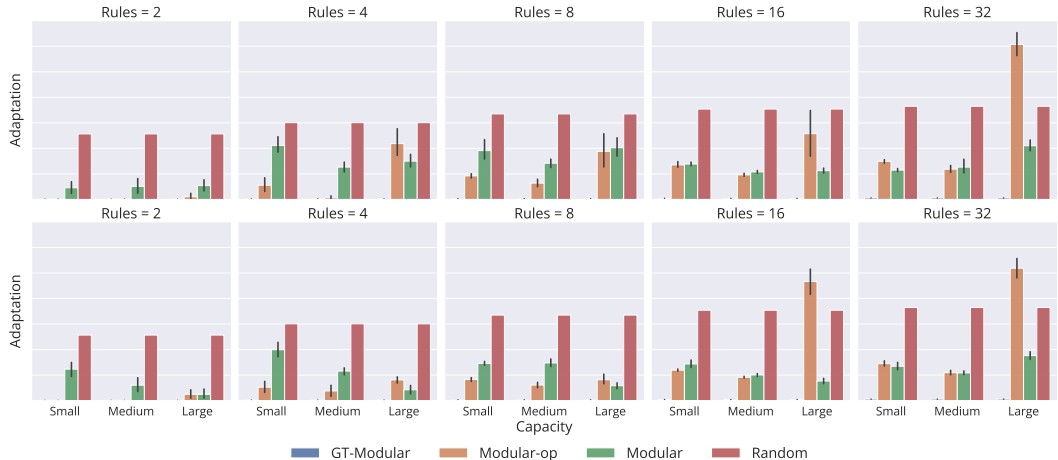

Figure 30: **Adaptation Metric for MHA-Classification Models.** Highlights how poor the specialization (*lower is better*) is of different models of varying capacities trained across different number of rules. Each bar on the graph is obtained from an average over five tasks, each with five seeds, totaling 25 runs.

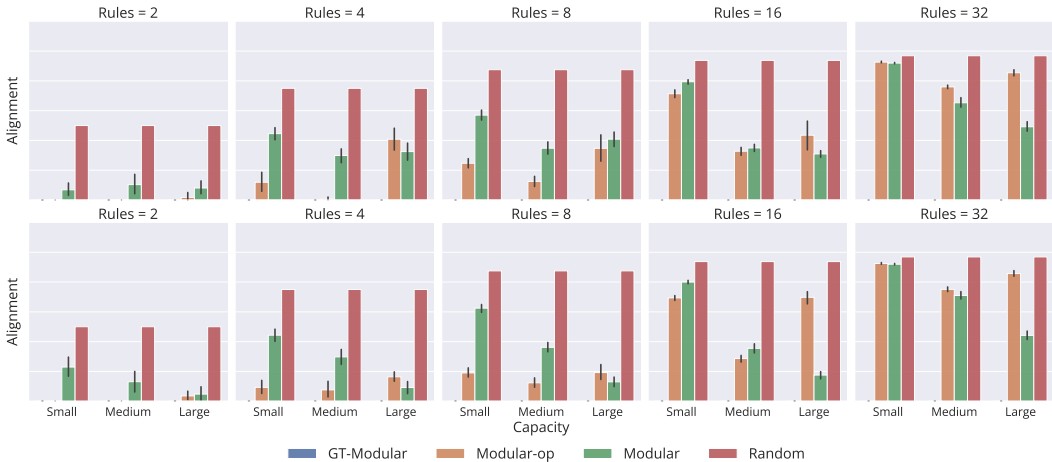

Figure 31: **Alignment Metric for MHA-Classification Models.** Highlights how poor the specialization (*lower is better*) is of different models of varying capacities trained across different number of rules. Each bar on the graph is obtained from an average over five tasks, each with five seeds, totaling 25 runs.

in the data and to Figures 69 - 73 where both the sequence length is potentially altered and also the distribution from which individual data points are sampled.

*Collapse-Avg.* For each rule setting and search version, we report the *Collapse-Avg* metric score of the different models and the three different model capacities in Figure 35.

*Collapse-Worst.* For each rule setting and search version, we report the *Collapse-Worst* metric score of the different models and the three different model capacities in Figure 36.

*Inverse Mutual Information.* For each rule setting and search version, we report the *Inverse Mutual Information* metric score of the different models and the three different model capacities in Figure 37.

*Adaptation.* For each rule setting and search version, we report the *Adaptation* metric score of the different models and the three different model capacities in Figure 38.

*Alignment.* For each rule setting and search version, we report the *Alignment* metric score of the different models and the three different model capacities in Figure 39.

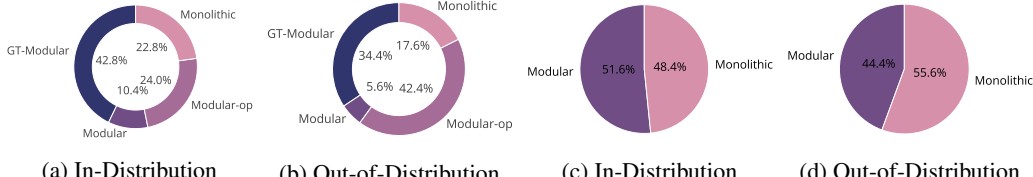

(a) In-Distribution      (b) Out-of-Distribution      (c) In-Distribution      (d) Out-of-Distribution

Figure 32: **Ranking Metric for MHA-Regression.** Each pie chart shows the number of times a model wins the competition (*higher is better*), which means outperforms the other models on a single task. Ranking is based on all models with in-distribution ranking in **(a)** and out-of-distribution ranking in **(b)**. On the contrary, rankings are based only on completely end-to-end trained Modular and Monolithic systems with in-distribution ranking in **(c)** and out-of-distribution ranking in **(d)**. For OoD, we only consider the extreme setting with largest sequence length and change in distribution of individual tokens.

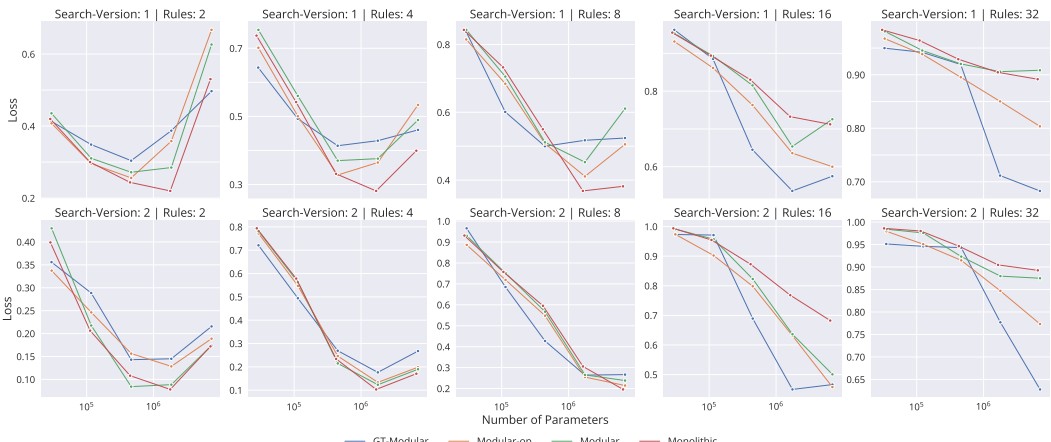

Figure 33: **In-Distribution Performance of MHA-Regression Models.** Performance (*lower is better*) of different models of varying capacities, different search versions in data and trained across different number of rules. Each point on the graph is obtained from an average over five tasks, each with five seeds, totaling 25 runs.

# F RNN

We provide detailed results of our RNN based experiments highlighting the effects of the training setting (Regression or Classification), the number of rules (ranging from 2 to 32) and the different model capacities. In these set of experiments, we use the RNN version of the data generating process (as highlighted in Section 3) and consider the models (highlighted in Section 4) with $f$ and $f_m$ modeled using simple RNN architectures.

**Task Setup.** We follow the task setup as described in Section 3. We consider the sequence length as 10 for training and the input $\mathbf{x}_n$ to be 32-dimensional.

For the task parameters, we sample $A_c, B_c \overset{\text{iid}}{\sim} \mathcal{N}(0, \frac{1}{\sqrt{32}}\mathbf{I})$ and $w \sim \mathcal{N}(0, \mathbf{I})$. Further, for the OoD generalization setup, we instead sample input from a different distribution, i.e., we use $\mathcal{N}(0, 2\,\mathbf{I})$ instead of standard normal. We also test with different sequence lengths, specifically 3, 5, 10, 20 and 30.

**Model Setup.** Input consists of a sequence of vectors $\{v_i\}_{i=1}^N$ where each vector $v_i$ is of dimensionality 32, as well as a set of rule contexts $\{c_i\}_{i=1}^N$ where $c_i \in \{0, \ldots, R\}$ with $R$ being the total number of rules. As in the MLP setup, we first use a single layered non-linear feed-forward network to independently encode each tuple $(v_i, c_i)$ to some latent space of dimension d. The encoded input then goes through a choice of model ranging from Monolithic, Modular, Modular-op and GT-Modular

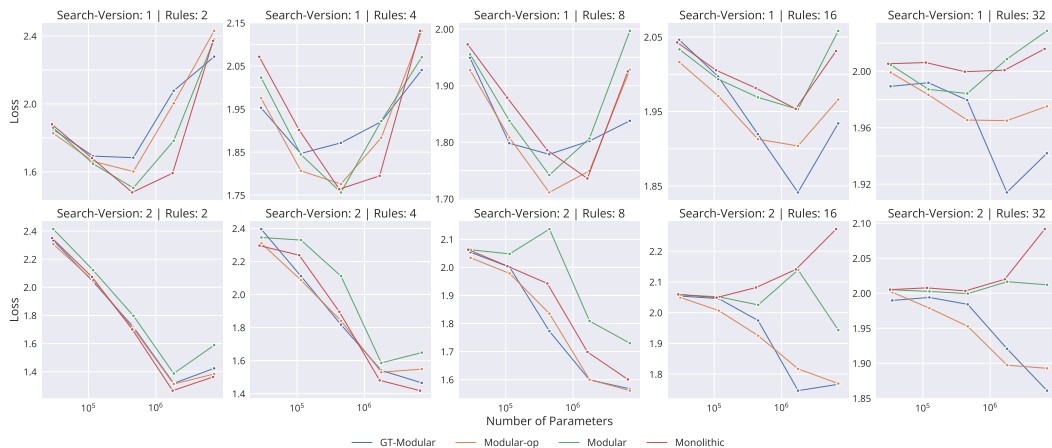

Figure 34: **Out-of-Distribution (Sequence Length: 30 - Individual Token Sampling: Altered) Performance on MHA-Regression Models.** Performance (*lower is better*) of different models of varying capacities, different search versions in data and trained across different number of rules. Each point on the graph is obtained from an average over five tasks, each with five seeds, totaling 25 runs.

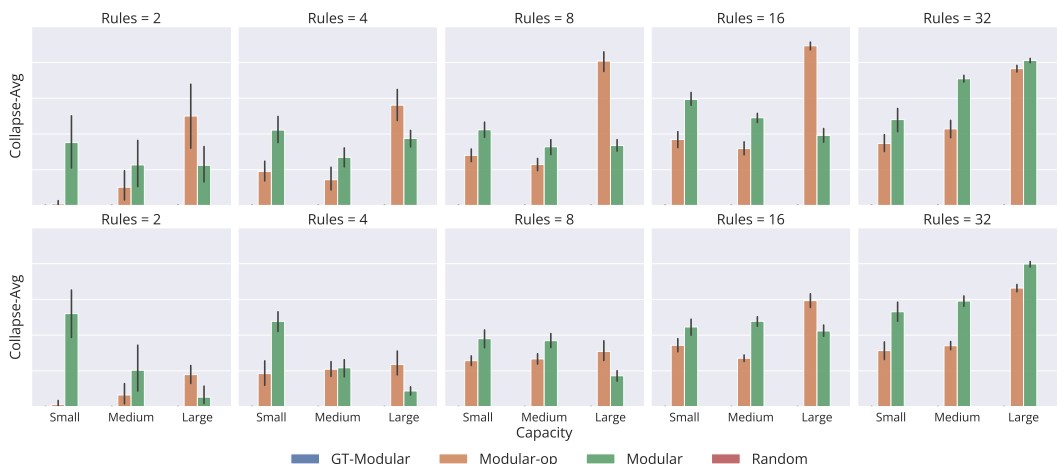

Figure 35: **Collapse-Avg Metric for MHA-Regression Models.** Highlights the amount of collapse (*lower is better*) suffered by different models of varying capacities trained across different number of rules. Each bar on the graph is obtained from an average over five tasks, each with five seeds, totaling 25 runs.

which gives an output in $\mathbb{R}^d$ which is then fed to a single-layered non-linear feed-forward decoder network to give the final prediction $\{\hat{y}_i\}_{i=1}^N$.

*Monolithic*: This model consists of a single LSTM Cell that gets the encoded sequence as input and outputs a corresponding sequence of vectors in $\mathbb{R}^d$.

*Modular*: This model consists of R different LSTM Cells (modules) each of which gets the encoded sequence as input and outputs a corresponding activation score $p_{i,m}$ and prospective output $h_{i,m}$ for each token. The actual output of this modular system at each token can be understood as $\sum_m p_{i,m} h_{i,m}$ which incorporates the output of each module in a soft manner as $p_{i,m}$ is obtained through a softmax. This output is then fed to a decoder, as in the other models.

*Modular-op*: This model is quite similar to the Modular system, with the only difference being that $p_{i,m}$ is not obtained from each module's computations but instead from a separate non-linear single-layered feed-forward network which gets $c_i$ as input and outputs a probability vector $p_i \in \Delta_R$ for each token $i$, i.e. the activation probability of each module for each token.

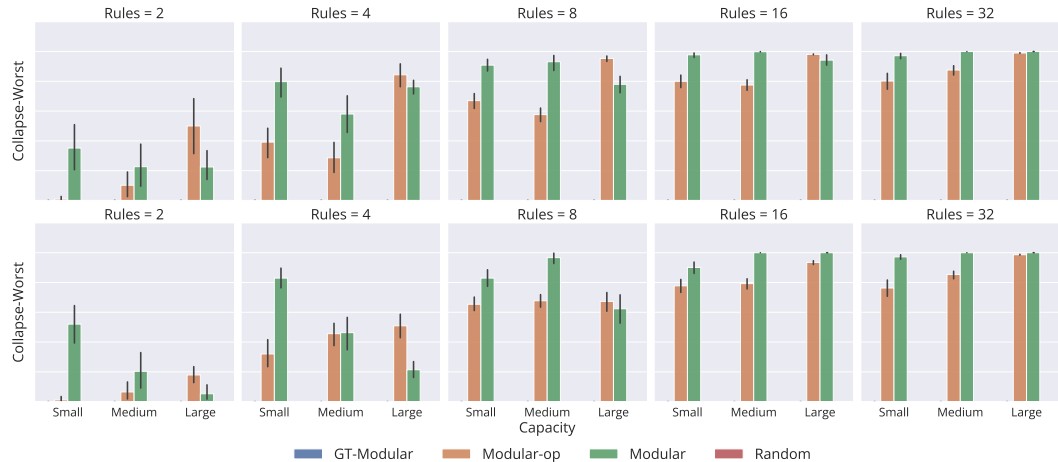

Figure 36: **Collapse-Worst Metric for MHA-Regression Models.** Highlights the amount of collapse (*lower is better*) suffered by different models of varying capacities trained across different number of rules. Each bar on the graph is obtained from an average over five tasks, each with five seeds, totaling 25 runs.

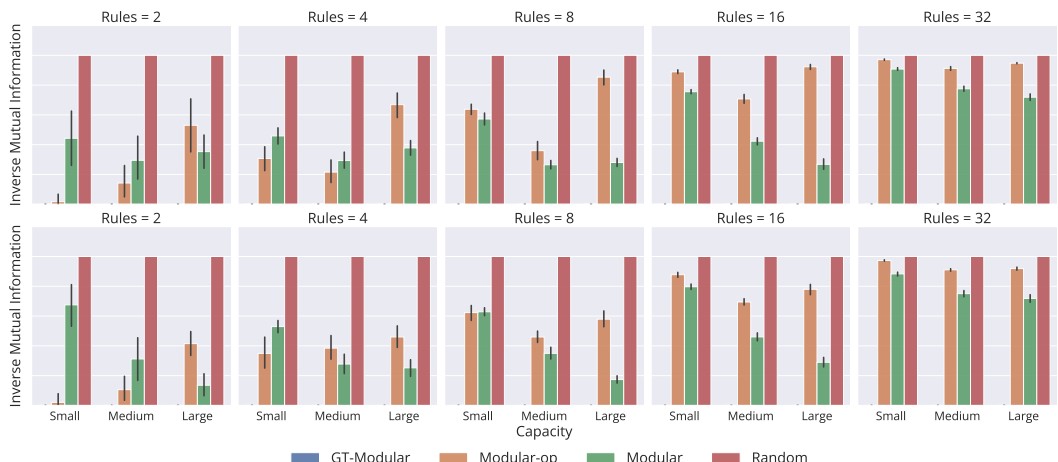

Figure 37: **Inverse Mutual Information Metric for MHA-Regression Models.** Highlights how poor the specialization (*lower is better*) is of different models of varying capacities trained across different number of rules. Each bar on the graph is obtained from an average over five tasks, each with five seeds, totaling 25 runs.

*GT-Modular*: This model is also quite similar to the Modular system, with the only difference being that $p_{i,m} = 1$ if $m$ is equal to $c_i$, otherwise $p_{i,m} = 0$. Thus, there is a unique sparse one-to-one correspondence between rule context and module selection. It can also be thought of as a Modular-op model with the separate network being an identity mapping.

For our experiments, we ablate over the encoding dimension $d$ and the dimensionality that controls the hidden size of the RNN over the set {(32, 128), (64,256), (128, 512), (256, 1024), (512, 2048)} and control for the number of parameters between the four different kinds of models considered. This requires a non-trivial computation because in a recurrent system, a monolithic RNN with hidden size 256 has more parameters than a mixture of 4 RNN models with 64 hidden size each, but has less parameters than a mixture of 4 RNN models with 256 hidden size. The computations done in the code are to obtain an estimate of the hidden size that maintains that the total number of parameters in the two systems are similar.

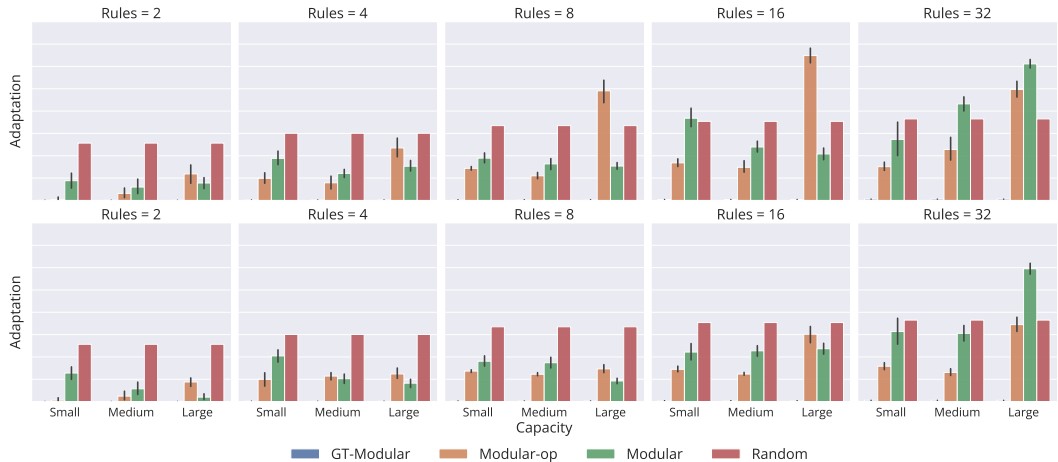

Figure 38: **Adaptation Metric for MHA-Regression Models.** Highlights how poor the specialization (*lower is better*) is of different models of varying capacities trained across different number of rules. Each bar on the graph is obtained from an average over five tasks, each with five seeds, totaling 25 runs.

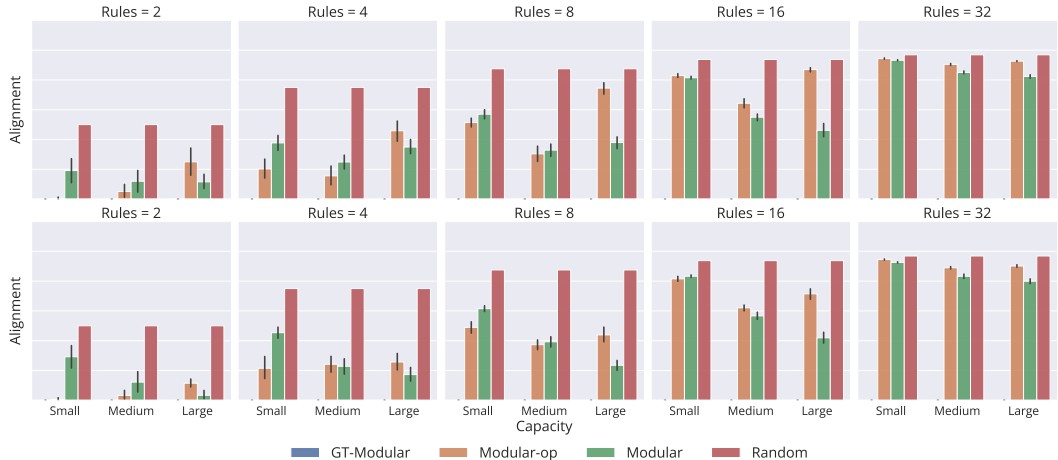

Figure 39: **Alignment Metric for MHA-Classification Models.** Highlights how poor the specialization (*lower is better*) is of different models of varying capacities trained across different number of rules. Each bar on the graph is obtained from an average over five tasks, each with five seeds, totaling 25 runs.

**Training Details.** We train all the models for 500,000 iterations with a batch-size of 256 and the Adam optimizer with learning rate of 0.0001, with gradient clipping at 1.0. For the classification tasks, we consider binary cross entropy loss, while for regression we consider the $l_1$ loss.

**Classification.** We first look at the results on the binary classification based RNN tasks. For reporting performance metrics, we consider all model capacities and all number of rules while for collapse and specialization metrics, we consider the smallest, mid-size and largest models and report over the different number of rules.

*Performance.* For ease of readability, we first provide a snapshot of the results through rankings in Figure 40. The rankings are based on the votes obtained by the different models. Given a task, averaged over the five training seeds, a vote is given to the model that performs the best. This provides a quick view of the number of times each model outperformed the rest.

We refer the readers to Figure 41 for the in-distribution performance and Figure 42 for the most extreme out-of-distribution performance (where we use the largest sequence length (30) and also

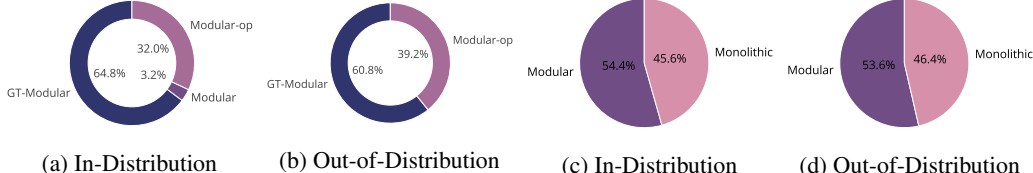

(a) In-Distribution      (b) Out-of-Distribution      (c) In-Distribution      (d) Out-of-Distribution

Figure 40: **Ranking Metric for RNN-Classification.** Each pie chart shows the number of times a model wins the competition (*higher is better*), which means outperforms the other models on a single task. Ranking is based on all models with in-distribution ranking in **(a)** and out-of-distribution ranking in **(b)**. On the contrary, rankings are based only on completely end-to-end trained Modular and Monolithic systems with in-distribution ranking in **(c)** and out-of-distribution ranking in **(d)**. For OoD, we only consider the extreme setting with largest sequence length and change in distribution of individual tokens.

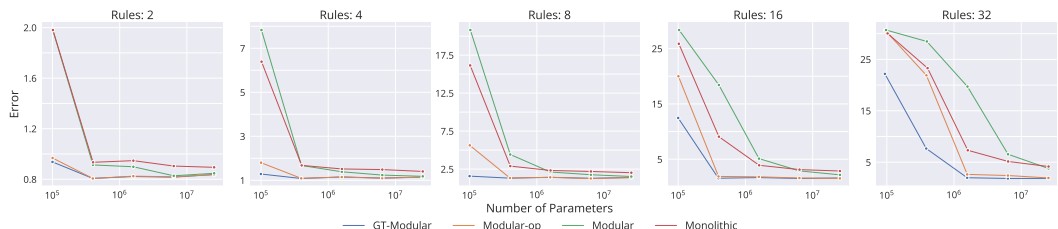

Figure 41: **In-Distribution Performance of RNN-Classification Models.** Performance (*lower is better*) of different models of varying capacities and trained across different number of rules. Each point on the graph is obtained from an average over five tasks, each with five seeds, totaling 25 runs.

change the distribution from which individual tokens are sampled) of the various models across both different model capacities as well as different number of rules. We also refer the reader to Figures 74 - 77 for the out-of-distribution case where only the sequence length is varied in the data and to Figures 78 - 82 where both the sequence length is potentially altered and also the distribution from which individual data points are sampled.

*Collapse-Avg.* For each rule setting, we report the *Collapse-Avg* metric score of the different models and the three different model capacities in Figure 43.

*Collapse-Worst.* For each rule setting, we report the *Collapse-Worst* metric score of the different models and the three different model capacities in Figure 44.

*Inverse Mutual Information.* For each rule setting, we report the *Inverse Mutual Information* metric score of the different models and the three different model capacities in Figure 45.

*Adaptation.* For each rule setting, we report the *Adaptation* metric score of the different models and the three different model capacities in Figure 46.

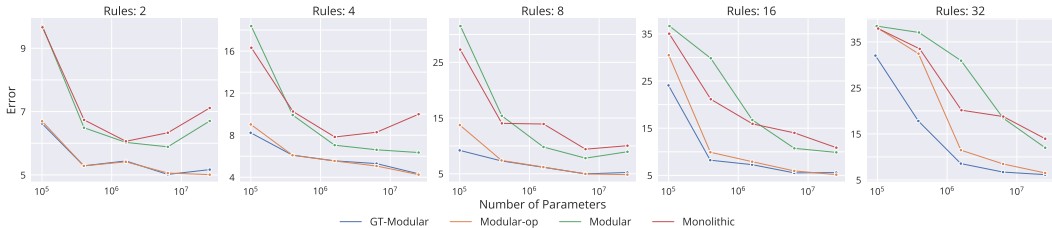

Figure 42: **Out-of-Distribution (Sequence Length: 30 - Individual Token Sampling: Altered) Performance on RNN-Classification Models.** Performance (*lower is better*) of different models of varying capacities and trained across different number of rules. Each point on the graph is obtained from an average over five tasks, each with five seeds, totaling 25 runs.

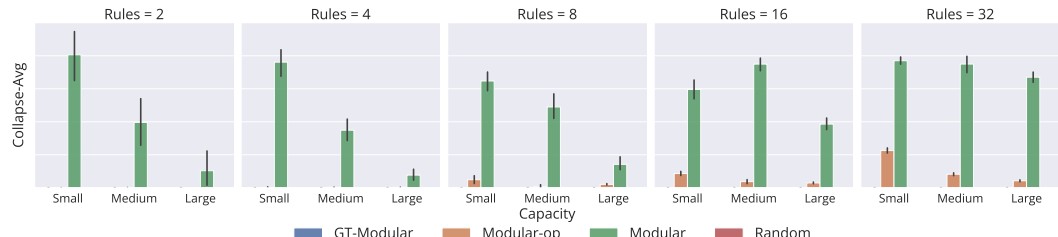

Figure 43: **Collapse-Avg Metric for RNN-Classification Models.** Highlights the amount of collapse (*lower is better*) suffered by different models of varying capacities trained across different number of rules. Each bar on the graph is obtained from an average over five tasks, each with five seeds, totaling 25 runs.

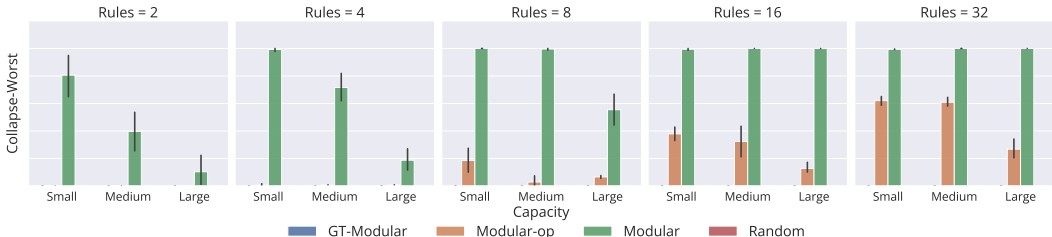

Figure 44: **Collapse-Worst Metric for RNN-Classification Models.** Highlights the amount of collapse (*lower is better*) suffered by different models of varying capacities trained across different number of rules. Each bar on the graph is obtained from an average over five tasks, each with five seeds, totaling 25 runs.

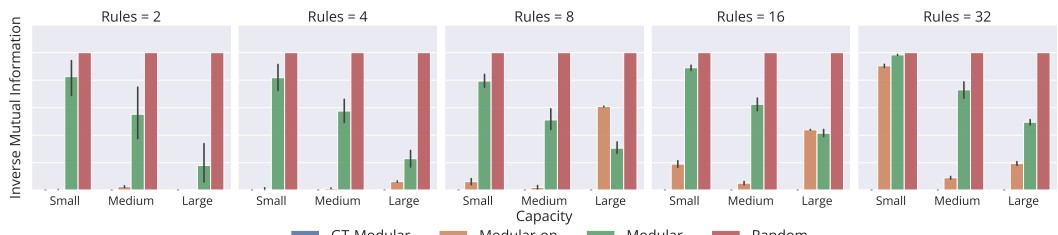

Figure 45: **Inverse Mutual Information Metric for RNN-Classification Models.** Highlights how poor the specialization (*lower is better*) is of different models of varying capacities trained across different number of rules. Each bar on the graph is obtained from an average over five tasks, each with five seeds, totaling 25 runs.

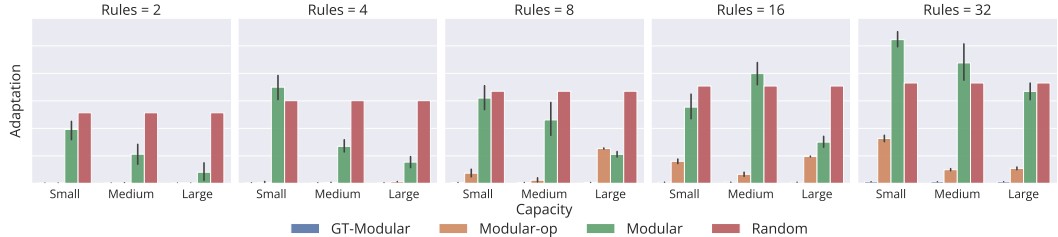

Figure 46: **Adaptation Metric for RNN-Classification Models.** Highlights how poor the specialization (*lower is better*) is of different models of varying capacities trained across different number of rules. Each bar on the graph is obtained from an average over five tasks, each with five seeds, totaling 25 runs.

*Alignment.* For each rule setting, we report the *Alignment* metric score of the different models and the three different model capacities in Figure 47.

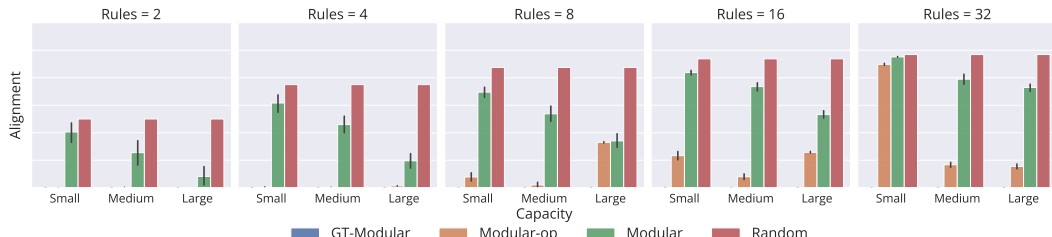

Figure 47: **Alignment Metric for RNN-Classification Models.** Highlights how poor the specialization (*lower is better*) is of different models of varying capacities trained across different number of rules. Each bar on the graph is obtained from an average over five tasks, each with five seeds, totaling 25 runs.

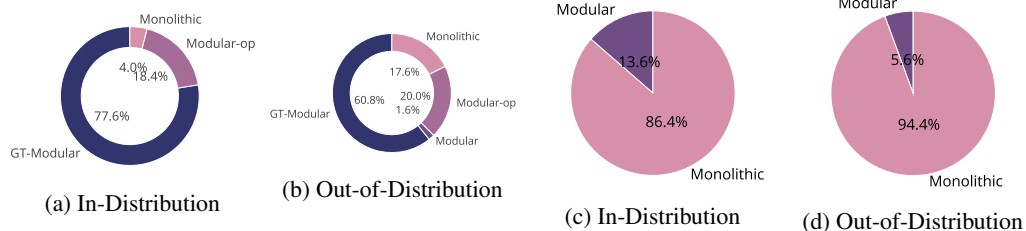

Figure 48: **Ranking Metric for RNN-Regression.** Each pie chart shows the number of times a model wins the competition (*higher is better*), which means outperforms the other models on a single task. Ranking is based on all models with in-distribution ranking in **(a)** and out-of-distribution ranking in **(b)**. On the contrary, rankings are based only on completely end-to-end trained Modular and Monolithic systems with in-distribution ranking in **(c)** and out-of-distribution ranking in **(d)**. For OoD, we only consider the extreme setting with largest sequence length and change in distribution of individual tokens.

**Regression.** Next, we look at the results on the regression based RNN tasks. For reporting performance metrics, we consider all model capacities and all number of rules while for collapse and specialization metrics, we consider the smallest, mid-size and largest models and report over the different number of rules.

*Performance.* For ease of readability, we first provide a snapshot of the results through rankings in Figure 48. The rankings are based on the votes obtained by the different models. Given a task, averaged over the five training seeds, a vote is given to the model that performs the best. This provides a quick view of the number of times each model outperformed the rest.

We refer the readers to Figure 49 for the in-distribution performance and Figure 50 for the most extreme out-of-distribution performance (where we use the largest sequence length (30) and also change the distribution from which individual tokens are sampled) of the various models across both different model capacities, search functions as well as different number of rules. We also refer the

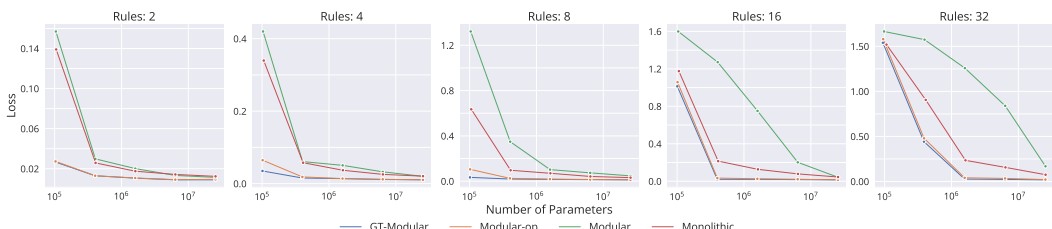

Figure 49: **In-Distribution Performance of RNN-Regression Models.** Performance (*lower is better*) of different models of varying capacities and trained across different number of rules. Each point on the graph is obtained from an average over five tasks, each with five seeds, totaling 25 runs.

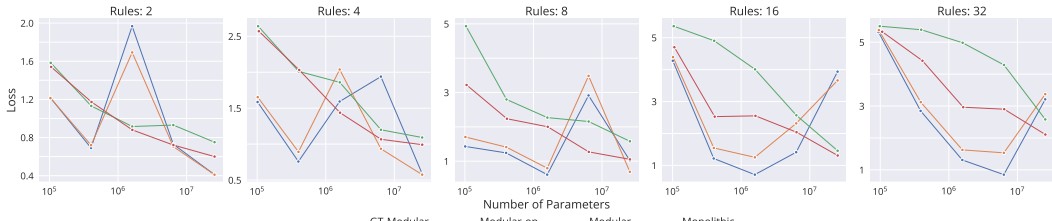

Figure 50: **Out-of-Distribution (Sequence Length: 30 - Individual Token Sampling: Altered) Performance on RNN-Regression Models.** Performance (*lower is better*) of different models of varying capacities and trained across different number of rules. Each point on the graph is obtained from an average over five tasks, each with five seeds, totaling 25 runs.

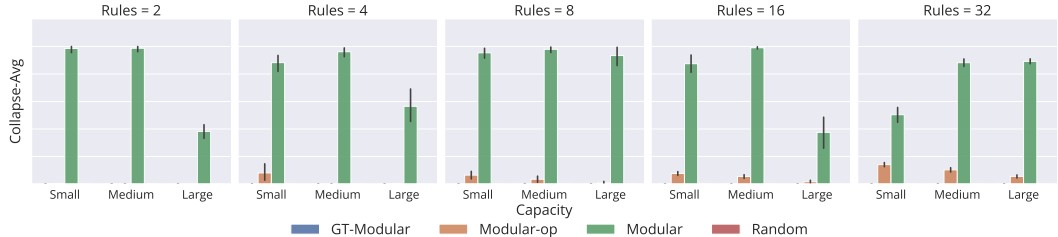

Figure 51: **Collapse-Avg Metric for RNN-Regression Models.** Highlights the amount of collapse (*lower is better*) suffered by different models of varying capacities trained across different number of rules. Each bar on the graph is obtained from an average over five tasks, each with five seeds, totaling 25 runs.

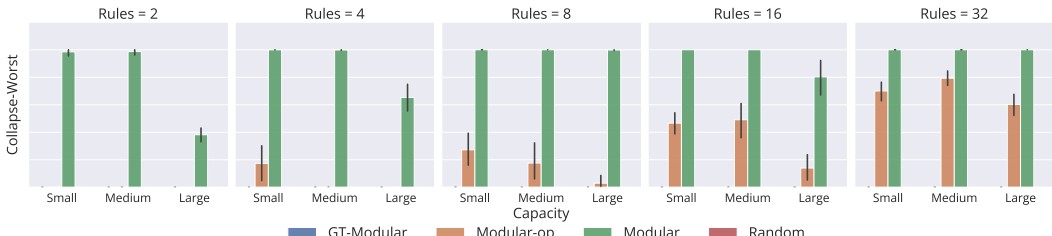

Figure 52: **Collapse-Worst Metric for RNN-Regression Models.** Highlights the amount of collapse (*lower is better*) suffered by different models of varying capacities trained across different number of rules. Each bar on the graph is obtained from an average over five tasks, each with five seeds, totaling 25 runs.

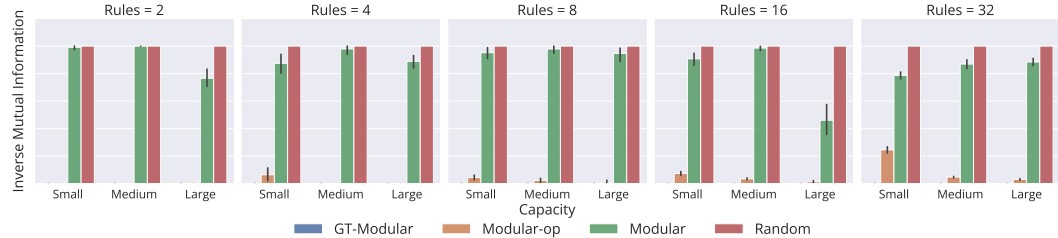

Figure 53: **Inverse Mutual Information Metric for RNN-Regression Models.** Highlights how poor the specialization (*lower is better*) is of different models of varying capacities trained across different number of rules. Each bar on the graph is obtained from an average over five tasks, each with five seeds, totaling 25 runs.

reader to Figures 83 - 86 for the out-of-distribution case where only the sequence length is varied

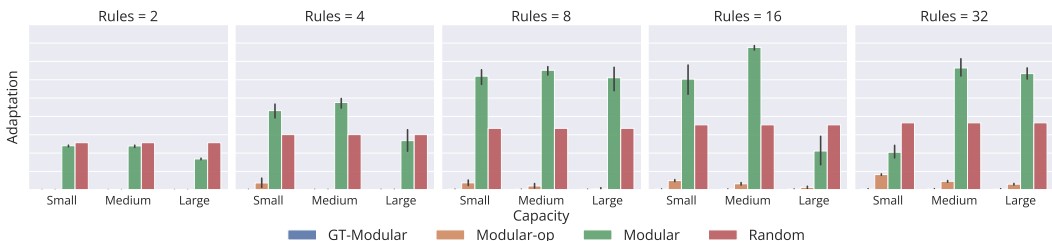

Figure 54: **Adaptation Metric for RNN-Regression Models.** Highlights how poor the specialization (*lower is better*) is of different models of varying capacities trained across different number of rules. Each bar on the graph is obtained from an average over five tasks, each with five seeds, totaling 25 runs.

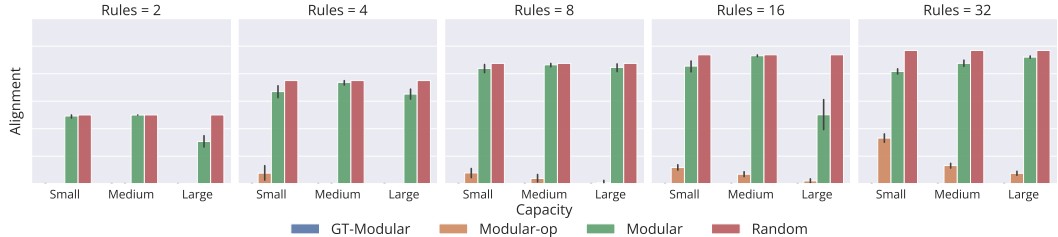

Figure 55: **Alignment Metric for RNN-Classification Models.** Highlights how poor the specialization (*lower is better*) is of different models of varying capacities trained across different number of rules. Each bar on the graph is obtained from an average over five tasks, each with five seeds, totaling 25 runs.

in the data and to Figures 87 - 91 where both the sequence length is potentially altered and also the distribution from which individual data points are sampled.

*Collapse-Avg.* For each rule setting, we report the *Collapse-Avg* metric score of the different models and the three different model capacities in Figure 51.

*Collapse-Worst.* For each rule setting, we report the *Collapse-Worst* metric score of the different models and the three different model capacities in Figure 52.

*Inverse Mutual Information.* For each rule setting, we report the *Inverse Mutual Information* metric score of the different models and the three different model capacities in Figure 53.

*Adaptation.* For each rule setting, we report the *Adaptation* metric score of the different models and the three different model capacities in Figure 54.

*Alignment.* For each rule setting, we report the *Alignment* metric score of the different models and the three different model capacities in Figure 55.

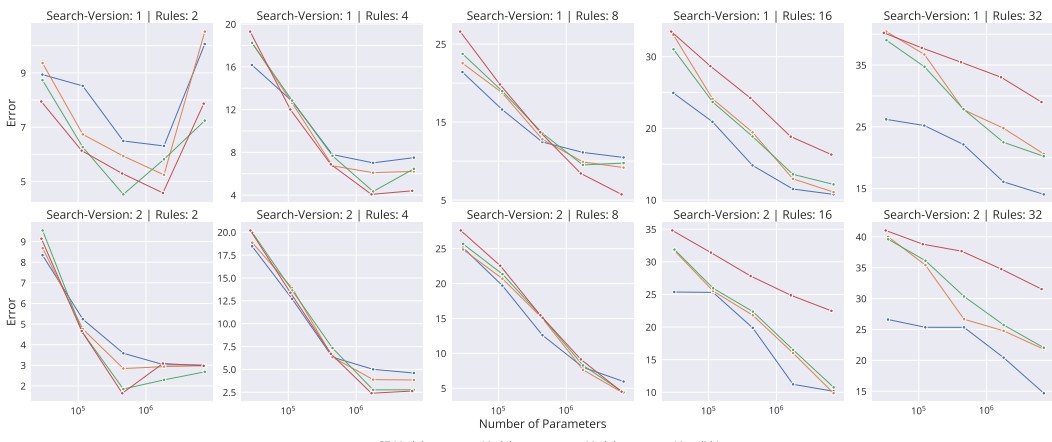

Figure 56: **Out-of-Distribution (Sequence Length: 3 - Individual Token Sampling: Same) Performance on MHA-Classification Models.** Performance (*lower is better*) of different models of varying capacities, different search versions in data and trained across different number of rules. Each point on the graph is obtained from an average over five tasks, each with five seeds, totaling 25 runs.

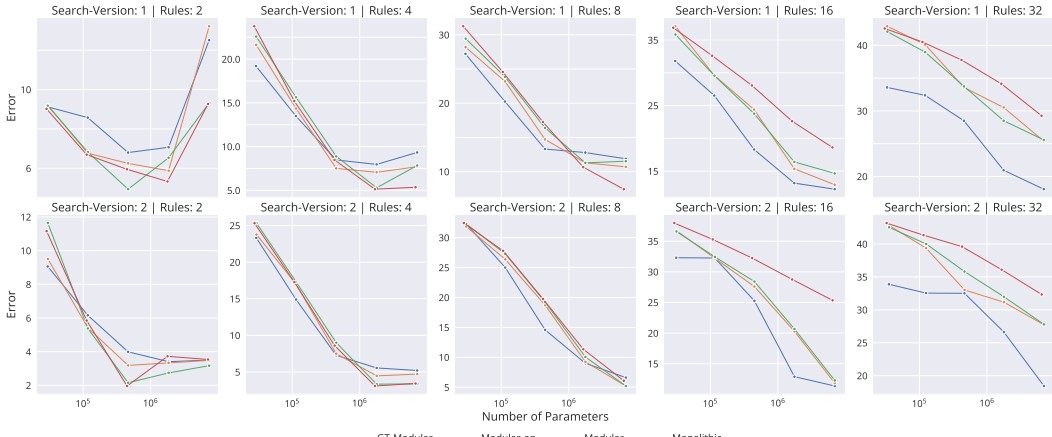

Figure 57: **Out-of-Distribution (Sequence Length: 5 - Individual Token Sampling: Same) Performance on MHA-Classification Models.** Performance (*lower is better*) of different models of varying capacities, different search versions in data and trained across different number of rules. Each point on the graph is obtained from an average over five tasks, each with five seeds, totaling 25 runs.

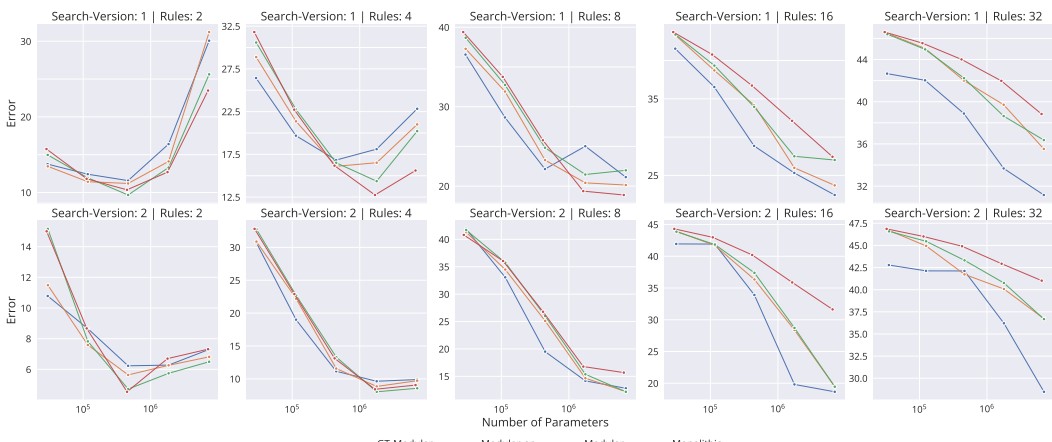

Figure 58: **Out-of-Distribution (Sequence Length: 20 - Individual Token Sampling: Same) Performance on MHA-Classification Models.** Performance (*lower is better*) of different models of varying capacities, different search versions in data and trained across different number of rules. Each point on the graph is obtained from an average over five tasks, each with five seeds, totaling 25 runs.

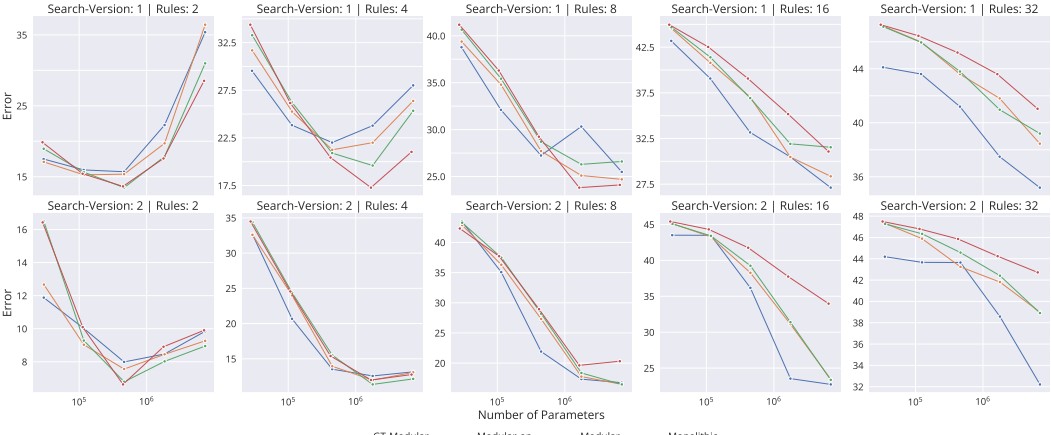

Figure 59: **Out-of-Distribution (Sequence Length: 30 - Individual Token Sampling: Same) Performance on MHA-Classification Models.** Performance (*lower is better*) of different models of varying capacities, different search versions in data and trained across different number of rules. Each point on the graph is obtained from an average over five tasks, each with five seeds, totaling 25 runs.

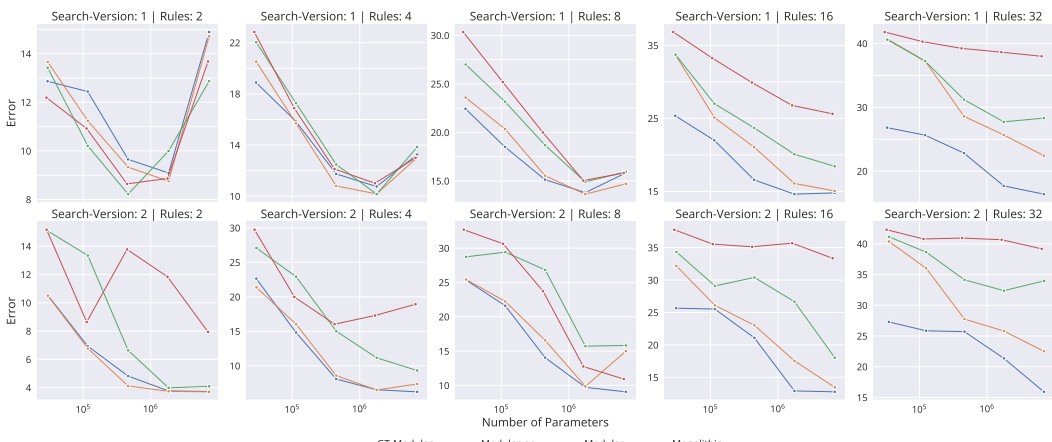

Figure 60: **Out-of-Distribution (Sequence Length: 3 - Individual Token Sampling: Altered) Performance on MHA-Classification Models.** Performance (*lower is better*) of different models of varying capacities, different search versions in data and trained across different number of rules. Each point on the graph is obtained from an average over five tasks, each with five seeds, totaling 25 runs.

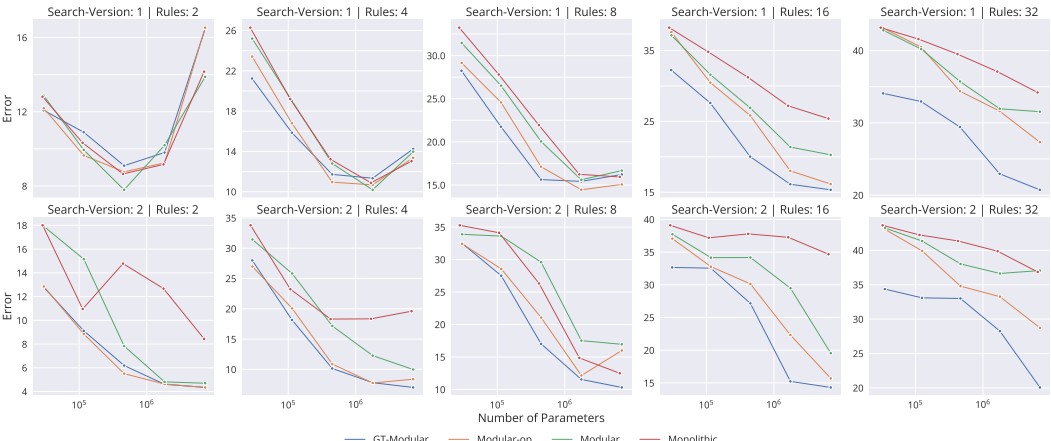

Figure 61: **Out-of-Distribution (Sequence Length: 5 - Individual Token Sampling: Altered) Performance on MHA-Classification Models.** Performance (*lower is better*) of different models of varying capacities, different search versions in data and trained across different number of rules. Each point on the graph is obtained from an average over five tasks, each with five seeds, totaling 25 runs.

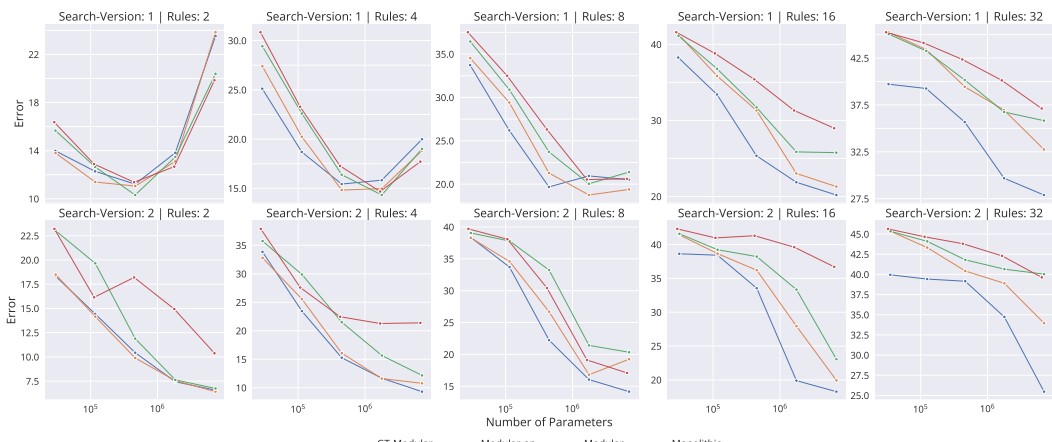

Figure 62: **Out-of-Distribution (Sequence Length: 10 - Individual Token Sampling: Altered) Performance on MHA-Classification Models.** Performance (*lower is better*) of different models of varying capacities, different search versions in data and trained across different number of rules. Each point on the graph is obtained from an average over five tasks, each with five seeds, totaling 25 runs.

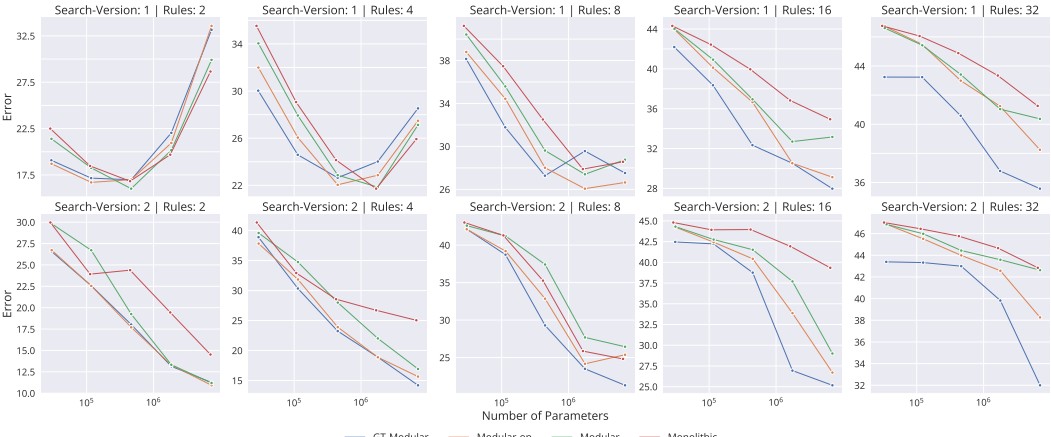

Figure 63: **Out-of-Distribution (Sequence Length: 20 - Individual Token Sampling: Altered) Performance on MHA-Classification Models.** Performance (*lower is better*) of different models of varying capacities, different search versions in data and trained across different number of rules. Each point on the graph is obtained from an average over five tasks, each with five seeds, totaling 25 runs.

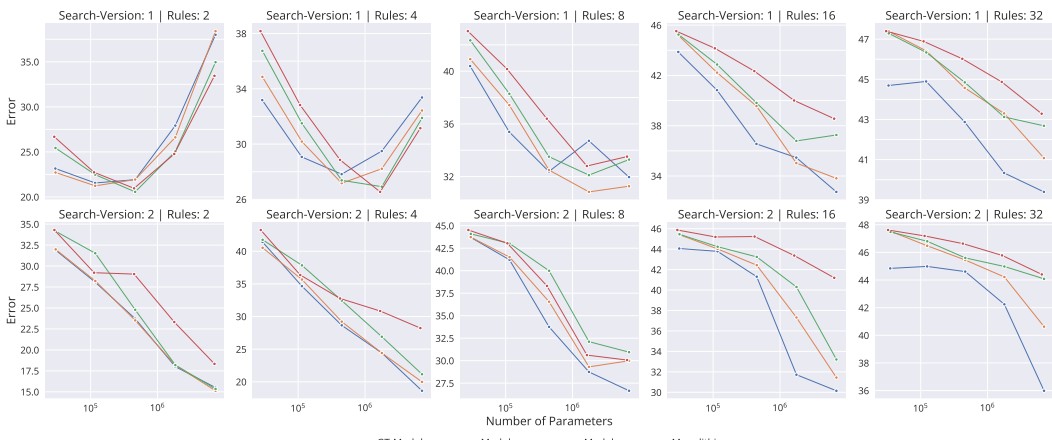

Figure 64: **Out-of-Distribution (Sequence Length: 30 - Individual Token Sampling: Altered) Performance on MHA-Classification Models.** Performance (*lower is better*) of different models of varying capacities, different search versions in data and trained across different number of rules. Each point on the graph is obtained from an average over five tasks, each with five seeds, totaling 25 runs.

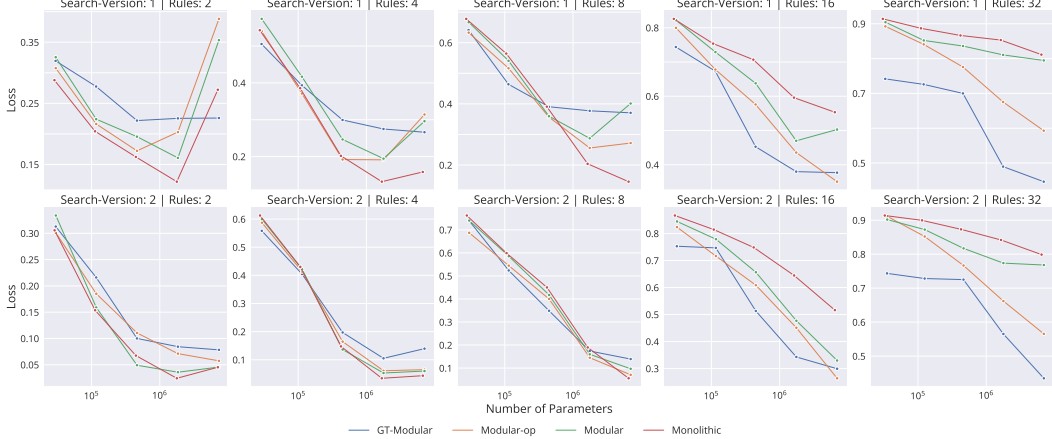

Figure 65: **Out-of-Distribution (Sequence Length: 3 - Individual Token Sampling: Same) Performance on MHA-Regression Models.** Performance (*lower is better*) of different models of varying capacities, different search versions in data and trained across different number of rules. Each point on the graph is obtained from an average over five tasks, each with five seeds, totaling 25 runs.

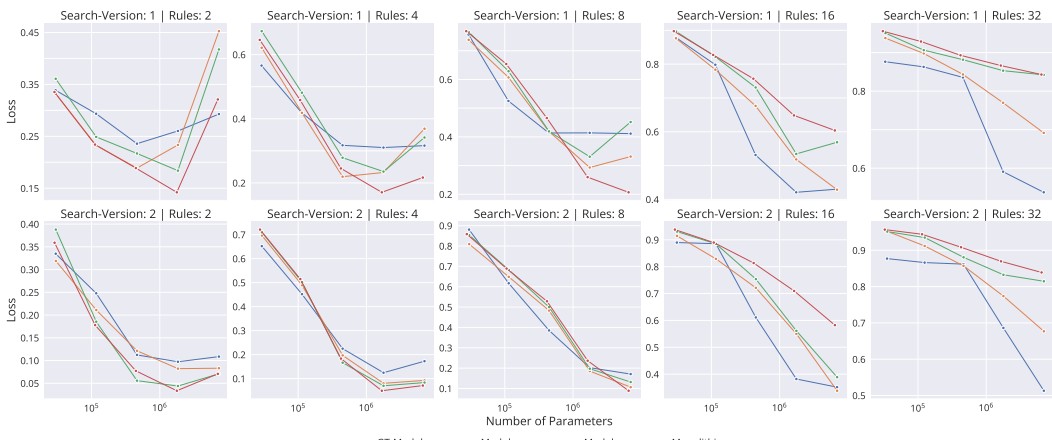

Figure 66: **Out-of-Distribution (Sequence Length: 5 - Individual Token Sampling: Same) Performance on MHA-Regression Models.** Performance (*lower is better*) of different models of varying capacities, different search versions in data and trained across different number of rules. Each point on the graph is obtained from an average over five tasks, each with five seeds, totaling 25 runs.

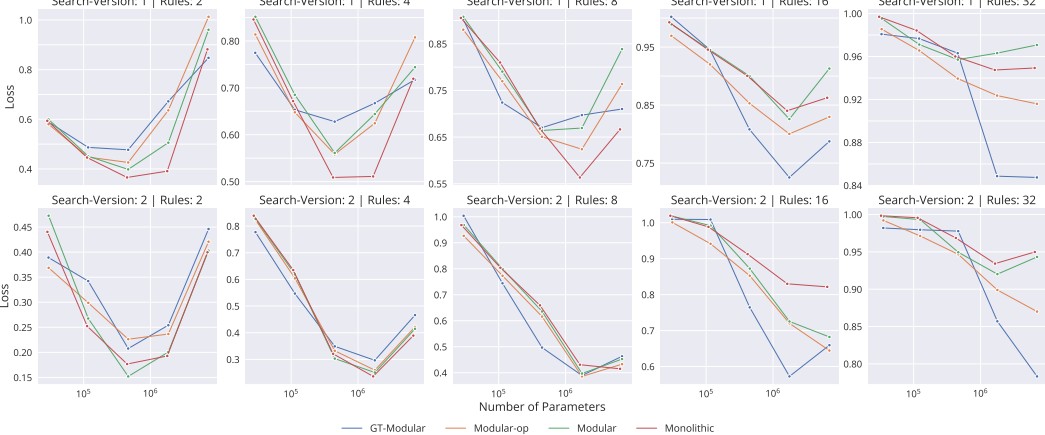

Figure 67: **Out-of-Distribution (Sequence Length: 20 - Individual Token Sampling: Same) Performance on MHA-Regression Models.** Performance (*lower is better*) of different models of varying capacities, different search versions in data and trained across different number of rules. Each point on the graph is obtained from an average over five tasks, each with five seeds, totaling 25 runs.

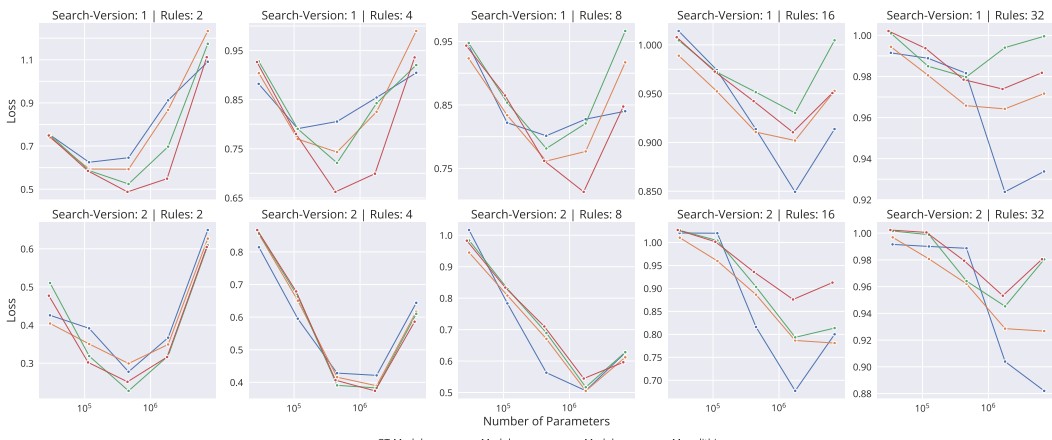

Figure 68: **Out-of-Distribution (Sequence Length: 30 - Individual Token Sampling: Same) Performance on MHA-Regression Models.** Performance (*lower is better*) of different models of varying capacities, different search versions in data and trained across different number of rules. Each point on the graph is obtained from an average over five tasks, each with five seeds, totaling 25 runs.

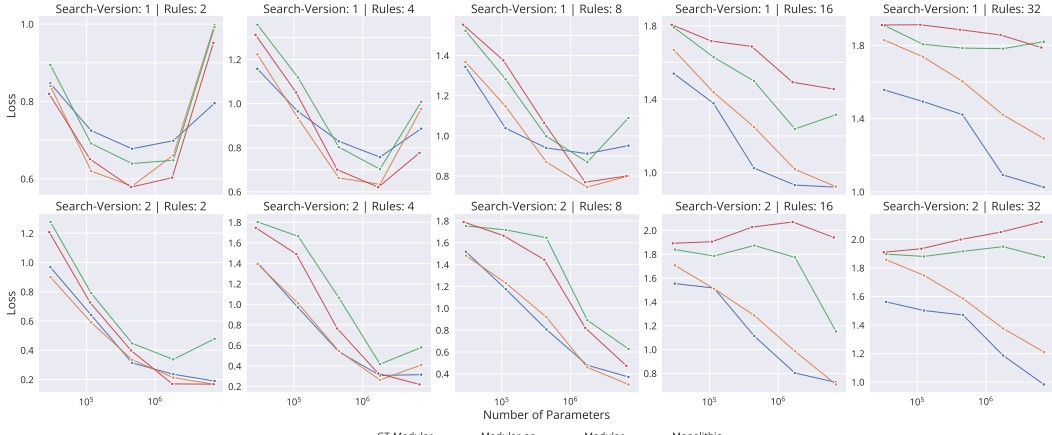

Figure 69: **Out-of-Distribution (Sequence Length: 3 - Individual Token Sampling: Altered) Performance on MHA-Regression Models.** Performance (*lower is better*) of different models of varying capacities, different search versions in data and trained across different number of rules. Each point on the graph is obtained from an average over five tasks, each with five seeds, totaling 25 runs.

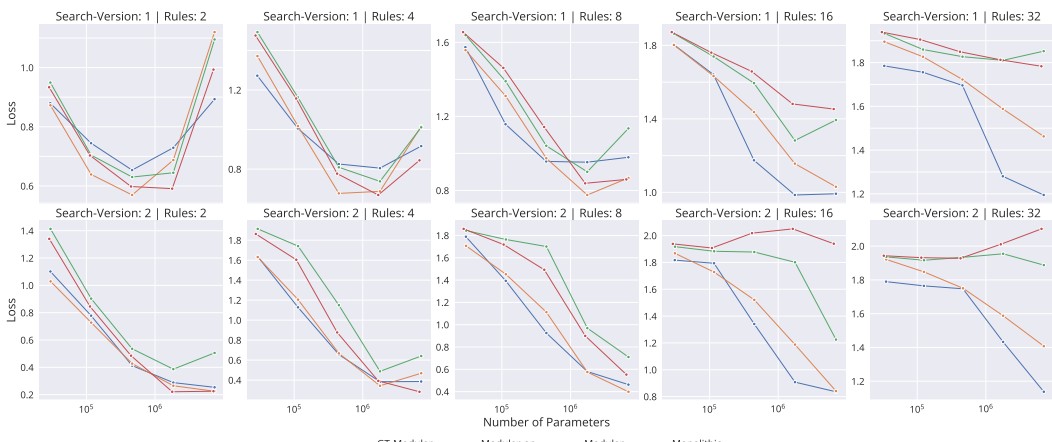

Figure 70: **Out-of-Distribution (Sequence Length: 5 - Individual Token Sampling: Altered) Performance on MHA-Regression Models.** Performance (*lower is better*) of different models of varying capacities, different search versions in data and trained across different number of rules. Each point on the graph is obtained from an average over five tasks, each with five seeds, totaling 25 runs.

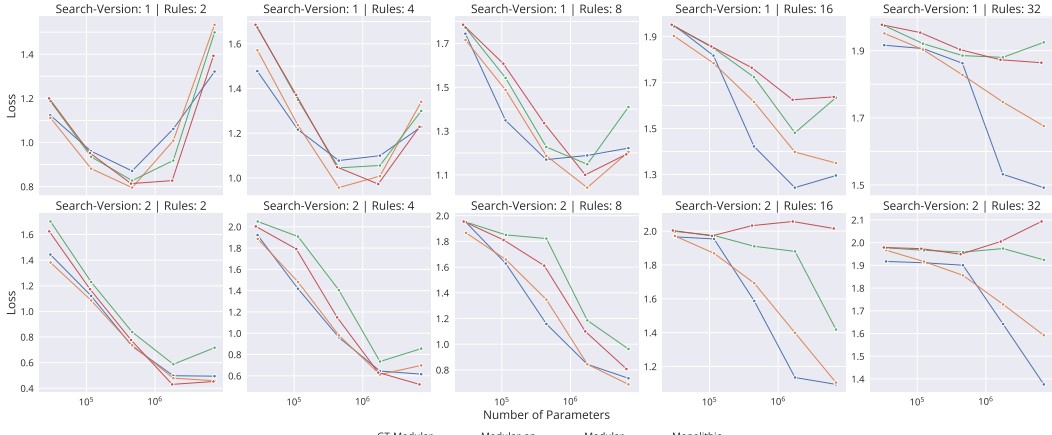

Figure 71: **Out-of-Distribution (Sequence Length: 10 - Individual Token Sampling: Altered) Performance on MHA-Regression Models.** Performance (*lower is better*) of different models of varying capacities, different search versions in data and trained across different number of rules. Each point on the graph is obtained from an average over five tasks, each with five seeds, totaling 25 runs.

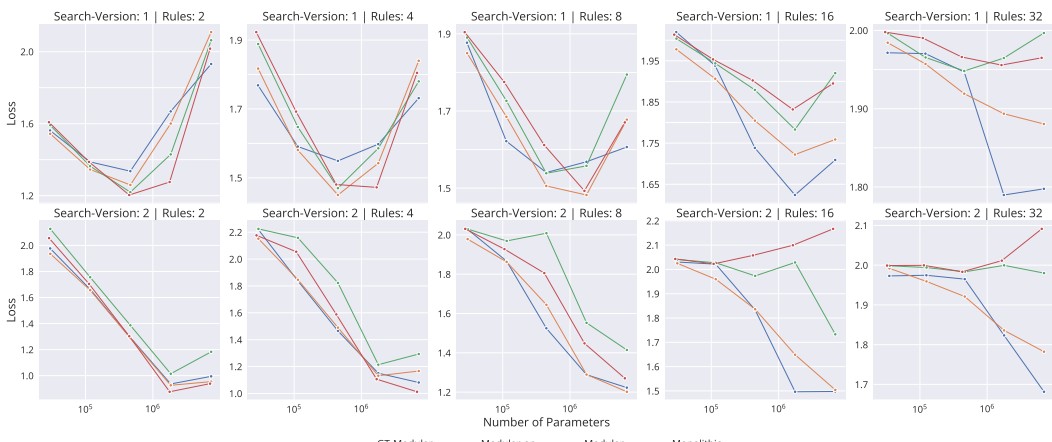

Figure 72: **Out-of-Distribution (Sequence Length: 20 - Individual Token Sampling: Altered) Performance on MHA-Regression Models.** Performance (*lower is better*) of different models of varying capacities, different search versions in data and trained across different number of rules. Each point on the graph is obtained from an average over five tasks, each with five seeds, totaling 25 runs.

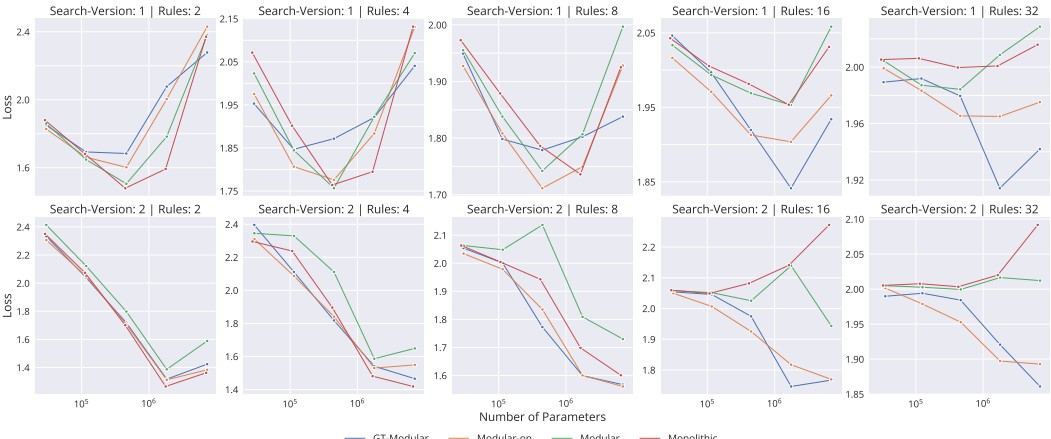

Figure 73: **Out-of-Distribution (Sequence Length: 30 - Individual Token Sampling: Altered) Performance on MHA-Regression Models.** Performance (*lower is better*) of different models of varying capacities, different search versions in data and trained across different number of rules. Each point on the graph is obtained from an average over five tasks, each with five seeds, totaling 25 runs.

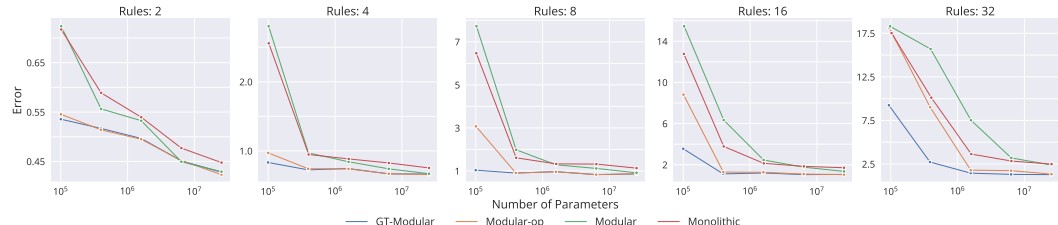

Figure 74: **Out-of-Distribution (Sequence Length: 3 - Individual Token Sampling: Same) Performance on RNN-Classification Models.** Performance (*lower is better*) of different models of varying capacities and trained across different number of rules. Each point on the graph is obtained from an average over five tasks, each with five seeds, totaling 25 runs.

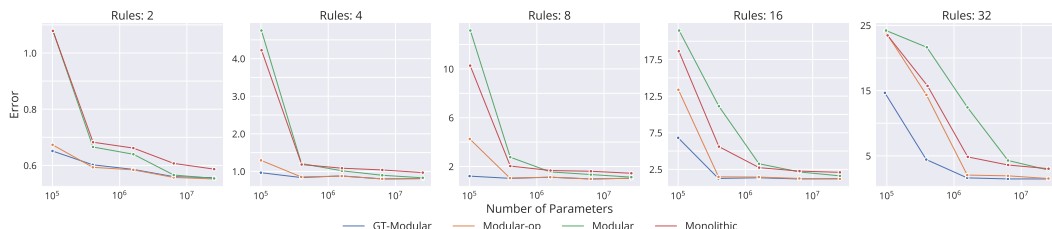

Figure 75: **Out-of-Distribution (Sequence Length: 5 - Individual Token Sampling: Same) Performance on RNN-Classification Models.** Performance (*lower is better*) of different models of varying capacities and trained across different number of rules. Each point on the graph is obtained from an average over five tasks, each with five seeds, totaling 25 runs.

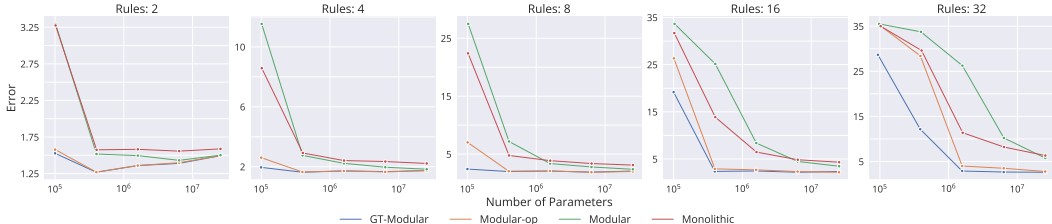

Figure 76: **Out-of-Distribution (Sequence Length: 20 - Individual Token Sampling: Same) Performance on RNN-Classification Models.** Performance (*lower is better*) of different models of varying capacities and trained across different number of rules. Each point on the graph is obtained from an average over five tasks, each with five seeds, totaling 25 runs.

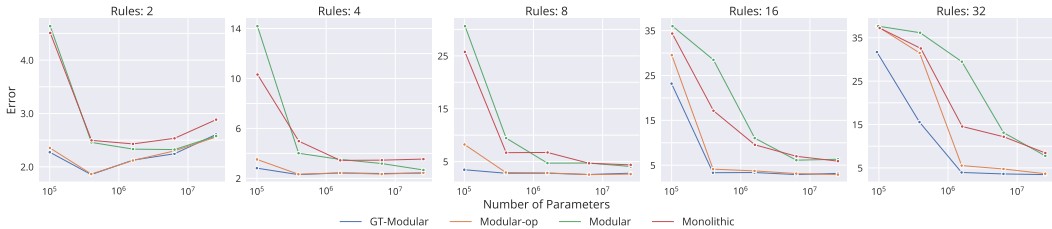

Figure 77: **Out-of-Distribution (Sequence Length: 30 - Individual Token Sampling: Same) Performance on RNN-Classification Models.** Performance (*lower is better*) of different models of varying capacities and trained across different number of rules. Each point on the graph is obtained from an average over five tasks, each with five seeds, totaling 25 runs.

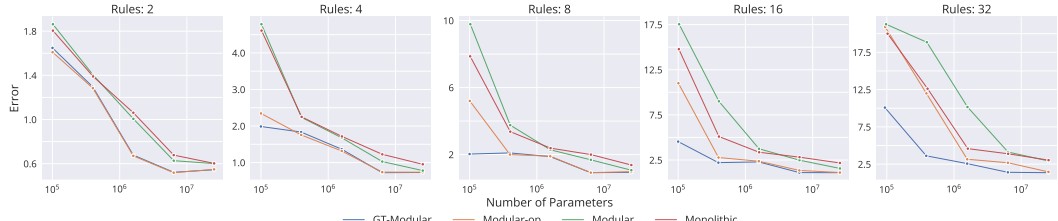

Figure 78: **Out-of-Distribution (Sequence Length: 3 - Individual Token Sampling: Altered) Performance on RNN-Classification Models.** Performance (*lower is better*) of different models of varying capacities and trained across different number of rules. Each point on the graph is obtained from an average over five tasks, each with five seeds, totaling 25 runs.

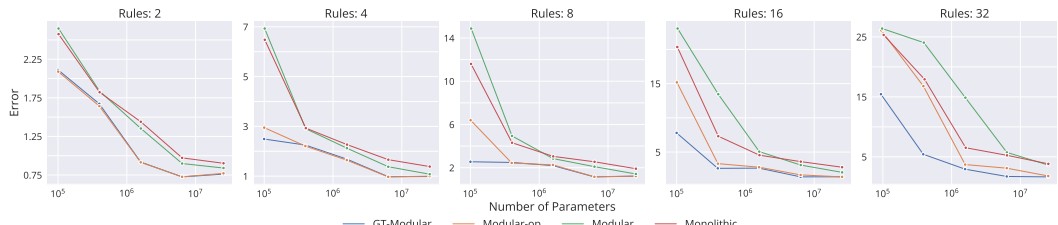

Figure 79: **Out-of-Distribution (Sequence Length: 5 - Individual Token Sampling: Altered) Performance on RNN-Classification Models.** Performance (*lower is better*) of different models of varying capacities and trained across different number of rules. Each point on the graph is obtained from an average over five tasks, each with five seeds, totaling 25 runs.

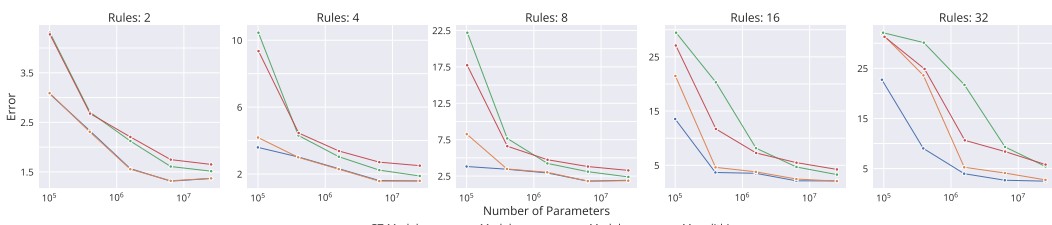

Figure 80: **Out-of-Distribution (Sequence Length: 10 - Individual Token Sampling: Altered) Performance on RNN-Classification Models.** Performance (*lower is better*) of different models of varying capacities and trained across different number of rules. Each point on the graph is obtained from an average over five tasks, each with five seeds, totaling 25 runs.

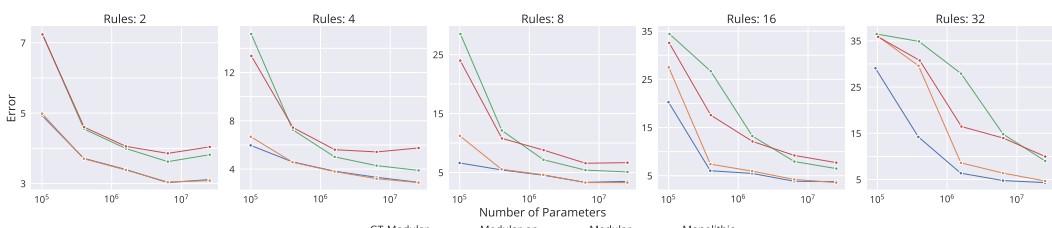

Figure 81: **Out-of-Distribution (Sequence Length: 20 - Individual Token Sampling: Altered) Performance on RNN-Classification Models.** Performance (*lower is better*) of different models of varying capacities and trained across different number of rules. Each point on the graph is obtained from an average over five tasks, each with five seeds, totaling 25 runs.

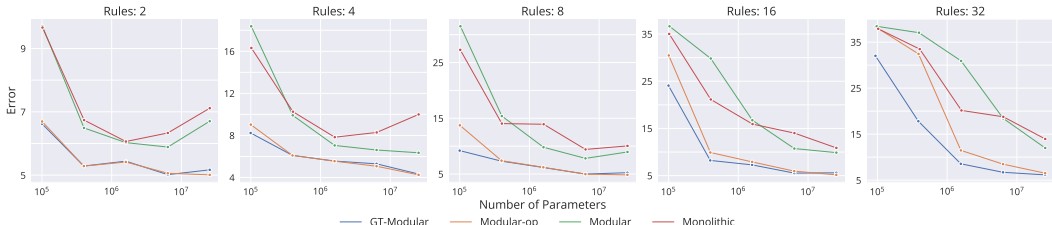

Figure 82: **Out-of-Distribution (Sequence Length: 30 - Individual Token Sampling: Altered) Performance on RNN-Classification Models.** Performance (*lower is better*) of different models of varying capacities and trained across different number of rules. Each point on the graph is obtained from an average over five tasks, each with five seeds, totaling 25 runs.

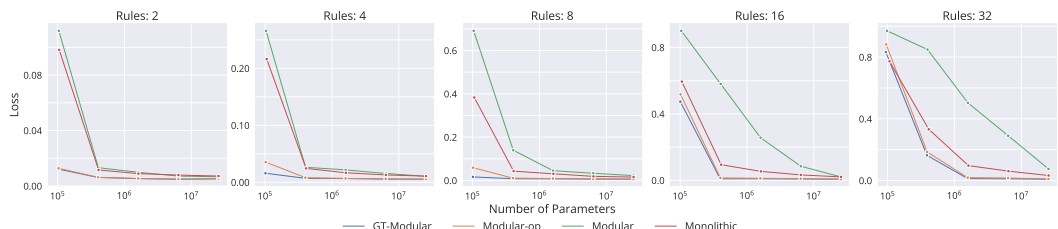

Figure 83: **Out-of-Distribution (Sequence Length: 3 - Individual Token Sampling: Same) Performance on RNN-Regression Models.** Performance (*lower is better*) of different models of varying capacities and trained across different number of rules. Each point on the graph is obtained from an average over five tasks, each with five seeds, totaling 25 runs.

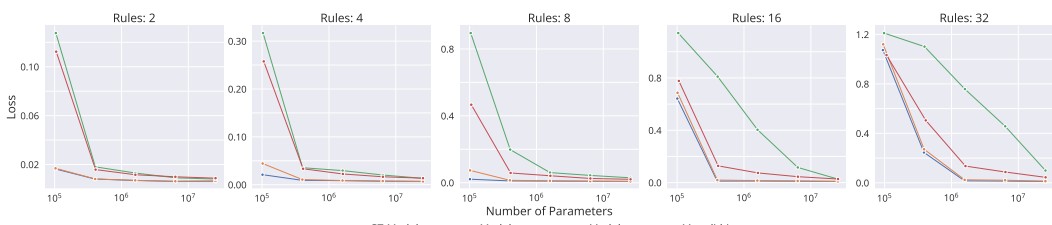

Figure 84: **Out-of-Distribution (Sequence Length: 5 - Individual Token Sampling: Same) Performance on RNN-Regression Models.** Performance (*lower is better*) of different models of varying capacities and trained across different number of rules. Each point on the graph is obtained from an average over five tasks, each with five seeds, totaling 25 runs.

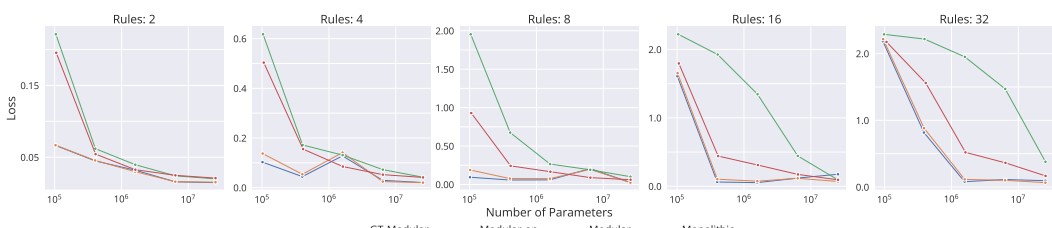

Figure 85: **Out-of-Distribution (Sequence Length: 20 - Individual Token Sampling: Same) Performance on RNN-Regression Models.** Performance (*lower is better*) of different models of varying capacities and trained across different number of rules. Each point on the graph is obtained from an average over five tasks, each with five seeds, totaling 25 runs.

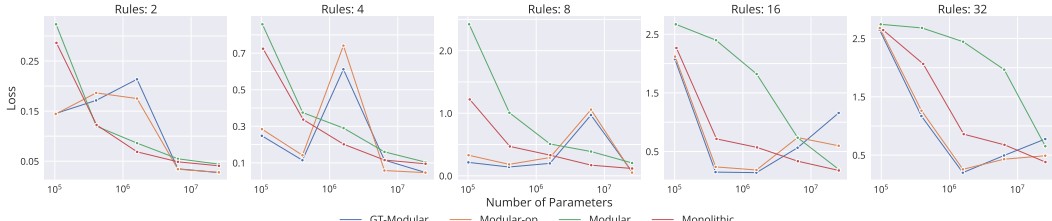

Figure 86: **Out-of-Distribution (Sequence Length: 30 - Individual Token Sampling: Same) Performance on RNN-Regression Models.** Performance (*lower is better*) of different models of varying capacities and trained across different number of rules. Each point on the graph is obtained from an average over five tasks, each with five seeds, totaling 25 runs.

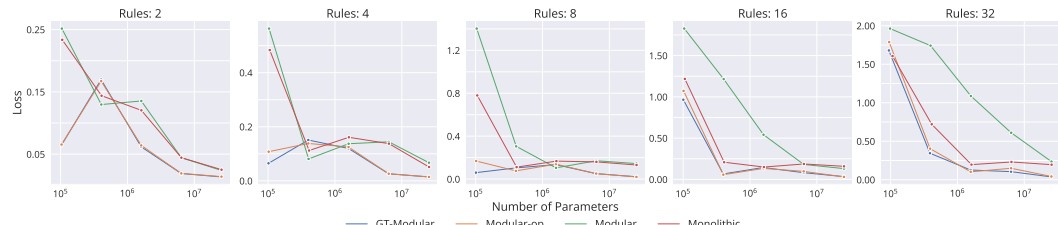

Figure 87: **Out-of-Distribution (Sequence Length: 3 - Individual Token Sampling: Altered) Performance on RNN-Regression Models.** Performance (*lower is better*) of different models of varying capacities and trained across different number of rules. Each point on the graph is obtained from an average over five tasks, each with five seeds, totaling 25 runs.

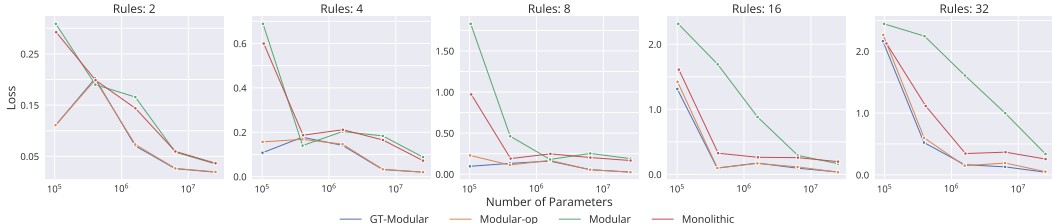

Figure 88: **Out-of-Distribution (Sequence Length: 5 - Individual Token Sampling: Altered) Performance on RNN-Regression Models.** Performance (*lower is better*) of different models of varying capacities and trained across different number of rules. Each point on the graph is obtained from an average over five tasks, each with five seeds, totaling 25 runs.

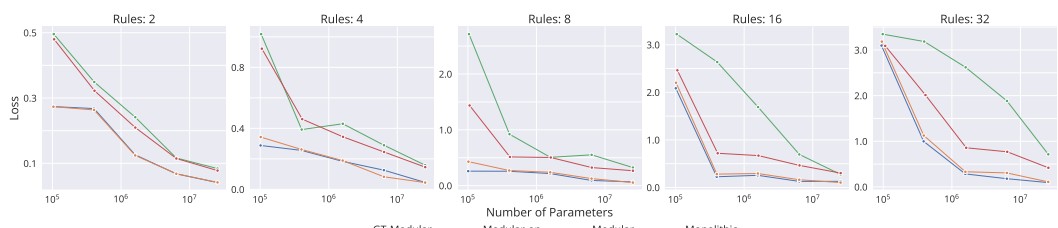

Figure 89: **Out-of-Distribution (Sequence Length: 10 - Individual Token Sampling: Altered) Performance on RNN-Regression Models.** Performance (*lower is better*) of different models of varying capacities and trained across different number of rules. Each point on the graph is obtained from an average over five tasks, each with five seeds, totaling 25 runs.

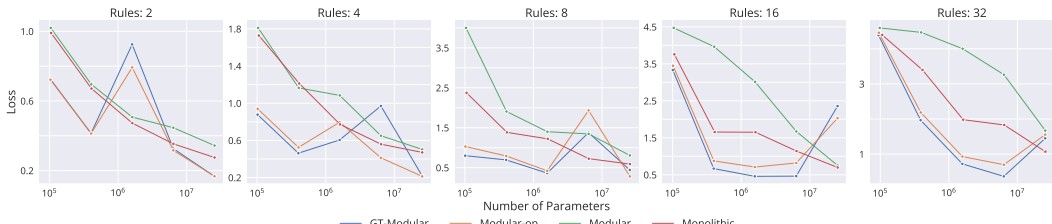

Figure 90: **Out-of-Distribution (Sequence Length: 20 - Individual Token Sampling: Altered) Performance on RNN-Regression Models.** Performance (*lower is better*) of different models of varying capacities and trained across different number of rules. Each point on the graph is obtained from an average over five tasks, each with five seeds, totaling 25 runs.

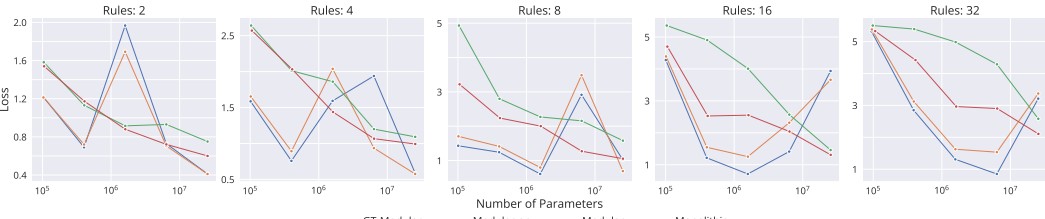

Figure 91: **Out-of-Distribution (Sequence Length: 30 - Individual Token Sampling: Altered) Performance on RNN-Regression Models.** Performance (*lower is better*) of different models of varying capacities and trained across different number of rules. Each point on the graph is obtained from an average over five tasks, each with five seeds, totaling 25 runs.