# OpenReview forum: "Is a Modular Architecture Enough?"
_NeurIPS.cc/2022/Conference — NeurIPS 2022 Accept_

### Official Review · Reviewer_QEMJ · 2022-07-08

**Rating:** 6
**Confidence:** 4
**Soundness:** 3 good
**Presentation:** 3 good
**Contribution:** 3 good

**Summary:**

This paper presents an elegantly designed experiment to evaluate the effectiveness of a number of modular architectures in various settings. The topic is an important one, in my opinion, since modular architectures have the potential to address a number of key issues in ML, including compositionality and continual learning. The question the authors address is whether backpropagation can discover the structure in data that is inherently suited to modular architectures (because it is synthetic and designed to be) and learn to specialise in modules accordingly. By comparing with monolithic architectures at one end of the spectrum and with modular architectures forced to specialise (thanks to oracular knowledge of the data) at the other, they are able to assess both a) the extent to which perfect modularisation improves performance, and b) how well modularity can be learned by backpropagation. The results suggest that - within the narrow setting of the experiment - a) modularity does improve performance on sufficiently complex data, but b) backpropagation struggles to discover the underlying structure in the data and to learn to specialise accordingly.

**Questions:**

What do the authors mean by a “mixture experts distribution”? I am only familiar with MoE in the context of architectures, not distributions (and a quick search on Google backs this up).

What is “I” in equations 5, 8, and 13? Why not just make it 1?

See also questions above.

**Limitations:**

See above

**Strengths And Weaknesses:**

The paper concerns the important topic of modularity and presents a well thought out experiment addressing an interesting question. The results are informative and interesting. Overall, this is good science, and the sort of thing we should see more of at the big ML venues. The main weakness of the paper, I feel, is that the synthetic data is very simple - just two real-valued variables, plus an integer context variable, parameterised by just two real values. Do we expect the results to apply with more complex datasets and large architectures? I’m not sure. Perhaps it’s easier for backpropagation to discover structure in the data at scale than in a small, simple dataset.

The paper isn't very explicit about the model architectures used in the experiment, either in the main paper or in the appendix. I assumed they would be very small, given the low-dimensionality of the synthetic data. Delving into the code (thanks for providing this), I see the model architectures are a bit more nuanced than I expected. I suspected this is to help ensure the modular and monolithic versions had the same numbr of parameters? But I also see that the modular architecture has a softmax in there. Does this realise some form of competition between the modules? If so, this isn't mentioned in the main paper.

---

> ### Author Response · Authors · 2022-08-02
> **Official Rebuttal**
>
> We thank the reviewer for their useful comments about our work as well as for recognizing its importance. We provide implementation details of the models as well as the connections to real world tasks in a separate comment to all reviewers above and have revised the main text and the Appendix to reflect the same. Below, we treat additional concerns, and are hopeful that our answers and edits adequately address the reviewer’s concerns.
>
> **Simplicity of Dataset**
>
> While we agree that the datasets considered are simple, this is by design to adequately evaluate models. However, we suspect there might be a misunderstanding about the complexity of the tasks considered. It is true that for our MLP experiments, the inputs are just two real values and an integer context. However, for experiments on recurrent domains, the input consists of a sequence of 32 dimensional vectors with the sequence length 10, as well as a sequence of integer contexts. Similarly, our experiments on attention consider inputs to be a set of $4R$ or $6R$ dimensional inputs along with integer context for each element of the set, where $R$ is the number of rules. Another measure of complexity for our tasks is the number of rules themselves, which we range from 2 to 32. In light of these experimental setups, we note that this is close to an infinite data regime.
>
> **Extrapolation to Larger Models and Complex Datasets**
>
> We perform our experimentation over a range of model sizes, with the larger models having 1 million or 10 million parameters depending on the setting (eg. check right-most points in the Figures 25 and 41 in the Appendix). We would also like to stress that the data considered is not small since new data is sampled at every training iteration (infinite-data regime). A working hypothesis which we have is that if specialization is so difficult to learn with simpler data available in abundance, it would be much more difficult to obtain in more complex and limited data regimes. This is further discussed in the comment on real-world applications and Appendix B where we talk about considerations to be kept in mind when extending to more real-world domains, as well as limitations.
>
> **Implementation Details**
>
> We thank the reviewer for pointing this out. We refer the reviewer to the general comment on the implementation details of all the models and tasks, and note that we have amended the text for increased clarity on this topic (see Appendices D - F). Furthermore, we clarify that ablation experiments run over a large number of model sizes, typically ranging from 100 thousand to 10 million parameters, which suggest clear trends and validity of our findings over a range of model sizes.
>
> **Nuance in Model Implementation**
>
> The nuances in the implementation details of attention-based and recurrent experiments that the reviewer is astutely pointing to is done to ensure that the sizes of the different models are similar. This requires a non-trivial computation because in a recurrent system, a monolithic RNN with hidden size 256 has more parameters than a mixture of 4 RNN models with 64 hidden size each, but has less parameters than a mixture of 4 RNN models with 256 hidden size. The computations done in the code are to obtain an estimate of the hidden size that maintains that the total number of parameters in the two systems are similar. We add this detail in the relevant sections of the Appendix (E and F), and apologize for the oversight.
>
> **Softmax in Modular Systems**
>
> We have added this detail in the Model Setup sections in the Appendices D, E and F. Essentially the $p_m$’s in Table 1 define the probability of activation of module $m$, and thus the vector $p$ represents a probability vector which can be obtained through a softmax. We thank the reviewer for pointing out this area of potential confusion.
>
> **Mixture of Experts Distribution**
> We apologize for the overload of notation here. When we talk about the data distribution, it is actually a Mixture distribution (https://en.wikipedia.org/wiki/Mixture_distribution) where each component of the mixture can be thought of as a distribution/functional mapping that an expert in a MoE model should learn to represent. We clarify this point in the revised manuscript.
>
> **I in the Equations**
>
> Since not all data is uni-dimensional, we use $I$ to represent the identity matrix. Hence, a number of times the data is sampled from a gaussian with identity covariance matrix if the data is multi-dimensional. We add a clarification statement in the text to this effect.
>
> We hope that the additional details and clarifications provided help paint a clearer picture of our work and resolve the reviewer’s concerns. We would also be happy to address any additional questions that the reviewer may have. We thank the reviewer for the engaged and astute comments which helped us considerably improve the paper.

---

> > ### Comment · Reviewer_QEMJ · 2022-08-05
> > **Thanks for response**
> >
> > Thanks to the authors for their response to my review. I think they are right that I had underestimated the complexity of the tasks. Mainly because of this, but also taking into account the other improvements, I have updated my overall score.

---

> > > ### Author Response · Authors · 2022-08-07
> > > **Additional Information?**
> > >
> > > Thanks to the reviewer for responding and updating their score. If the reviewer has any additional concerns or clarifications that we can help resolve, we would be happy to. Please do let us know in case of any questions.

---

### Official Review · Reviewer_uXNv · 2022-07-11

**Rating:** 6
**Confidence:** 4
**Soundness:** 3 good
**Presentation:** 4 excellent
**Contribution:** 2 fair

**Summary:**

The paper studies the benefits of MoE like modular network, in terms of many metrics, e.g.,  in/out of distribution performance, collapse-avg/worst, alignment, adaptation and inverse mutual information. The authors generate data, rules and tasks by synthetic neural process, and study monolithic, modular, modular-op model architectures against the ground-truth modular structure. They point out that an architecture with modular prior is not enough to perfectly learn the ground truth.

**Questions:**

1. It seems that modular-op usually comes with better performance in all metrics defined by the authors, but how general this conclusion can be? This is an important conclusion, which probably can guide us to design task level, task and input level, or input level (e.g., token level) gating. My major concern is that, the generating process of the synthetic data may not necessarily match any real-world case.

2. The conclusion of paper seems a little bit straight-forward, considering the synthetic data-generation process. It would be more informative If the authors can further study transfer ability on modular structures.

3. In the experiment, is the capacity of a monolithic model the same as modular & modular-op & gt-modular?

**Strengths And Weaknesses:**

\+ Whether modularity architecture helps multi-task learning is studied from a well-defined perspective.

\+ The data process and the developed metrics sound reasonable.

\+ The paper is well-organized and easy to follow.

\- It's unknown how the proposed data process can impact on real-world rules & data setting.

\- A more important and meaningful metric, in addition to the proposed ones, could be transfer ability, or compositional generalization ability on new task, which is thought to be a key advantage of sparse and modularity design. The conclusion of the paper is less informative without this part.

\- MoE structure is only one of the implementations of modular architecture, and the title is somehow ambiguous.

---

> ### Author Response · Authors · 2022-08-02
> **Official Rebuttal (1/2)**
>
> We thank the reviewer for their insightful comments about our work, and are hopeful that the steps taken (outlined below) will address their concerns. We also refer the reviewer to the general comment that was made to all reviewers for additional and pertinent information. Importantly, we now provide connections to real world tasks in one of the separate comments above.
>
> **Impact on Real-World Data Setting**
>
> We refer the reader to the separate comment on the connections to real world settings as well as possible extensions to more real world and complex settings. We hope that this discussion would provide insights into the impact of this work on more real-world domains, and that our efforts are adequate to address the reviewer’s concerns.
>
> **Transfer Ability**
>
> We agree with the reviewer that this is an exciting topic to explore with our experimental setup. Furthermore, it would be interesting to also consider performance and sample complexity based metrics for transfer ability across different tasks and even some pre-training and fine-tuning setup. However, to do this, we require not only mixture-distribution based tasks but also some notion of similarity between tasks because transfer of knowledge only makes sense when there is some commonality between the pre-training and fine-tuning setups and thus, one needs to design tasks and mixture distributions that are related to each other so that analysis can be done on pre-training on a set of tasks and then fine-tuning on another set of different but related tasks. This requires a lot more deliberation on the design choices and we believe that it should be a stand-alone contribution in itself and out of scope of this work. Once a proper set of task distributions are decided, our metrics can then be readily usable for further analysis. We refer the reviewer to the separate comment on real-world extensions and considerations for a more detailed discussion about this. Finally, we thank the reviewer for this insightful comment which has spurred early thinking that will likely lead to a follow up project. In sum, we respectfully reiterate that we believe the question of transfer ability falls outside of the scope of the current paper, if investigated properly, and hope the reviewer agrees with the value of our current results as well as its potential for expansion into future work.
>
> **MoE Structure**
>
> We use the MoE structure as the implementation choice of modular systems since it is the most commonly used implementation of modular systems (see examples cited below). We would be quite interested to know other implementations that do not share this flavor or cannot be framed as an MoE.
>
> * Recurrent Independent Mechanisms; Goyal et. al 2019
> * Learning to Combine Top-Down and Bottom-Up Signals in Recurrent Neural Networks with Attention over Modules; Mittal et. al 2020
> * Object Files and Schemata: Factorizing Declarative and Procedural Knowledge in Dynamical Systems; Goyal et. al 2020
> * Dynamic Inference with Neural Interpreters; Rahaman et. al 2021
> * Compositional Attention: Disentangling Search and Retrieval; Mittal et. al 2021
> * Transformers with competitive ensembles of independent mechanisms; Lamb et. al 2021
> * Fast and slow learning of recurrent independent mechanisms; Madan et. al 2021
> * Neural Production Systems; Goyal et. al 2021
> * Routing Networks and the Challenges of Modular and Compositional Computation; Rosenbaum et. al 2019
>
> **Generalizability of Modular-op System**
>
> Our primary focus of using Modular-op system is as part of the benchmark to illustrate (a) the benefits of a modular system that decides module selection based on the correct information, i.e. ignores the irrelevant futures for module selection, and (b) to show the downsides of gradient-based learning for specialization as it also suffers from the problems of collapse and specialization, just to a lesser extent.
>
> We first see that in the synthetic setting we considered, there is only one notion of specialization which is governed by the rule contexts $c$. In this setting, by driving the module selection through only $c$, we get better performance. Instead of leveraging it as a model for real-world tasks, we instead propose using Modular-op and GT-Modular to rank a notion of specialization for a task. A concrete example would be to think of multilingual language modeling and see in this setup whether driving specialization (either in Modular-op or GT-Modular) at the language level leads to better performance, or at the level of families of languages, or neither.
>
> Once we have obtained a notion of specialization with the help of GT-Modular and Modular-op systems, we can leverage additional task inputs according to that notion for better performance.

---

> > ### Author Response · Authors · 2022-08-02
> > **Official Rebuttal (2/2)**
> >
> > **Straight-forward Conclusion**
> >
> > We would like to argue how our findings are not obvious (straight-forward). We test on very simple rule-based data settings in an infinite-data regime, which is arguably the easiest setting to discover specialization since there is an abundance of data. However, even in this simplified setting we see that specialization is not always obtained equally in all considered architectures, which points to two possibilities: (a) either obtaining perfect specialization is impossible without any manual engineering, or (b) we need some inductive biases or regularization schemes to incentivize specialization and prevent collapse in such systems.
> >
> > Building on this conclusion, our contribution also provides a test-bed for the study of such inductive biases by providing concrete quantitative metrics that can be used to quantify the problems experienced by such systems. We believe this is a first and important step to move away from single-example visualizations as evidence for specialization to more rigorous quantitative assessment. We see our synthetic suite of tasks as an evaluation tool for network architectures, and not as toy tasks as a proxy for real-world ones. We believe network architecture design needs to move beyond trial and error, and that careful architectural evaluation will be an integral part of system design in the future. Our work is an early but necessary step in that direction.
> >
> > **Capacity**
> >
> > We refer the readers to the details about the implementations provided in a separate common comment above. All our corresponding comparisons between models are controlled for the number of parameters, and we also do analysis over different model sizes.
> >
> > We hope that the additional details and clarifications provided provide a clearer picture of our work and resolve the reviewer’s concerns. We would also be happy to address any additional questions that the reviewer may have.

---

> > > ### Author Response · Authors · 2022-08-07
> > > **Request for Discussion**
> > >
> > > Given that the author-reviewer discussion period is coming to a close soon, we request the reviewer to let us know if our responses have resolved their concerns and if there are any other questions that we can address.

---

> > > ### Comment · Reviewer_uXNv · 2022-08-08
> > > **Thanks for response**
> > >
> > > Thanks for your response. I agree that transfer ability is better considered in another paper. I also agree that real world case is more complex and the paper gives a more clear setting, the work is an early but necessary step in the direction of learning with modularity. I have updated my score (5->6).
> > >
> > > 1. The authors may clarify in their paper that MoE is a typical implementation to equip modularity prior, but it is not the only choice and MoE is not equal to modularity. I agree modular network is usually implemented by MoE nowadays. But it's also a very strong prior compared to the definition of modularity itself,  ``a pattern of connectedness in which elements are grouped into highly connected subsets, which are more loosely connected to other such groups.'' [2]. Some others [1] also take e.g., sparse regularization as modular priors. We can also implement modular prior inspected by [3] using task-specific masks.
> > >
> > > 2. In the experiments, we see the advantage to achieve perfect specialization, i.e., GT-modular. However, is it really a necessary path for AI to achieve such perfect specialization in real-world cases, so that it can learn things modularly and generalize more well? I still have concerns about how the conclusion made by the paper can guide the real-world AI design.
> > >
> > > [1] Clune, J., Mouret, J. B., & Lipson, H. (2013). The evolutionary origins of modularity.
> > >
> > > [2] Wagner, G. P., Pavlicev, M., & Cheverud, J. M. (2007). The road to modularity.
> > >
> > > [3] Csordás, R., van Steenkiste, S., & Schmidhuber, J. (2020). Are neural nets modular? inspecting functional modularity through differentiable weight masks.

---

> > > > ### Author Response · Authors · 2022-08-08
> > > > **Additional Clarifications**
> > > >
> > > > We thank the reviewer for their time and response and are grateful for the score increase.
> > > >
> > > > 1. We understand the reviewer's point and will revise the text to mention the specific type of modularity that we are talking about, which are dynamic modules that can be queried and are interpretable. While one can have modular systems embedded in a more general network through loose and highly connected circuits, it becomes non-trivial to discover them which poses another challenge. Further, we talk about dynamic sparsity here, which is context dependent selection of different "modules" or circuits, which is not the case in static sparse regularized networks. We will add a discussion point on these in a revision of the draft to further clarify and emphasize the type of modularity that we discuss in the work.
> > > > 2. We completely echo the reviewer's sentiments here. However, we would like to point that to understand whether this perfect specialization is a necessary path for AI, we do need a test-bed and metrics to build the foundation for a systematic study of specialization over simple to complex domains. As described in the additional comment on connections to real-world settings, one can use this same style of analysis with the notion of specialization being at the level of languages in multilingual language modeling, to discover whether this notion of specialization is actually meaningful and beneficial in terms of performance.
> > > >
> > > > We hope that this further clarifies the reviewer's concerns. Please do not hesitate to communicate other details that you believe would improve the paper.

---

### Official Review · Reviewer_H4Mv · 2022-07-11

**Rating:** 7
**Confidence:** 4
**Soundness:** 4 excellent
**Presentation:** 3 good
**Contribution:** 3 good

**Summary:**

This paper carefully and thoroughly examines recent trends around modularity in neural architectures, with a special focus on recent sparse mixture-of-experts (MoE) models through construction of synthetic “rule-based” tasks. These tasks specifically target both the learning and generalization potential of these architectures, showing how various architectural inductive biases perform in the presence of multiple “rules/tasks” (different pathways in an MoE for example), and in-distribution/out-of-distribution data. Using the proposed rule-based data generation procedure and evaluating three core architectures (MLPs, self-attention, and RNNs with and without various modular architectural tweaks), the results show the impact of modular-constrained specialization (it helps!), and a small gap between “modular” and “monolithic” systems trained end-to-end (we need to do better at training modular systems!).

**Questions:**

- Nit: Could the title be a bit more descriptive? I understand the desire for something short and punchy, but this paper does a lot of really cool stuff that should be expressed in the title?
- Perhaps something like “Evaluating Learning & Generalization of Modular Inductive Biases in Neural Architectures through Rigorous Control Tasks”?

**Limitations:**

I believe this paper could do a better job of stating the limitations with respect to the fully synthetic nature of the proposed control tasks. These are absolutely useful; but there will always be a faction of scientists who want to see how real data (especially at scale) interacts with the story presented in this work!


**Strengths And Weaknesses:**

The strengths of this paper are in its clarity and simplicity. It sets out to rigorously test the abilities of sparse, modular architectures vs. the “monolithic” architectural equivalents — what can these modular architectures learn that monolithic architectures cannot? In an ideal world, are modular architectures better?

Being able to construct a simple process for generating data and evaluating these hypotheses is a strong contribution of this work; going further to test the various types of generalization, collapse modes, and carefully probe the “end-to-end” modular learning vs. an “oracle” learning are just additional strengths that really help contextualize what it happening.

The weakness of this paper is that there’s little analysis of the existing sparse-MoE models that are trained on tremendous amounts of natural data (e.g., Switch-Transformers, MoE Language Models). It’d be interesting to see if you can construct synthetic language tasks that capture the same type of modularity and show that even when fine-tuning (or zero/few-shot finetuning these existing base models), the existing failure modes still appear!

---

> ### Author Response · Authors · 2022-08-02
> **Official Rebuttal**
>
> We thank the reviewer for their insightful comments and recommendations for our work. We provide connections to real world tasks in a separate general comment above and take this opportunity to further address additional concerns about our work.
>
> **Analysis on existing sparse MoE models**
>
> We agree with the reviewer on the point of expanding this analysis to some form of synthetic language based tasks and the impact of MoE based systems on such tasks. While it is an important direction to pursue, we think that this would require careful construction of the mixture distributions in the synthetic language domain which is a contribution on its own, and would be best suited as a follow-up and is out of the scope of the current work. In addition, we feel that even performing analysis on either multiple language modeling or multilingual translation could be important avenues to explore and discover if perfect specialization at the language level is important or not, and then further to see how well MoE systems are able to specialize accordingly. Also on the note of existing MoE models, we can see the Recurrent Network experiments as a specific case of one object file and R schematas in the SCOFF model of Goyal et. al 2020; Object Files and Schemata: Factorizing Declarative and Procedural Knowledge in Dynamical Systems for a system with $R$ rules.
>
> Therefore, we want to reiterate that while the proposed analysis of MoE systems trained on large datasets is important, it would require two additional considerations: (a) careful design of mixture distributions in the synthetic language domain such that there is only one notion of specialization, and (b) design of tasks and mixture distributions that are related to each other so that analysis can be done on pre-training on a set of tasks and then fine-tuning on another set of different but related tasks. These would require a lot more deliberations on the design choices if done right, and we believe that they should be stand-alone contributions in themselves. Once a proper set of task distributions are decided, our metrics can then be readily usable for further analysis. We hope that the reviewer can still recognize the value of the current contribution as a first step to develop evaluation tools for network architectures, which we foresee to be further used and developed in the future.
>
> **Title**
>
> We thank the reviewer for the suggestions about the title but unfortunately it is out of our hands as it cannot be changed during the rebuttal stage of the submission.
>
> **Limitations**
>
> We thank the reviewer for pointing this out. While we cover some of the limitations of our current analysis in the Future Work Section in Appendix A, we understand that we could have done a better job at articulating the limitations of the analysis done. To rectify this, we update the Appendix with additional details on Limitations (Appendix A), outlining the synthetic nature of the current tasks as well as the possible complexities that can be faced when extrapolating to more complex domains (Appendix B).
>
> We hope that the additional details and clarifications provided help paint a clearer picture of our work and resolve the reviewer’s concerns. We would also be happy to address any additional questions that the reviewer may have.

---

> > ### Author Response · Authors · 2022-08-07
> > **Request for Discussion**
> >
> > Given that the author-reviewer discussion period is coming to a close soon, we request the reviewer to let us know if our responses have resolved their concerns and if there are any other questions that we can address.

---

> > > ### Comment · Reviewer_H4Mv · 2022-08-08
> > > **Rebuttal Response**
> > >
> > > Thanks so much for your rebuttal! I understand your points, and believe this paper to have merits - I think it should be accepted, and my score currently reflects that!
> > >
> > > Hoping that the other reviewers can similarly see the merits of this work!

---

### Official Review · Reviewer_HQDS · 2022-07-11

**Rating:** 8
**Confidence:** 4
**Soundness:** 4 excellent
**Presentation:** 3 good
**Contribution:** 4 excellent

**Summary:**

The submission is a comprehensive rethinking and assessment of the research in modular network. It develops a series of benchmarks and metrics to evaluate the benefit of existing works using modular architecture. Specifically, It performs experiments on four kinds of model corresponding to different levels of specialization and obtains some empirical findings about the design of modular network.

**Questions:**

It seems that the description is very high-level and the implementation detail is omitted. For example, in modular setting, how is the confidence score computed? Similarly, in modular-op setting, how to decide which module to evoke (we only know it is decided on $\mathbf{c}$)?

**Limitations:**

limitations are non-applicable / adequately addressed.

**Strengths And Weaknesses:**

Strengths:

1. The submission is a pioneer in systematically evaluating the performance of modular network in a unified framework.
2. Modular network is a heated topic attracting wide interest, and the work is of great significance to the community.

Weaknesses:

None

---

> ### Author Response · Authors · 2022-08-02
> **Official Rebuttal**
>
> We thank the reviewer for their praise concerning the significance of our work, and their comments on the lack of implementation details. To address this, we have provided detailed implementation details as well as extensions and connections to real-world settings in separate common comments above. We have also revised the Appendix to reflect this update and provide code for our experiments along with the submission. We plan to open-source our code for the community in order to ease reproducibility and help conduct further analysis. We hope that the implementation details clarify the reviewer’s doubts, and we would be happy to address any additional questions that the reviewer may have.

---

> > ### Author Response · Authors · 2022-08-07
> > **Request for Discussion**
> >
> > Given that the author-reviewer discussion period is coming to a close soon, we request the reviewer to let us know if our responses have resolved their concerns and if there are any other questions that we can address.

---

### Author Response · Authors · 2022-08-02
**Implementation Details (1/2)**

To address commonly asked questions about implementation details, we provide an overview of our framework below. In addition, we considerably improve the presentation of such details in the text with precise reference to Appendices to facilitate readability. A fluid description of this setup with a balance between details and readability is difficult to produce. We appreciate the comments and suggestions from all reviewers which considerably improves our contribution.

**Multi-layer Perceptron (MLP)**

Input consists of numbers $x_1 \in \mathbb{R}$ and $x_2 \in \mathbb{R}$ as well as the rule context $c \in \\{0, …, R\\}$, with $R$ being the total number of rules. The model consists of two encoders $E_x$ and $E_c$ where $E_x$ maps $x_1$ and $x_2$ independently to $\mathbb{R}^{d}$ and $E_c$ maps $c$ to $\mathbb{R}^{d}$. Each of the encoders are implemented as non-linear neural networks with a single hidden layer. The encoded inputs are then concatenated together and fed to a model chosen from Monolithic, Modular, Modular-op and GT-Modular. The output of this model lies in $\mathbb{R}^d$ and is fed to a non-linear decoder with a single hidden layer to provide the final prediction $\hat{y}$.

*Monolithic*: This model consists of a non-linear single layered neural network that gets the concatenation of the three encodings as input and outputs a vector in $\mathbb{R}^d$.

*Modular*: This model consists of R different non-linear single layered neural networks (modules), each of which gets the concatenation of the three encodings as input and outputs a corresponding activation score $p_m$ and prospective output $h_m$. The actual output of this modular system can be understood as $\sum_m p_m h_m$ which incorporates the output of each module in a soft manner as $p_m$ is obtained through a softmax. This output is then fed to a decoder, as in the other models.

*Modular-op*: This model is quite similar to the Modular system, with the only difference being that $p_m$ is not obtained from each module’s computations but instead from a separate non-linear network with one hidden layer which gets the encoding of c as input and outputs a probability vector $p \in \Delta_R$, i.e. the activation probability of each module.

*GT-Modular*: This model is also quite similar to the Modular system, with the only difference being that $p_m = 1$ if m is equal to c, otherwise $p_m = 0$. Thus, there is a unique sparse one-to-one correspondence between rule context and module selection. It can also be thought of as a Modular-op model with the separate network being an identity mapping.

For our experiments, we ablate over the encoding dimension $d$ and the hidden layer size of the model over the set \{(32, 128), (64,256), (128, 512), (256, 1024), (512, 2048)\} and control for the number of parameters between the four different kinds of models considered.

Details about the task setup and model training can be found in Appendix D. We have also updated Appendix D with the above additional details about the models.

**Multi-head Attention (MHA)**

Input consists of a set of vectors $\\{v_i\\}^N_{i=1}$ where each vector $v_i$ is of dimensionality $4R$ for Search-Version 1 and $6R$ for Search-Version 2 (Appendix E) as well as a set of rule contexts $\\{c_i\\}_{i=1}^N$ where $c_i \in \{0, …, R\}$ with $R$ being the total number of rules. As in the MLP setup, we first use a single layered non-linear feed-forward network to independently encode each tuple $(v_i, c_i)$ to some latent space of dimension $d$. The encoded input then goes through a choice of model ranging from Monolithic, Modular, Modular-op and GT-Modular which gives an output in $\mathbb{R}^d$ which is then fed to a single-layered non-linear feed-forward decoder network to give the final prediction $\\{ \hat{y}_i \\}$ with $i=1..N$.

*Monolithic*: This model consists of a single Multi-Head Attention block with $2R$ heads that gets the encoded input set and outputs a corresponding set of vectors in $\mathbb{R}^d$. We keep the number of heads as $2R$ to allow for learning for all rules, as each rule requires 2 heads.

*Modular*: This model consists of $R$ different Multi-Head Attention blocks (modules) with 2 heads each, each of which gets the encoded input set and outputs a corresponding activation score $p_{i,m}$ and prospective output $h_{i,m}$ for the $i^{th}$ token. The actual output of this modular system at each token can be understood as $\sum_m p_{i,m} h_{i,m}$ which incorporates the output of each module in a soft manner as $p_{i,m}$ is obtained through a softmax. This output is then fed to a decoder, as in the other models.

---

> ### Author Response · Authors · 2022-08-02
> **Implementation Details (2/2)**
>
> *Modular-op*: This model is quite similar to the Modular system, with the only difference being that $p_{i,m}$ is not obtained from each module’s computations but instead from a separate non-linear single-layered feed-forward network which gets $c_i$ as input and outputs a probability vector $p_i \in \Delta_R$ for each token $i$, i.e. the activation probability of each module for each token.
>
> *GT-Modular*: This model is also quite similar to the Modular system, with the only difference being that $p_{i,m} = 1$ if $m$ is equal to $c_i$, otherwise $p_{i,m} = 0$. Thus, there is a unique sparse one-to-one correspondence between rule context and module selection. It can also be thought of as a Modular-op model with the separate network being an identity mapping.
>
> For our experiments, we ablate over the encoding dimension $d$ and the hidden size which defines the heads dimensionality of the model over the sets \{(32, 128), (64,256), (128, 512), (256, 1024), (512, 2048)\} and control for the number of parameters between the four different kinds of models considered.
>
> Details about the task setup and model training can be found in Appendix E. We have also updated Appendix E with the above additional details about the models.
>
> **Recurrent Neural Network (RNN)**
>
> Input consists of a sequence of vectors $\\{v_i\\}^N_{i=1}$ where each vector $v_i$ is of dimensionality 32, as well as a set of rule contexts $\\{c_i\\}_{i=1}^N$ where $c_i \in \{0, …, R\}$ with $R$ being the total number of rules. As in the MLP setup, we first use a single layered non-linear feed-forward network to independently encode each tuple $(v_i, c_i)$ to some latent space of dimension d. The encoded input then goes through a choice of model ranging from Monolithic, Modular, Modular-op and GT-Modular which gives an output in $\mathbb{R}^d$ which is then fed to a single-layered non-linear feed-forward decoder network to give the final prediction $\\{ \hat{y}_i \\}$ with $i=1..N$.
>
> *Monolithic*: This model consists of a single LSTM Cell that gets the encoded sequence as input and outputs a corresponding sequence of vectors in $\mathbb{R}^d$.
>
> *Modular*: This model consists of R different LSTM Cells (modules) each of which gets the encoded sequence as input and outputs a corresponding activation score $p_{i,m}$ and prospective output $h_{i,m}$ for each token. The actual output of this modular system at each token can be understood as $\sum_m p_{i,m} h_{i,m}$ which incorporates the output of each module in a soft manner as $p_{i,m}$ is obtained through a softmax. This output is then fed to a decoder, as in the other models.
>
> *Modular-op*: This model is quite similar to the Modular system, with the only difference being that $p_{i,m}$ is not obtained from each module’s computations but instead from a separate non-linear single-layered feed-forward network which gets $c_i$ as input and outputs a probability vector $p_i \in \Delta_R$ for each token $i$, i.e. the activation probability of each module for each token.
>
> *GT-Modular*: This model is also quite similar to the Modular system, with the only difference being that $p_{i,m} = 1$ if $m$ is equal to $c_i$, otherwise $p_{i,m} = 0$. Thus, there is a unique sparse one-to-one correspondence between rule context and module selection. It can also be thought of as a Modular-op model with the separate network being an identity mapping.
>
> For our experiments, we ablate over the encoding dimension $d$ and the dimensionality that controls the hidden size of the RNN over the set \{(32, 128), (64,256), (128, 512), (256, 1024), (512, 2048)\} and control for the number of parameters between the four different kinds of models considered.
>
> Details about the task setup and model training can be found in Appendix F. We have also updated Appendix F with the above additional details about the models.
>
> We have also provided the code with the submission for ease of reproducibility and will be open-sourcing our code for the community to use.

---

### Author Response · Authors · 2022-08-02
**Impact, Extensions and Connections to Real-World Domains (1/2)**

We provide a detailed discussion about the impact of our analysis to real-world domains as well as additional considerations for researchers to take into account when considering the different real-world extensions proposed here.

**Understanding Large MoE Models**

MoE based models have also been shown to be quite successful in large-scale domains (Fedus et. al 2021, Shazeer et. al 2017, Lepikhin et. al 2020, Zuo et. al 2021, Wang et. al 2022). However, it is not clear whether they only offer ease of optimization or also benefits in performance through some notion of specialization. We believe it is an important research question to understand if their performance gains are linked to specialization, and if they are, how far are we from perfect specialization and how to reach there. To this end, we believe that our metrics can provide concrete quantitative assessment of the level of specialization obtained, and we believe that improving the capacity for specialization in our settings would extrapolate to more complex domains too. There are already some partial works that try to address the problems that we quantify; for example, the switch transformer uses a load-balancing term to prevent collapse. Certain works (Zuo et. al 2021, Wang et. al 2022) also show that context dependent routing often doesn’t provide additional benefits over random routing and one possible reason for this could be that context dependent routing is often severely sub-optimal in obtaining specialization as seen in our experiments.

* Switch Transformers: Scaling to Trillion Parameter Models with Simple and Efficient Sparsity; Fedus et. al 2021
* Outrageously Large Neural Networks: The Sparsely-Gated Mixture-of-Experts Layer; Shazeer et. al 2017
* GShard: Scaling Giant Models with Conditional Computation and Automatic Sharding; Lepikhin et. al 2020
* Taming Sparsely Activated Transformer with Stochastic Experts; Zuo et. al 2021
* AdaMix: Mixture-of-Adapter for Parameter-Efficient Tuning of Large Language Models; Wang et. al 2022)

In sum, consider our synthetic task setup along with evaluation metrics as an integrated tool for model architecture evaluation, rather than toy tasks. We think such architecture evaluation will be important to develop in the future, as networks start to exploit more modular structure. We consider our work as a systematic contribution toward this, helping to go beyond trial-and -error network design.

**Further Comments About Extensions to Real-World Settings**

The reason we consider synthetic settings is to have a very clear definition of specialization, that is, there is only one criteria which should drive specialization in our experiments and that is the rule context c. However, in more complex domains and multi-task settings, it is not so clear anymore. For example, between CIFAR10 and ImageNet classification, specialization could be at the level of dataset (CIFAR10 vs ImageNet) or at the level of object types (living vs non-living objects or ground vs water vs sky objects) or even at the lower level details (like presence of features like eyes, wings, wheels, etc.). Another example is Multilingual Language Modeling, where the notion of specialization could be tied to individual languages, or to different language families.

Even though the level of specialization is unclear in such complex domains, a possible way forward is taking a handful of notions of specialization and testing whether any of them leads to better performance in MoE models (eg. whether specialization at the level of language leads to better multilingual language modeling metrics in large MoE network styled as GT-Modular). That is, through GT-Modular and Modular-op styled MoE models, we can at least now test whether a designed notion of specialization is good for the task or not. Further, we can also extrapolate ways of improving specialization and reducing collapse from our synthetic domain to large-scale MoE systems which might lead not only to better performance but also more optimal sparse gating systems.

---

> ### Author Response · Authors · 2022-08-02
> **Impact, Extensions and Connections to Real-World Domains (2/2)**
>
> **Pre-training and Fine-tuning Extensions**
>
> It is also possible to extend this analysis to test for transfer ability of models by constructing a set of tasks to pre-train on and then another set of tasks to perform fine-tuning on, with the hypothesis that a well-specialized system should learn better or faster during fine-tuning. However, we would like to point out that this is not a simple extension since it also requires a clear notion of consistency/similarity between different rules. One could assume that training on certain rules and testing on completely unrelated rules is not of as much importance, and hence it requires a notion of similarity between tasks (eg. KL divergence between different mixture components as a notion of similarity; but it would require a move away from deterministic computations to noisy rules). Even after obtaining such a metric, it would provide another axis of study; i.e. how much similarity should be there between tasks for modularity to provide benefits. While an important question, we believe that it is a different research question from what we try to answer, which is the sub-optimality of modular systems in obtaining specialization.
>
> **Synthetic Language Task Extensions**
>
> Our setup can also be extended to testing of language models (LMs) by modeling the data distribution as some form of a mixture distribution in an underlying probabilistic context free grammar (pCFG) and analyzing whether current MoE systems specialize on the notion of experts in this setting.
>
> **Usage in Statistical Modeling and Neuroscience**
>
> Mixture distributions and Mixture of Experts based models have been widely used in Machine Learning and are applied in a number of real-world scenarios. They are often used to model statistical populations with subpopulations where each subpopulation could be modeled by a specific density and the mixture weights would reflect the proportion of each subpopulation. In this regard, we can look at our analysis at trying to determine in a general case of mixture distributions, how well can an MoE model discover the subpopulations, how can we evaluate it and whether it leads to any benefits in terms of performance.
>
> In the recurrent domain, connections of the proposed data-distribution and modeling assumption can be made with switching linear dynamical systems (sLDS) which have been shown to be widely successful in modeling non-stationary interactions between high-dimensional neural populations (Fox et. al 2008, Fox et. al 2010, Wulsin et. al 2013, Glaser et. al 2020). Our recurrent-based data is reflective of the modeling assumptions in sLDS and our RNN models can be seen as an implementation of a flexible mixture-of-experts based system in this domain, however without incorporating the bayesian or stochastic perspective which is an important next step as outlined in Appendix A. Since such works rely on learning to discover low-dimensional structure in neurons through mixtures, we believe that our analysis would benefit this direction of research too by quantifying the extent to which an expert orients with a subpopulation.
>
> * Nonparametric Bayesian Learning of Switching Linear Dynamical Systems; Fox et. al 2008
> * Bayesian Nonparametric Inference of Switching Dynamic Linear Models; Fox et. al 2010
> * Parsing Epileptic Events Using a Markov Switching Process for Correlated Time Series; Wulsin et. al 2013,
> * Recurrent Switching Dynamical Systems Models for Multiple Interacting Neural Populations; Glaser et. al 2020
>
> *Final Comments*
>
> We believe that the extensions that the reviewers have suggested (Synthetic Language tasks, Transfer ability, etc.) require careful consideration in the data-setup and would lead to stand-alone contributions in their own sense to answer questions that are different (but related) from the questions asked in this work. We are happy to incorporate discussions into these extensions in our Appendices A and B as important future work. In these sections, we also discuss limitations of our work and additional considerations that researchers would have to take into account when designing extensions along the provided directions.

---

### Meta-Review · Area_Chair_Nf6x · 2022-08-24

**Recommendation:** Accept
**Confidence:** Certain

**Metareview:**

This study investigates modular architectures, their properties, and their effectiveness in a class of synthetic yet informative scenarios. The reviewers unanimously recommend this paper for acceptance, some of them with high praise, and I enjoyed it as well: I suspect it will be read widely and have a lasting impact on our thinking about modularity.

**Award:**

Yes

---

### Decision · Program_Chairs · 2022-09-14

Accept